# Time-of-day defines NAD$^+$ efficacy to treat diet-induced metabolic disease by synchronizing the hepatic clock in mice

Quetzalcoatl Escalante-Covarrubias[1], Lucía Mendoza-Viveros[1,2], Mirna González-Suárez [1], Román Sitten-Olea [1], Laura A. Velázquez-Villegas [3], Fernando Becerril-Pérez[1], Ignacio Pacheco-Bernal[1], Erick Carreño-Vázquez[2], Paola Mass-Sánchez[1], Marcia Bustamante-Zepeda[1], Ricardo Orozco-Solís [2,4] & Lorena Aguilar-Arnal [1] ✉

The circadian clock is an endogenous time-tracking system that anticipates daily environmental changes. Misalignment of the clock can cause obesity, which is accompanied by reduced levels of the clock-controlled, rhythmic metabolite NAD$^+$. Increasing NAD$^+$ is becoming a therapy for metabolic dysfunction; however, the impact of daily NAD$^+$ fluctuations remains unknown. Here, we demonstrate that time-of-day determines the efficacy of NAD$^+$ treatment for diet-induced metabolic disease in mice. Increasing NAD$^+$ prior to the active phase in obese male mice ameliorated metabolic markers including body weight, glucose and insulin tolerance, hepatic inflammation and nutrient sensing pathways. However, raising NAD$^+$ immediately before the rest phase selectively compromised these responses. Remarkably, timed NAD$^+$ adjusted circadian oscillations of the liver clock until completely inverting its oscillatory phase when increased just before the rest period, resulting in misaligned molecular and behavioral rhythms in male and female mice. Our findings unveil the time-of-day dependence of NAD$^+$-based therapies and support a chronobiology-based approach.

In the last few decades, the prevalence of obesity has become epidemic through the world and is a major risk factor for type 2 diabetes (T2D)[1]. The main cause appears as combined inappropriate nutrition and sedentary lifestyles. Overweight, insulin resistance, β-cell dysfunction, increased circulating glucose and lipids and non-alcoholic fatty liver disease (NAFLD) characterize the pathophysiology of T2D[2]. Countless research efforts have explored pharmacological treatments for T2D and associated pathologies leading to promising compounds, which together with lifestyle interventions constitute first-line treatments[3].

During the last few years, the circadian system has been increasingly recognized as a key actor for development and treatment of diet-induced metabolic dysfunction. Yet, circadian rhythms in the clinical practice remain largely overlooked and time-of-day is hardly considered in treatment decisions[4–8].

Circadian rhythms are evolutionary conserved 24-h cycles in physiology dictated by an intrinsic circadian clock. In mammals, the suprachiasmatic nucleus (SCN), a master timekeeper in the hypothalamus, receives photic cues from the retina to align internal and

[1]Departamento de Biología Celular y Fisiología, Instituto de Investigaciones Biomédicas, Universidad Nacional Autónoma de México, 04510 Mexico City, Mexico. [2]Laboratorio de Cronobiología y Metabolismo, Instituto Nacional de Medicina Genómica, 14610 Mexico City, Mexico. [3]Departamento de Fisiología de la Nutrición, Instituto Nacional de Ciencias Médicas y Nutrición Salvador Zubirán, 14080 Mexico City, Mexico. [4]Centro de Investigación sobre el Envejecimiento, Centro de Investigación y de Estudios Avanzados, 14330 Mexico City, Mexico. ✉e-mail: loreaguilararnal@iibiomedicas.unam.mx

external time. The SCN distally synchronizes ancillary oscillators in peripheral tissues. Importantly, certain cues such as nutritional inputs effectively synchronize peripheral clocks[9]. Aligned synchrony between all body clocks maintains homeostasis and health, for example, by adjusting metabolic performance to daily environmental fluctuations. Conversely, persistent circadian misalignment is a cause of severe diseases, including obesity and metabolic syndrome, T2D, or cardiovascular disease, amongst others[10–12]. At the molecular level, the circadian machinery is expressed in almost all cell types and consists of transcriptional-translational autoregulatory feedback loops. The positive loop is driven by the CLOCK:BMAL1 transcriptional activator, which rhythmically binds to E-box genomic elements, thereby activating transcription of many genes including the circadian repressors, Period (*Per1-3*) and Cryptochrome (*Cry1-2*). PER:CRY complexes directly repress CLOCK:BMAL1 leading to transcriptional silencing. A number of interlocked regulatory loops, such as the one governed by RORs/REV-ERBα to regulate *Bmal1* expression, intertwine to confer complexity, redundancy, and robustness to circadian rhythms[13]. Consequently, a set of clock-controlled genes (CCGs) ranging from 5–25% depending on the tissue or cell type, display transcriptional circadian rhythms[14]. Notably, rhythmic transcripts are functionally related, including rate-limiting enzymes, hence providing means to adjust the pace of many metabolic pathways around the day and driving rhythms in the tissue metabolome[15–18]. A paradigmatic example is illustrated by daily rhythms in nicotinamide adenine dinucleotide (NAD+) bioavailability, imposed by circadian oscillations in the clock-controlled gene *Nampt*, the rate-limiting enzyme for the NAM salvage pathway to NAD+[19,20]. Several lines of evidence demonstrate that the molecular clock and NAD+ oscillations sustain mitochondrial function and bioenergetics, manifested in daily rhythms in respiration, fatty acid oxidation, or nutrient utilization[21–25]. Indeed, it is considered that clock-controlled NAD+ biosynthesis occupies a fundamental position connecting circadian metabolic pathways[26–28].

NAD+ and its phosphorylated and reduced forms NADH, NADP+, and NADPH, are coenzymes for hydride transfer enzymes, crucial to biological redox reactions. NAD+/NADH ratio is a basic determinant of the rate of catabolism and energy production[29,30]. In fed state or nutrient overload NAD+/NADH ratio falls, and a prolonged redox imbalance potentially leads to metabolic pathologies, such as diabetes[31]. Along these lines, extensive research demonstrates that NAD+ levels significantly decline in metabolic tissues of mice and patients with obesity[32–37]. NAD+ decay itself may contribute to metabolic dysfunction by distinct mechanisms, including increased oxidative stress and ROS production, disbalance in the oxidative-reductive capacity, disrupted $Ca^{2+}$ homeostasis, or reduced activity of sirtuins[38,39]; a class of deacetylase enzymes using NAD+ as cofactor and known to influence mitochondrial function and metabolism. In recent years, NAD+ has emerged as a target for the treatment of metabolic diseases, as boosting endogenous NAD+ levels has been proven effective against diet-induced metabolic pathologies, including insulin resistance, hyperglycemia, hypertriglyceridemia and NAFLD[32,33,35,36,40–45]. All these studies aim to increase NAD+ levels either genetically or pharmacologically, yet they mostly overlook the circadian trait of NAD+ bioavailability. Consequently, the implications of circadian rhythms in the function and effectiveness of NAD+ boosters as a therapy for diet-induced metabolic dysfunction remain largely obscure.

In this work, we aimed to characterize the metabolic consequences of rhythms in NAD+ levels. To approach this question, we used a mouse model of diet-induced obesity (DIO), which is known to present decreased, non-rhythmic levels of NAD+[15–17], and pharmacologically recovered daily rhythms of NAD+ with a peak at the onset of the active phase. To do so, we used a daily timed intraperitoneal (IP) injection with NAD+ itself at ZT11. We show that obese mice with enforced NAD+ oscillations improved metabolic health, significantly lost weight, and corrected NAFLD. Our analyses revealed that hepatic

transcriptional signatures of inflammation disappeared in these mice. Indeed, hepatic signaling involving AMPK, AKT, mTOR was rewired after restoring rhythmic NAD+ in obese mice, providing increased insulin sensitivity during the active period. Together, we demonstrated that a single daily injection with NAD+ treats the pathophysiology of diet-induced obesity, with comparable efficiency to NAD+ precursors. Remarkably, these metabolic and molecular improvements were not recapitulated by obese mice with antiphase increase of NAD+, at the onset of the rest phase, which showed only selective recovery of metabolic health. Further analyses demonstrated that lipid oxidative pathways and the molecular clock are central mediators for phase-dependent, differential effects of NAD+. Particularly, NAD+ provided at the onset of the rest phase uncoupled oscillations between central and peripheral clocks, by means of inverting the phase of the hepatic clock while food intake and activity remained rhythmic. Collectively, our findings reveal that timed NAD+ supply can shape the oscillatory phase of the hepatic molecular clock in vivo and expose a previously unappreciated time-dependent effect of NAD+ as a treatment for metabolic dysfunction, paving the way for chronotherapy and personalized medicine.

## Results
### A timed treatment with NAD+ reverses the metabolic phenotype of diet-induced obesity
To understand whether daily NAD+ administration improves metabolic fitness in obesity, we used a mouse model of diet-induced obesity (DIO) where instead of increasing NAD+ by chronic supplementation with metabolic precursors, we directly supplied the metabolite itself in a daily single IP injection scheduled at ZT11, corresponding to an hour before the normal circadian rise of hepatic NAD+[16,17,21,27,46]. Hence, after 8 weeks of high-fat diet (HFD) feeding, mice were treated for 22 days with saline solution (HF group) or 50 mg/Kg of NAD+ (HFN group, Supplementary Fig. S1a, see Methods section), at ZT11 (Fig. 1a). Mice fed a chow diet were included as a control (CD group).

At week 8 on HFD, mice displayed expected increase in body weight which was accompanied by significantly higher caloric intake during both light and dark periods[47] (Fig. 1b, Supplementary Fig. S1b–d). Notably, after 14 days of NAD+ chronotherapy, a significant decrease in total body weight was observed in obese treated mice (HFN) with respect to their obese non-treated littermates (HF), which was sustained after 22 days (Fig. 1b; $P < 0.05$, Two-way ANOVA, Tukey post-test). At the end of the treatment, hepatic NAD+ content was measured by HPLC, showing the expected oscillation with a peak at ~ZT12 in control mice (CD, Fig. 1c) which is mostly disrupted in HFD-fed mice (HF, Fig. 1c, Supplementary Fig. S1e)[16,21,36,46]. Importantly, in the HFN group, the acrophase of NAD+ was restored to ZT12 (HFN, Fig. 1c, Supplementary Fig. S1e), hence daily rhythms in hepatic NAD+ content was reinstated in obese mice (Supplementary Fig. S1d; $P < 0.001$, $F$ test performed with CircWave).

We sought to assess physiological indicators of metabolic health and found that circulating insulin levels were much lower in the HFN group when compared to the HF group, with a major effect during the early active phase (Fig. 1d, ZT12-18, $P < 0.001$ Two-way ANOVA, Tukey post-test) and a six-hour phase delay in the oscillatory pattern (Supplementary Fig. S1f). Indeed, circulating insulin in HFN mice appeared largely comparable to their control littermates. Overall, we didn´t find major differences in body temperature between treated and untreated obese mice, suggesting that circadian-controlled thermogenic processes[48] are probably not involved in the metabolic benefits observed upon restoring NAD+ oscillations (Supplementary Fig. S1g–j).

It has been extensively demonstrated that glucose tolerance and insulin sensitivity follow daily rhythms imposed by the circadian system[49], hence we evaluated them at two different time points, ZT4 and ZT16. As expected, before NAD+ treatment, HFD-fed mice showed impaired glucose tolerance at both ZT (Supplementary Fig. S1k, l).

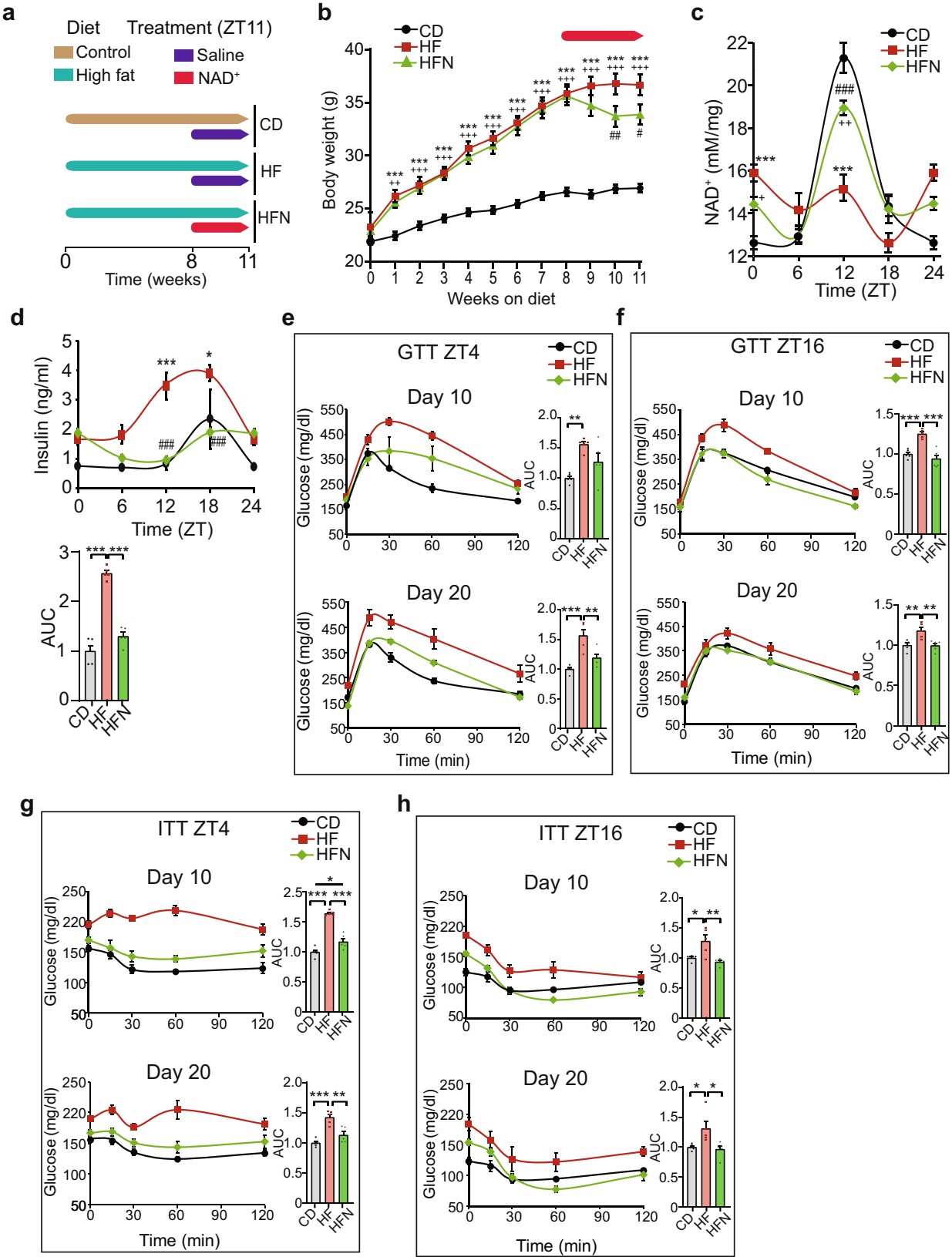

Remarkably, after 10 days, restoring NAD$^+$ oscillations in obese mice significantly ameliorated glucose tolerance, specifically at ZT16 (Fig. 1e, f, AUC HF vs HFN at day 10: $P < 0.001$, one-way ANOVA, Tukey post-test). After 20 days of treatment, this improvement was also apparent at ZT4 (Fig. 1e, AUC HF vs HFN at day 20: $P = 0.0084$, one-way ANOVA, Tukey post-test). As both insulin and glucose levels were lower in NAD$^+$-treated mice, insulin sensitization might occur. Accordingly, glucose clearance upon insulin IP injection was largely enhanced by NAD$^+$ chronotherapy (Fig. 1g, h, Supplementary Fig. S1m, n). Notably, this effect was already evident in the HFN group after 10 days of treatment independently of the time when measurements were performed (Fig. 1g, h, AUC HF vs HFN at day 10: $P < 0.005$, one-way

**Fig. 1 | A NAD⁺ chronotherapy at ZT11 improves the pathophysiology of diet-induced obesity. a** Schematic diagram of the study design. Mice were fed either a normocaloric diet (CD) or a high-fat diet for 11 weeks. At week 8, a subgroup of high-fat-fed mice was supplied a chronotherapy with NAD⁺, consisting of a daily intra-peritoneal injection of 50 mg/Kg of NAD⁺ at ZT11 for three weeks (HFN). The rest of the mice were injected with vehicle (saline solution). **b** Weekly body weight ($n = 20$ mice for CD and HFN and 17 for HF). Red arrow indicates the period of treatment with NAD⁺ or saline at ZT11. (**c**) Hepatic NAD⁺ content measured by HPLC along the day at the indicated times for all groups after the experimental paradigm ($n = 5$ biological replicates per time point and group, and three technical replicates). **d** Serum levels of insulin along the day at indicated times ($n = 5$ biological replicates

per time point, and 2 technical replicates). AUC: area under the curve. **e–h** Glucose (**e, f** GTT) and insulin (**g,h**; ITT) tolerance tests were performed at both the rest (ZT4) and the active (ZT16) period at the indicated days (10 and 20) after the beginning of treatments ($n = 5$ mice per point, except for HFN in **f**, where $n = 6$). AUC: area under the curve. CD control diet fed mice, HF high-fat diet fed mice, HFN high-fat diet fed, NAD⁺ treated mice at ZT11. Data represent mean ± SEM and were analyzed by two-way ANOVA using Tukey posttest, except when comparing AUC, where one-way ANOVA followed by Tukey's posttest was used. *$p < 0.05$, **$p < 0.01$, ***$p < 0.001$. $p$ values are provided in Supplementary Data 1. Points at ZT24 are duplicates of ZT0 replotted to show 24-h trends. Symbol key for comparisons: *CD vs HF; ⁺CD vs HFN; #HF vs HFN. See also Supplementary Fig. S1.

ANOVA, Tukey post-test). Interestingly, NAD⁺ treatment at ZT11 promoted a slight, albeit not significant, improvement in insulin tolerance with respect to control lean mice when tested at ZT16 (Fig. 1h). These results demonstrate that a chronotherapy with NAD⁺ injected just before starting the active phase improves glucose tolerance by increasing insulin sensitivity in DIO mice. Collectively, the restitution of NAD⁺ bioavailability at ZT12 recovers its basal hepatic oscillation and reverses the metabolic syndrome associated with diet-induced obesity.

Histological staining with Oil-Red-O (ORO) was used to semi-quantitatively assess hepatic steatosis (Fig. 2a), revealing that obese mice treated with NAD⁺ significantly decreased hepatic neutral lipid content (Fig. 2b, c, One-way ANOVA, Tukey's posttest). Furthermore, a quantitative assay specific for hepatic triglycerides, the major form of fatty acids storage, revealed that these were globally reduced in obese mice after restoring hepatic NAD⁺ oscillations (Fig. 2d, Two-way ANOVA, Tukey's posttest). Importantly, the NAD⁺ treatment recovered their oscillatory pattern which is generally disrupted in obese mice[16] (Fig. 2d, Supplementary Fig. S2). Additionally, this timed NAD⁺ therapy reduced the accumulation of carbonylated proteins in liver lysates to normal levels (Fig. 2e) and augmented mitochondrial biogenesis (Fig. 2f). Together, these results indicated that increasing hepatic NAD⁺ levels at ~ZT12 recovers glucose homeostasis and successfully restrains liver pathology and oxidative stress of HFD-fed mice.

At the molecular level, the master regulator of lipid metabolism PPARγ protein[50] was overexpressed across the day in the livers from HFD-fed mice, while those treated with NAD⁺ showed markedly reduced PPARγ levels (Fig. 2g). A similar trend was evidenced for the transcription factor CEBPα, a known positive regulator of *Pparγ* gene expression and adipogenesis[51,52] (Fig. 2g), further reinforcing the notion that a gene expression program involving lipid metabolism might be modified in NAD⁺ treated mice.

### Extensive transcriptional reorganization driven by timed NAD⁺ treatment

To address the extent of the transcriptional rewiring in the liver of NAD⁺-treated obese mice, we performed a transcriptomic analysis at light (ZT6) and dark (ZT18) phases in mouse livers from CD, HF, and HFN groups. 76 common genes were differentially expressed (DE) between day and night in all groups (Fig. 3a, Supplementary Data 2), with comparable expression levels. Amongst these, a number of transcripts related to circadian control were apparent, including *Clock*, *Arntl* (*Bmal1*), *Cry1*, *Nr1d2* (*Rev-Erbβ*), *Rorc*, *Tef*, *Nfil3* or *Ciart* (Fig. 3a, Supplementary Data 2), suggesting that circadian rhythms were mostly preserved by the NAD⁺ treatment at ZT11. Accordingly, rhythms in the core clock proteins BMAL1, CRY1, PER2, and REV-ERBα were overall sustained in the HFN group (Fig. 3b, c). Interestingly, a significant reduction in CRY1 protein levels at ZT12 was observed in the HFN group compared to the HF (Fig. 3b, c).

An extensive circadian transcriptional reprogramming is induced by high-fat diet in the liver[16,53], hereafter we identified 524 day-to-night DE transcripts in CD mice, 1684 in HF mice and 599 in the HFN mice (Fig. 3d, >1.25 fold-change, $P < 0.05$). Out of these, 322 transcripts were

exclusively fluctuating in the CD group, 1327 fluctuated solely in the HF, interestingly, 306 newly fluctuating transcripts appeared in the HFN group (Fig. 3d, e, Supplementary Data 2). Functional analyses revealed that indeed, many of these DE genes participated in shared biological processes including transport, metabolic processes and response to oxygen (Fig. 3f, Supplementary Data 2). As expected, day-to-night transitions in gene expression were more evident for genes implicated in lipid metabolism in HFD-fed mice independently of NAD⁺ treatment (Fig. 3g, Supplementary Data 2). Remarkably, a set of genes functionally related to immune system processes appeared significantly enriched solely in the HF mice (Fig. 3g, Supplementary Data 2). Interestingly, timed NAD⁺ supply imposed new and specific day-to-night transcriptional fluctuations in genes functionally related to response to stress and starvation (Fig. 3g, Supplementary Data 2). Hence, we reasoned that a time-of-day specific transcriptional response to NAD⁺ might be responsible for the beneficial effects of rhythmic restitution of this metabolite.

To further dissect the expression program imposed by NAD⁺, we identified DE genes between groups, examined specifically at day (ZT6) or night (ZT18). At ZT6, 724 hepatic transcripts were significantly DE between CD and HF mice, while 936 transcripts varied when comparing HF and HFN groups, with 182 (12%) overlapping transcripts (Fig. 4a, Supplementary Data 3). At ZT18, 1731 genes were DE in livers from CD and HF mice, and 698 were DE between HF and HFN mice, appearing 118 (5%) common transcripts (Fig. 4a, Supplementary Data 3). Interestingly, most of these DE shared transcripts recovered their expression in the obese NAD⁺ treated (HFN) mice to control conditions (Fig. 4b). Common DE genes between CD-HF and HF-HFN comparisons at ZT6 were specifically enriched for biological processes related to regulation of the immune response, including both innate and adaptive immune system pathways (Fig. 4c, Supplementary Data 3). Furthermore, a direct assessment for distinctive gene sets between HF and HFN groups at ZT6 using GSEA[54] (Gene Set Enrichment Analysis) showed that IL6-JAK-STAT3 and TGFβ signaling were the highest enriched hallmarks (Supplementary Fig. S3a). Indeed, timed NAD⁺ treatment in obese mice suppressed the hepatic expression of inflammatory markers including *Stat3*, *Stat6*, *Tgfb1*, *Il1r1*, *Il6st*, *Tnfrsf1a*, *Tnfrsf1b*, *Smad3* or *Smad6* (Supplementary Fig. S3b). This supports the notion that timed NAD⁺ treatment just before the onset of the active phase in obese mice abolishes the inflammatory environment associated with insulin resistance in NAFLD[55,56] specifically during their resting period. Accordingly, at ZT18, genes recovered to normal conditions by NAD⁺ appeared mostly enriched for Lipid Metabolic Processes, particularly fatty acid biosynthesis and storage (*Plin2*, *Abhd5*, *Acsm1*, *Hsd17b12*, *Chka*) (Fig. 4d, Supplementary Data 3). Furthermore, a GSEA comparing HF and HFN groups at ZT18 revealed highest enrichment in the hallmarks Cholesterol Homeostasis and MTORC1 Signaling (Supplementary Fig. S3c), with significant downregulation of transcripts coding for major regulatory proteins and rate-limiting enzymes triggered by de novo NAD⁺ oscillations (Supplementary Fig. S3d; Cholesterol Homeostasis: *Hmgcr*, *Hmgcs1*, *Sqle*, *Acss2*, *Lss* or *Stard4*; MTORC1 signaling: *Acaca*, *Acly*, *Me1*, *Adipor2*, *Psma3*, *Psma4*, *Psmd14* or *Psmc6*). As shown, these hepatic expression

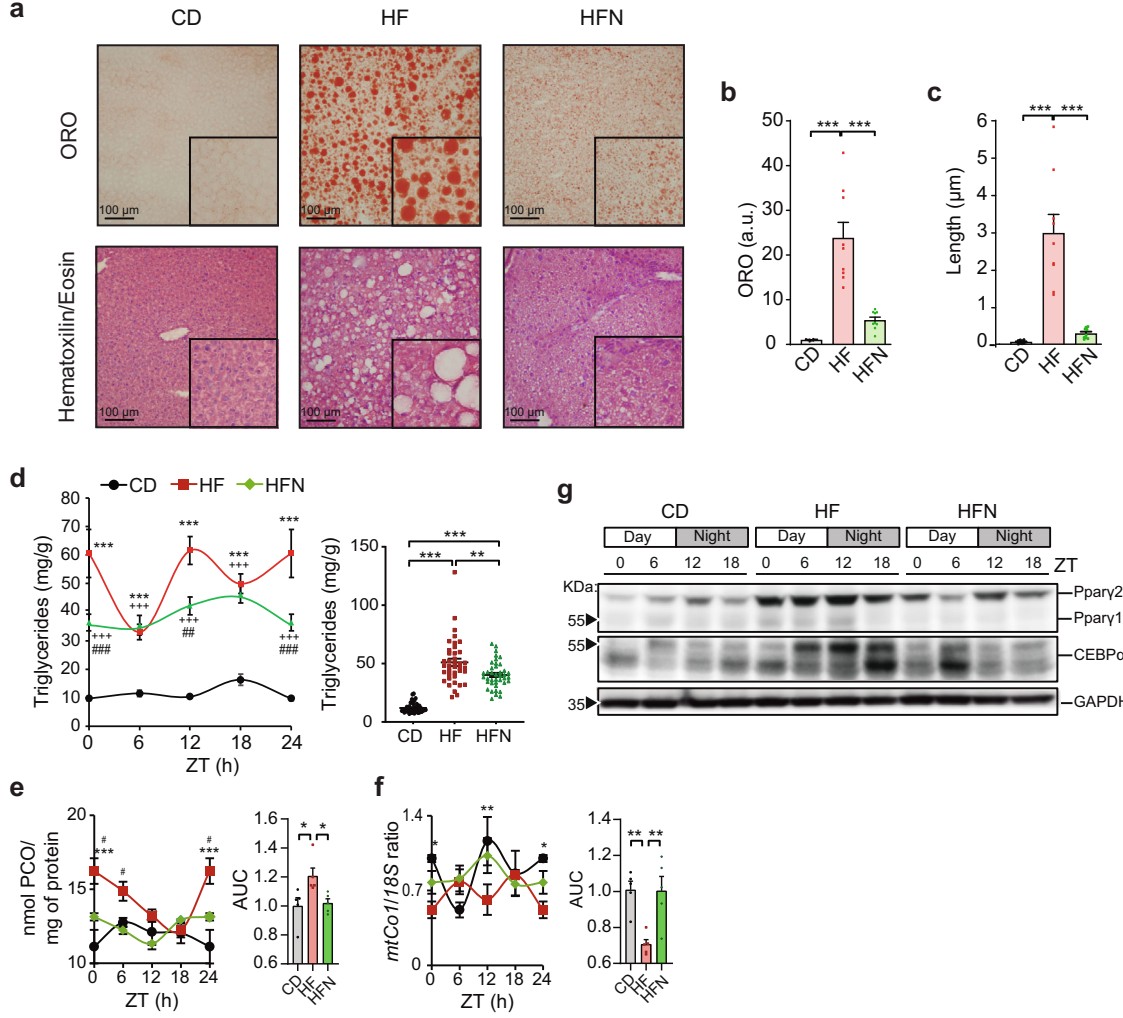

**Fig. 2 | NAD+ chronotherapy ameliorates NAFLD. a** Representative hepatic histopathology. Upper panel: Oil-red-O stain (ORO). Lower panel: hematoxilin/eosin. Images were acquired at ×20 optical magnification, and a detailed ×100 digital magnification is shown. Data were reproduced in three biological and three technical replicates. **b** Quantification of ORO signal (arbitrary units). Signal for control mice was set to 1 ($n = 3$ biological and three technical replicates). **c** The length of lipid droplets was compared between groups ($n = 3$ biological and three technical replicates). **d** Distribution of hepatic triglyceride content across the day (left), and comparisons from all measurements (right) ($n = 5$ mice per time-point, with 2 technical replicates). **e** Protein carbonyl levels (PCO) in liver lysates were measured at the indicated times of day ($n = 5$ mice per time point, two technical replicates). AUC area under the curve. **f** Relative mtDNA copies of mtCO1 respect to 18 S DNA measured by Real-time PCR ($n = 5$ biological and two technical replicates). Inset area under the curve, content from CD group was set to 1. **g** Western blot from PPARγ1, 2, and CEBPα proteins in the mouse liver at the indicated times of day (ZT). GAPDH was used as loading control (WB was performed from 3 biological replicates with comparable results). Uncropped blots in Source Data. CD Control diet fed mice, HF High-fat diet fed mice, HFN High-fat diet fed, NAD+ treated mice at ZT11. Data represent mean ± SEM and were analyzed by two-way ANOVA using Tukey posttest, except for bar graphs, where one-way ANOVA followed by Tukey's posttest was used. *$p < 0.05$, **$p < 0.01$, ***$p < 0.001$. Exact $p$ values are provided in Supplementary Data 1. Symbol key for comparisons: * CD vs HF; + CD vs HFN; # HF vs HFN. See also Supplementary Fig. S2.

changes at ZT18 were accompanied by improvement of hyperlipidemia and fatty liver traits after restoring NAD+ oscillations in obese mice (Fig. 2a–d). Pathway analyses revealed transcriptional mechanisms restituted by NAD+, with significant enrichment of NFkB, HIF1 and HNF3 transcriptional networks (Fig. 4d); and de novo motif discovery in the promoters of genes whose expression appeared dysregulated only in the HF group identified strong similarities to NFkB-p65/RELA and FOXA1/FOXA2 (HNFα/HNF3β) binding sites (Fig. 4e). Accordingly, out of 83 transcripts recovered by NAD+ at ZT18, 11 (13%) are previously described direct targets of FOXA2[57] (Supplementary Fig. S3e); interestingly, FOXA2 is a key regulator of lipid metabolism which becomes dysregulated in diabetic, insulin resistant mice[57,58]. Together, these data indicate that the inflammatory transcriptional signature related to NAFLD is abolished after timed NAD+ treatment, possibly through coordinating the action of transcription factors such as NFkB or FOXA2, and intracellular signaling involving the MTORC1 pathway.

## Insulin signaling and rhythmicity in nutrient sensing pathways are rescued by NAD+ oscillations in obese mice

Transcriptional networks uncovered in these analyses together with measurements of metabolic parameters are strongly suggestive of restored insulin sensitivity and nutrient-sensing molecular pathways after reestablishing NAD+ oscillations in obese mice. To confirm this at the molecular level, we first evaluated phosphorylation of AKT1, a key kinase effector of insulin signaling[59], along the day. As previously described, AKT1 phosphorylation at Ser 474 (p-AKT(S473)) appeared cyclic in CD fed mice[60], with a peak at ZT18 (Fig. 4f, Supplementary Fig. S4a), coincident with highest food intake (Fig. S1C). In contrast, in HFD-fed mice, p-AKT(S473) was constitutively low, suggestive of insulin resistance in the liver of obese mice (Figs. 4F, S4A). Remarkably, we found restoring hepatic NAD+ oscillations in obese mice specifically increased p-AKT(S473) at ZT12 (Fig. 4f, Supplementary Fig. S4a. $P < 0.0001$; two-way ANOVA, Tukey post-test), hence imposing daily

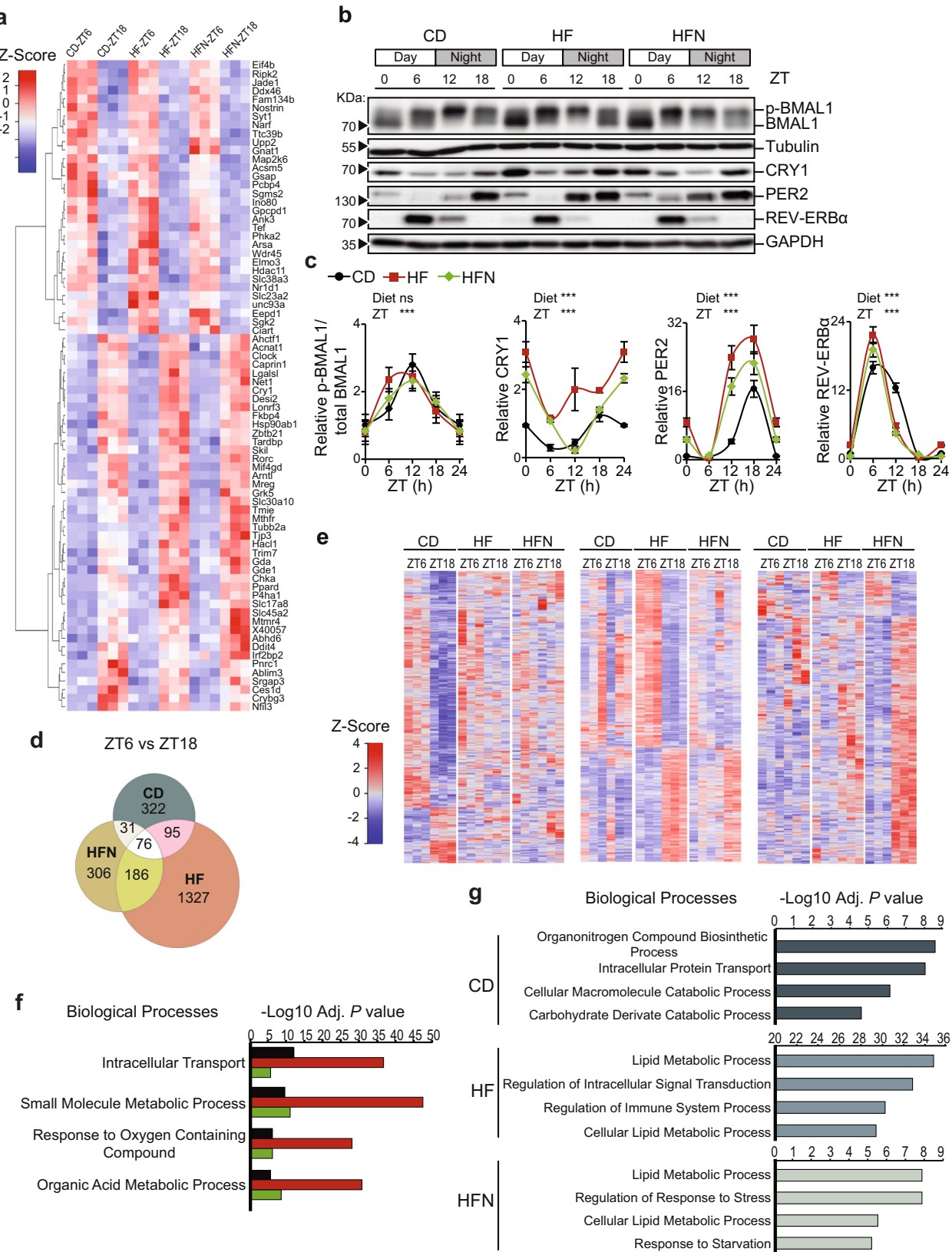

oscillations to insulin signaling. Furthermore, diurnal rhythms in AMPK phosphorylation at T172 were also restored by NAD$^+$ treatment in obese mice, although with a unique peak at ZT12, which was six hours phase delayed compared to their control, lean littermates (Fig. 4f, Supplementary Fig. S4b). This is in keeping with our previous observation of a reduction of CRY1 protein in the HFN group specifically at

ZT12 (Fig. 3b, c), as AMPK rhythmically phosphorylates and destabilizes CRY1[61]. Concomitantly, the AMPK substrate ULK[62] appeared hyperphosphorylated in the livers of NAD$^+$ treated mice at ZT12 (Supplementary Fig. S4c). Following the lead from our transcriptomic analyses, we also explored mTORC1 function. mTOR S2448 phosphorylation and activity appear rhythmic in the mouse liver,

**Fig. 3 | Timed NAD⁺ supply induce a reprogramming of hepatic transcripts DE between day and night without altering the dynamics of clock proteins.**
**a** Heatmaps of 76 common differentially expressed (DE) transcripts between day (ZT6) and night (ZT18) in all groups. **b** Circadian protein expression of BMAL1, REV-ERBα, CRY1, and PER2 in the whole cell extracts from CD, HF, and HFN livers was determined by western blot. Tubulin or GAPDH were used as a loading control. Data were reproduced in three independent experiments. **c** Quantification of western blots from $n = 3$ mice except for BMAL1, where $n = 4$ mice. Measurements were normalized to the loading control, and data from CD at ZT0 was set to 1. Means ± SEM are presented. ***$p < 0.001$, Two-way ANOVA; n.s. non significative. Points at ZT24 are duplicates of ZT0 replotted to show 24-h trends. Statistical details and

$p$ values are provided in Supplementary Data 1. **d** Overlap of DE transcripts between day (ZT6) and night (ZT18) in all groups (FDR < 0.05; fold-change >1.3). **e** Heatmaps of distinct groups of DE transcripts between day (ZT6) and night (ZT18). Left: 322 transcripts DE exclusively in CD; center: 1327 DE exclusively in HF; right: 306 transcripts DE exclusively in HFN. **f** Shared biological processes for DE transcripts between day and night from all groups. **g** Non-shared biological process for DE transcripts between day and night. **f, g** The Adj. $P$ value corresponds to the FDR q-value. This is the false discovery rate analog of hypergeometric $p$ value after correction for multiple hypothesis testing according to Benjamini and Hochberg. CD control diet-fed mice, HF High-fat diet-fed mice, HFN High-fat diet fed mice, NAD⁺ treated at ZT11.

---

coordinating a number of functions around the day, including ribosome biogenesis[63,64] (Fig. 4g). High-fat feeding constitutively induced mTOR phosphorylation, and timed NAD⁺ treatment in obese mice downregulated it (Figs. 4G, S4D). We investigated the phosphorylation of p70-S6K (S6K) as a readout of the activity of mTORC1[65], and found that the diurnal profile of activation of S6K-Thr389 phosphorylation was completely restored by NAD⁺ chronotherapy in the livers of HFD-fed mice (Fig. 4g, Supplementary Fig. S4e). Additional mTOR downstream signaling revealed by phosphorylation of 4EBP1(Thr37/46) was reduced in obese, NAD⁺ treated mice (Fig. 4g, Supplementary Fig. S4f). We also observed that the mTORC1 agonist p90-S6K (RSK)[65,66] and its activity as monitored by its phosphorylation in the Thr359 were significantly downregulated in the HFN group respect to the HF (Fig. 4g, Supplementary Fig. S4g). These results reinforce our pathway and gene set enrichment analyses comparing HF and HFN groups, consistent with reduced function of mTORC1 pathway after recovering NAD⁺ oscillations in obese mice.

### A unique NAD⁺ transcriptional signature identifies new pathways linked to metabolic improvement

We sought to explore the transcriptional signature induced by oscillatory NAD⁺, by identifying DE genes specifically in the HFN group. We found just 74 genes changing their expression at ZT6, and 196 at ZT18 (Fig. 4h, i, Supplementary Data 4). Functional analyses did not retrieve any significant enrichment for these genes at ZT6; however, it became very evident that at ZT18, a large part of the DE genes after NAD⁺ treatment were overexpressed and functionally involved in intracellular vesicle transport and catabolic processes (Fig. 4i, j, Supplementary Data 4). Indeed, five members of the Rab family of small GTPases, know regulators of membrane trafficking[67], were specifically overexpressed after NAD⁺ treatment, including *Rab1b*, *Rab7a*, *Rab10* which are largely involved in mediating autophagy[68–71], and *Rab6a*, *Rab8a*, which also mediate receptor trafficking in response to insulin signaling[72,73]. Additional overexpressed genes by NAD⁺ known to regulate autophagy were *Psen1*[74], *Vps28*[75].

A search for de novo motif enrichment within the promoters of NAD⁺-induced genes yielded matrices with high similarity to the binding sites for NR2E1 (TLX) and HNF4α TFs, both implicated in maintaining lipid homeostasis in the liver[76,77]. Also, a motif recognized by IRF3 and NR4A1 (Nur77) appeared significantly enriched ($P = 1e-5$), and interestingly, Nur77 has been shown to regulate the cytoplasmic shuttling of LKB1, hereby phosphorylating and activating AMPK[78]. Together, these data indicate that oscillatory NAD⁺ in obese mice activates a gene expression program favoring processes highly demandant for vesicle trafficking, such as translocation of membrane receptors or autophagy, and reinforce the idea of pharmacological supply of NAD⁺ preferably targeting activation of AMPK even in the context of high caloric feeding.

### Time-of-day determines the efficacy of NAD⁺ as a treatment for diet-induced metabolic dysfunction

To investigate if the beneficial effects of pharmacological restitution of NAD⁺ oscillations depend on the time of the day, we supplied NAD⁺ in

opposite phase to its natural rhythmicity, hereby at the end of the active phase in mice, ZT23 (HFN23 group). In these HFN23 mice, oscillations of hepatic NAD⁺ were induced with antiphase respect to CD and HFN mice, showing a peak at ZT0 and decreasing at ZT12-18 (Supplementary Fig. S5a, b). As shown in mice treated at ZT11 (HFN), these also showed mild, albeit non-significant, weight loss after one week of treatment (Fig. 5a, week 9). Contrary to the HFN group, mice supplied at ZT23 gained weight during weeks 10 and 11 (Fig. 5a). Instead, after three weeks of treatment, mice treated with NAD⁺ at ZT11 had lost ~5% of body weight, while those treated at ZT23 were ~2% heavier (Fig. 5b), illustrating significant differences on the efficacy of the treatment depending on the time of administration. Notably, total food intake was comparable for all high-fat-fed mice, and before and after the treatment no significant differences were found (Fig. 5c, Supplementary Fig. S5c, Two-way ANOVA with post-test). Serum insulin was significantly higher in mice injected at ZT23 particularly during the dark phase (Fig. 5d, Supplementary Fig. S5d), indicating insulin resistance in these mice, although NAD⁺ therapy was effective to reduce fasting serum glucose independently of the time of supply (Supplementary Fig. S5e). These results indicate that in obese mice treated with NAD⁺ at ZT23, insulin clearance or the feedback inhibition of insulin secretion are impaired, which is a sign of persistent metabolic dysfunction in these mice[79]. Along these lines, we performed GTT and ITT at ZT4, because the effects of NAD⁺ supply at ZT11 tended to be more pronounced during the light phase (Fig. 1e–h). We found that at the end of the treatment with NAD⁺ at ZT23 (day 20, HFN23), glucose and insulin tolerance showed non-significant improvement compared to the HF-fed mice. Actually, the NAD⁺ treatment at ZT11 was significantly more favorable to improving glucose homeostasis than at ZT23 (Fig. 5e, f, Supplementary Fig. S5f, g; One-way ANOVA followed by Tukey's posttest). Quantification of the relative improvement to the obese non-treated mice showed that after 10 days of treatment, NAD⁺ was effective to improve GTT and ITT only when supplied at ZT11, but not at ZT23. At the end of the treatment (day 20), NAD⁺ supply at ZT11 showed still a significantly better performance than at ZT23 (Fig. 5g, Two-way ANOVA followed by Tukey's posttest).

Circulating triglycerides, largely known to be reduced by the NAD⁺ precursor niacin[80,81], were decreased along the day to normal levels by NAD⁺ only when the treatment was performed at ZT11, but not at ZT23 (Fig. 5h). Interestingly, serum triglycerides were rhythmic for all groups; yet specific to the HFN23 group was that highest levels appeared at daytime, thus presenting antiphase daily oscillations (Fig. 5h). We found a very significant reduction in serum triglycerides only when NAD⁺ was injected at ZT11, while injection at ZT23 kept serum triglyceride levels significantly higher than injection at ZT11 (Fig. 5h, $P < 0.05$; One-way ANOVA with Tukey's post-test). Opposite, hepatic steatosis was reduced to a similar extent in HFN and HFN23 groups (Fig. 5i–k, Supplementary Fig. S5g–i), while inflammatory cytokines known to be increased in the liver from HFD-fed mice[82,83], appeared overall reduced in obese mice treated with NAD⁺, accompanied by decreased expression of pro-inflammatory cytokine genes (*Tgfb1*, *Il6*, *Il1a*, *Il1b*, *Ifng*), the macrophage recruiter gene *Csf2*, (also known as GM-CSF) and macrophage markers (*Cd11b*, *Cd11c*) (Fig. 5l–m), evidencing that NAD⁺

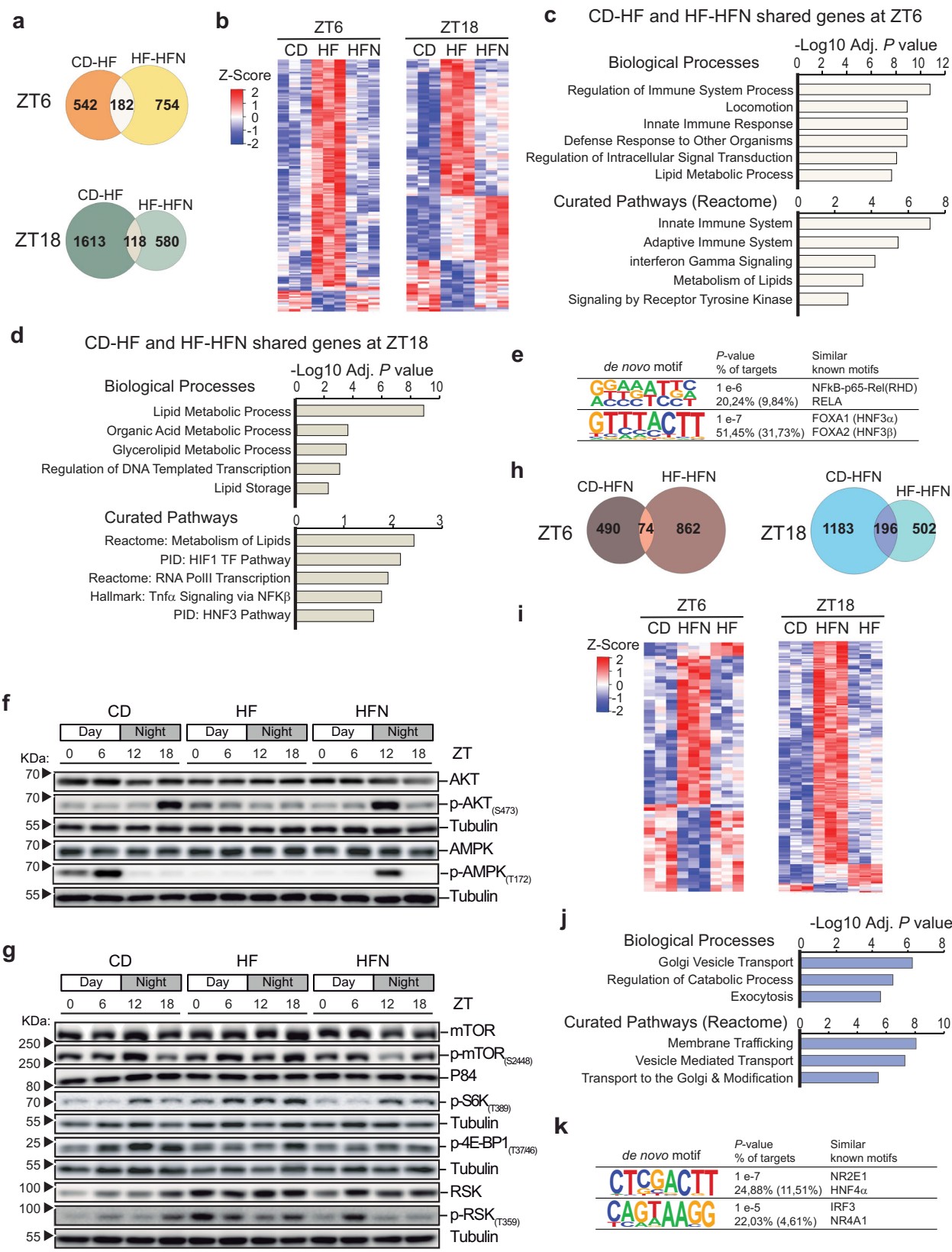

ameliorates these obesity-associated physiological parameters independently of the time of treatment. However, we observed opposite daily dynamics in hepatic PPARγ protein levels and its transcriptional activator CEBPα (Fig. 5n–o), which together with the serum triglycerides analyses, suggest that lipid metabolism might be distinct between HFN and HFN23 groups.

**NAD⁺ chronotherapy at ZT11 effectively coordinates hepatic intracellular signaling and gene expression driving lipid oxidation**

To further disentangle the molecular pathways responsible for the physiological differences in glucose and insulin tolerance, and circulating triglycerides, between HFN and HFN23 groups of obese mice, we

**Fig. 4 | A NAD+ chronotherapy at ZT11 corrects abnormal gene and protein expression from crucial molecular effectors of liver disease and triggers a specific transcriptional signature. a, b** Overlap (**a**) and heatmap (**b**) of DE genes when comparing CD-HF and HF-HFN groups either at day (ZT6) or ant night (ZT18) (*n* = 3). **c, d** Functional annotation of CD-HF and HF-HFN shared genes at daytime (**c**) or nighttime (**d**). **e** Homer de novo motif discovery analyses from promoters of genes DE exclusively in the HF group. **f, g** Circadian protein expression of AKT, p-AKT(S473), AMPK, and p-AMPK(T172) (F) or the mTOR pathway (g) in whole cell extracts from CD, HF, and HFN livers determined by western blot. Tubulin or p84 were used as a loading control. Images represent 3–4 independent experiments.

Uncropped blots in Source Data. **h, i** Overlap (**h**) and heatmaps (**i**) of DE genes when comparing CD-HFN and HF-HFN groups either at day (ZT6) or at night (ZT18) (*n* = 3). **j** Functional annotation of shared genes DE in analyses CD-HFN and HF-HFN at nighttime. **k** Homer de novo motif discovery analyses from promoters of genes whose expression is altered exclusively in the HFN group. Adj. *P* value corresponds to the FDR q-value. This is the false discovery rate analog of hypergeometric *p* value after correction for multiple hypothesis testing according to Benjamini and Hochberg. CD control diet fed mice, HF high-fat diet fed mice, HFN High-fat diet fed, NAD+ treated mice at ZT11. See also Supplementary Figs. S3, S4 and Supplementary Data 4.

compared nutrient-sensing signaling in the liver from these mice. Western blot experiments showed that providing NAD+ at ZT23 to obese mice did not recapitulate hepatic AKT phosphorylation and activity, as did at ZT11 (Fig. 6a–b), hereby confirming that insulin signaling remains defective in mice treated at ZT23, as suggested by the ITT (Fig. 5f, Supplementary Fig. S5b). Additionally, the response to starvation signaling converging into AMPK-T172 phosphorylation and subsequent activation triggered at ZT12 after reinstating NAD+ oscillations was not induced in the livers of the HFN23 group (Fig. 6a–s). Furthermore, nutrient sensing by mTOR pathway appeared active through the day in livers from HFN23 group, as shown by persistent phosphorylation of p70-S6K-T389 (Fig. 6c–d), and contrasting with the rhythmic pattern observed in the HFN11 group. Moreover, RSK-T359 appeared hyperphosphorylated in the HFN23 group, also showing antiphase dynamics compared with the HFN group (Fig. 6c–d). These data clearly show that increased NAD+ levels at the end of the activity period are less efficient in synchronizing mTOR signaling pathway than high NAD+ at the onset of activity, and reinforce the notion of a chronotherapeutic approach as the best therapy for the treatment of metabolic diseases by NAD+ boosters.

It is widely accepted that AMPK regulates lipid metabolism through phosphorylation of acetyl-CoA carboxylase 1 (ACC1) at Ser79 and ACC2 at Ser212. These in turn downregulate the production of malonyl-CoA, the major substrate for fatty acid synthase (FAS) and a strong inhibitor of carnitine palmitoyl transferase 1 (CPT-1). Consequently, fatty acid synthesis is suppressed in favor of lipid oxidation, partially through activation of the rate-limiting step sustained by CPT-1[84]. Additionally, increased fatty acid oxidation has been largely recognized as a major metabolic outcome after pharmacological increase of NAD+[40,85], and this process appears rhythmic in mouse liver with increased rate near the end of the rest period[21]. Also, our gene expression data revealed a unique NAD+ transcriptional signature involving genes pertaining to catabolic processes at ZT18 (Fig. 4j). Hence, we sought to explore the diurnal transcriptional profile of genes involved in lipid oxidation. Selected transcripts from the microarray data and the key rate-limiting enzymes *Cpt1a, Cpt2, Acox1, Abcd1* were quantified in the livers from all groups (Fig. 6e–g). As expected, we found that genes involved in β-oxidation, either mitochondrial (*Cpt1a, Cpt2, Acot2, Crat, Acaa1b, Acsm1, Echs1*; Fig. 6E) or peroxisomal (*Acox1, Abcd1, Slc27a2*; Fig. 6f), and in ω-oxidation (*Cyp4a10, Cyp4a14, Cyp4a31*; Fig. 6g) were globally overexpressed in HFD-fed mice compared to lean mice[86]. Interestingly, almost all genes were significantly overexpressed specifically at ZT18 in obese mice treated with NAD+ at ZT11 (HFN), but not at ZT23 (HFN23). Indeed, fatty acid oxidation-related genes were highly expressed at the end of the rest period (-ZT12)[86]; yet, unique to the HFN group was that the breadth of transcriptional activity further extended through the active period, reaching significantly higher levels than in the non-treated, obese mice (HF) at ZT18 (Fig. 6e–g, *P* < *0.05, **0.01, ***0.001 Two-way ANOVA with Tukey post-test). Hereby, expression of these genes at ZT18 was altered depending on the time of NAD+ treatment, in a way that the treatment at ZT11 significantly enhanced their expression, whereas in mice treated at ZT23, expression was significantly reduced to levels largely comparable to the CD littermates (Supplementary

Fig. S6c, One-way ANOVA with Tukey posttest). Accordingly, housekeeping genes *Tbp* and *Rplp* presented no significant variations (Supplementary Fig. S6d). To address whether these transcriptional changes in HFN mice are functional, we measured global fatty acid oxidation rates and CPT-1-dependent mitochondrial respiration in liver explants. Notably, we found a significant induction of fatty acid oxidation in HFN mice (Fig. 6h), which was accompanied by increased CPT-1-mediated mitochondrial respiration when palmitoyl-CoA was supplied as a substrate (Fig. 6i). Accordingly, a significant rise in maximal respiration and ATP production was evident in mitochondria from HFN mice (Fig. 6j), demonstrating functional implications for transcriptional variations from fatty acid oxidation genes in HFN mice. Together, these data indicate that increased hepatic NAD+ levels at the beginning of the active phase induce AMPK phosphorylation and activity, favoring a transcriptional program of genes involved in fatty acid oxidation which extends through the active phase, increasing mitochondrial respiration and fatty acid consumption capacities, and possibly contributing to weight loss and decreased hepatic and circulating triglycerides specifically in HFN mice.

While obese mice treated with NAD+ at ZT23 presented some metabolic ameliorations mostly consisting of improved basal circulating glucose levels and reduced hepatic steatosis and inflammatory markers, we did not find consistent changes in gene expression or nutrient sensing signaling. Intriguingly, our microarray data showed that transcripts with highest fold-change after NAD+ treatment were Metallothionein 1 and 2 (*Mt1* and *Mt2*), two antioxidants and longevity regulators known to protect from HFD-induced obesity[87–89], and these transcripts were significantly more overexpressed in obese mice treated with NAD+ at ZT23 (Supplementary Fig. S6e, *P* < 0.001 HFN vs HFN23; Two-Way ANOVA with Tukey´s posttest). A similar case was found for the gene lipocalin 2 (*Lcn2*), which encodes for a secreted protein protective against NAFLD[90]. Hence, while NAD+ chronotherapy works optimally at ZT11, its supply at ZT23 induces distinct protective pathways responsible for a mild, albeit noticeable, improvement of HFD-induced metabolic disease.

## Timed NAD+ treatment reorganizes the hepatic clock

Chronotherapy with NAD+ at ZT11 and ZT23 led to significantly different consequences in metabolic fitness and daily gene expression in the liver of obese mice. Hence, we reasoned that the molecular clock might be responsible for daily variations in the effectiveness of the treatment. Thereby, we compared the hepatic clock protein expression along the day between obese mice treated at ZT11 and at ZT23 (Fig. 7a). Western blot analyses revealed a remarkable impact of NAD+ treatment at ZT23 in the dynamic expression of the clock proteins CRY1, PER2 and REV-ERBα, which displayed an almost complete antiphase dynamic (Fig. 7a, b). Concomitantly, BMAL1 phosphorylation was also 6–10 h phase-shifted by NAD+ treatment at ZT23, being higher at ZT0 (Fig. 7a, b, Two-Way ANOVA). Subsequently, we explored the expression of clock genes across the day (Fig. 7c). Strikingly, NAD+ treatment at ZT23 in obese mice induced a transcriptional rewiring of clock genes, whose expression almost perfectly mirrored that of the other groups, demonstrating that at ZT23, NAD+ synchronizes the hepatic clock genes' expression. Consequently, the average

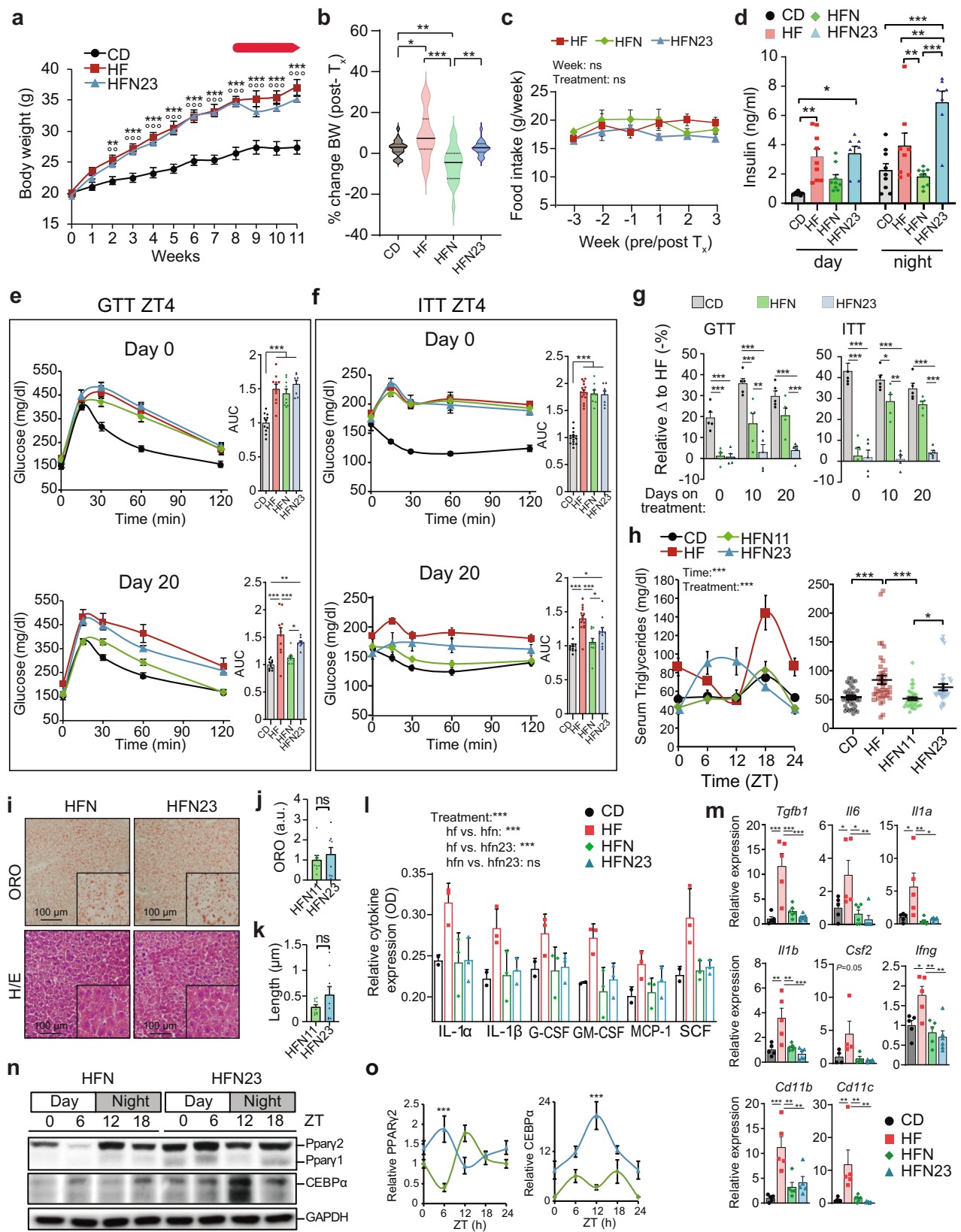

acrophases of the oscillations, defined as the highest point of the fitted wave by CircWave, were displaced by 10–12 h in the HFN23 group in all tested clock genes, both activators (*Bmal1, Clock*) and repressors (*Cry1, Per1, Per2,* and *Rev-Erbα*) (Fig. 7c, Supplementary Fig. S7a). Indeed, NAD⁺ chronotherapy did not compromise rhythmicity in clock gene expression (Supplementary Fig. S7a, $P < 0.05$; CircWave $F$ test).

To determine whether the observed antiphase dynamic of the clock transcriptional regulators was functional, we selected the genes *Dbp, Tef, Nfil3,* and *Noct,* whose expression is directly and mostly controlled by the core clock machinery, and analyzed their hepatic expression around the day (Fig. 7d). Coincident with the clock gene expression, the daily transcriptional profile of clock-controlled genes appeared

**Fig. 5 | Time-of-day dependent response to NAD+ treatment in obese mice determines the efficacy of the chronotherapy to improve glucose homeostasis.** **a** Weekly body weight ($n = 15$). Red arrow indicates the period of NAD+ or saline treatment at ZT23. **b** Percent change in body weight between weeks 8–11 ($n = 24$ CD, 23 HF, 19 HFN HFN23). **c** Weekly food intake three weeks before and after treatment ($n = 33$ mice). **d** Serum insulin ($n = 9$ mice per group, 7 for HFN23). **e, f** Glucose (GTT) and insulin (ITT) tolerance tests before and after (days 0–20) NAD+ treatment $n = 8$–16, see Supplementary Data 1 for exact $n$ **g** Relative delta for the AUC from GTT and ITT respect to HF group. ($n = 5$). **h** Daily serum triglycerides ($n = 10$ mice per time point, 9 for HFN23) (left), and comparisons from all measurements (right). **i** Representative hepatic histopathology. Upper: Oil-red-O stain. Lower: Hematoxilin/Eosin. Images acquired at ×20 optical magnification, detailed ×100 digital magnification is shown **j** ORO signal in arbitrary units. CD data were set to 1

**k** Length of lipid droplets ($n = 3$ biological, three technical replicates in **j** and **k**) **l** Relative pro-inflammatory cytokines in livers from 3 mice at ZT6 (2 CD mice were assayed for basal references). **m** RT-qPCR determined hepatic mRNA. $n = 5$ CD, HF; 6 HFN, HFN23, at ZT6. **n** Western blots from mouse liver **o** Quantification of WB from $n = 4$ HFN, 5 HFN23. Measurements were normalized to GAPDH, and data from CD at ZT0 was set to 1. Two-way ANOVA with Bonferroni post-test. CD control diet, HF High-fat diet, HFN High-fat diet, NAD+ treated at ZT11; HFN23: High-fat diet, NAD+ treated at ZT23 mice. For circadian plots, points at ZT24 are same as ZT0, replotted to show 24-h trends Data represent mean ± SE analyzed by one-way or two-way ANOVA using Tukey posttest. $*p < 0.05$, $**p < 0.01$, $***p < 0.001$; ns: non-significant. Exact p values are provided in Supplementary Data 1. * CD vs HF; °CD vs HFN23; ×HF vs HFN23. See also complementary Fig. S5.

rhythmic for all groups, and phase-inverted specifically in the obese mice treated with NAD+ at ZT23, with a significant phase shift of 11–13 h for *Dbp*, *Tef*, and *Noct*, and ~8 h for *Nfil3* expression (Fig. 7d, Supplementary Fig. S7e). To further reinforce the notion that NAD+ supply displaces the phase of circadian oscillations from the hepatic clock, we analyzed livers from obese mice treated at ZT5 (HFN5) or at ZT17 (HFN17). We found expected phase advance in clock protein oscillations in livers from HFN5 mice, while treatment at ZT17 resulted in phase delayed oscillations (Supplementary Fig. S7b, c) of distinct extents with respect to oscillations detected in HF mice (Supplementary Fig. S7c). This was paralleled by coherent phase shifts in clock and clock-controlled genes expression (Supplementary Fig. S7a, d). As expected, the phase shift was more evident in HFN17 than in HFN5, because the mammalian clock is reluctant to phase advances and more susceptible to phase delays[91–94]. Overall, these analyses show that timed NAD+ supply reshapes hepatic circadian oscillations of the clock machinery, adjusting their phase to the time of treatment.

Because redox rhythms regulate DNA binding of CLOCK:BMAL1 heterodimers in vitro[95], and the NAD+ precursor NR increases BMAL1 recruitment to chromatin in livers from aged mice[46], we hypothesized that inverted expression of clock genes in HFN23 might be driven by time-specific recruitment of BMAL1 to chromatin. To test this, we performed ChIP analyses to measure BMAL1 binding to regulatory E-boxes of clock and clock-controlled genes (Fig. 7e, f). As described[96], we observed increased recruitment of BMAL1 at ZT6 in livers from CD, HF, and HFN groups of mice for all tested E-boxes. Notably, in livers from HFN23 mice, BMAL1 binding appeared significantly increased at ZT18, consistent with inverted expression (Fig. 7e, f; $P < 0.05$, Two-way ANOVA with Sidak´s post-test). A non-related region at the 3' UTR region of *Dbp* gene was used as a negative control. We further evaluated the effect of NAD+ supplementation in the expression of NAD+ biosynthesis and salvage genes *Nmrk1*, *Nampt*, *Nmnat3*, and *Nadk* which also showed inverted phase specifically in HFN23 mice (Fig. 7g). Accordingly, BMAL1 binding to their regulatory elements was increased at ZT18 in HFN23 mice, yet specific to these group of genes was that NAD+ treatment significantly potentiated BMAL1 recruitment to chromatin. Finally, we explored the expression from TFs collaborating with the clock machinery to sustain a rhythmic transcriptional reprogramming in obesity[16,86]: *Pparg2*, *Ppara*, and *Srebf1c*. Transcription for these genes was phase-inverted specifically in HFN23 mice, which was also accompanied by differential BMAL1 chromatin recruitment (Fig. 7h). Also, expression levels of additional TFs related to hepatic lipid metabolism *Hnf4a*, *Foxa2*, *Foxo1*, and *Cebpa* were altered to a similar extent (Supplementary Fig. S8a). Antiphase expression of key transcription factors regulating hepatic lipid metabolism might underlie the inverted pattern of circulating triglycerides in HFN23 mice (Fig. 5h), but other lipids synthetized in the liver might be affected. Accordingly, hepatic cholesterol levels also showed a phase-inverted pattern in the liver of HFN23 mice (Supplementary Fig. S8b, c), reinforcing the idea that NAD+-mediated synchronization of transcriptional rhythms in the liver inverts hepatic lipid metabolism.

Together, this data demonstrates a time-dependent transcriptional response to NAD+ therapy in the liver of obese mice, through the synchronization of BMAL1 recruitment to chromatin and rhythmic transcription of clock and clock-controlled genes. Hereby, BMAL1 plays a pivotal role translating fluctuations in NAD+ levels to shape circadian transcription.

### Feeding and locomotor activities remain largely aligned to light in NAD+-treated mice

A phase-inverted hepatic clock has been previously shown for mice subjected to inverted feeding rhythms, where the SCN clock remains aligned to light-dark cycles[97,98]. At this regard, in all tested groups of mice, clock gene expression in the SCN remained largely intact after NAD+ treatment at two selected circadian times (Fig. 8a, Two-way ANOVA), and locomotor behavior analyses showed that overall, NAD+ treatment preserved alignment between light-dark and rest-activity patterns (Fig. 8b). Quantification of locomotion in 30 min bins revealed that after NAD+ treatment, mice became significantly less active for either 90 min (HFN) or 30 min (HFN_23) windows (Fig. 8c, Supplementary Fig. S8d, Two-way ANOVA followed by Sidak's posttest). We next questioned whether feeding cycles might be altered by NAD+ treatment because as previously reported, this is a cause for uncoupled central and peripheral clocks[97,98], while NAD+ itself can influence feeding behavior through implicated hypothalamic circuits[99,100]. Notably, daily food intake appeared rhythmic and aligned to light-dark cycles for all groups of HF diet fed mice (Fig. 8d, e), showing a more robust day-to-night difference the obese mice treated with NAD+ at ZT11 (Fig. 8e). Furthermore, we found similar observations when applying a therapy with the NAD+ precursor nicotinamide (NAM), which was previously described to boost hepatic NAD+ after IP injection in one hour[101]. Hence, three weeks with NAM chronotherapy performed best when applied at ZT11 to improve body weight, GTT and ITT (Supplementary Fig. S9a–d). As shown for NAD+, the NAM treatment at ZT23 inverted the expression of the hepatic molecular clock (Supplementary Fig. S9e), while keeping behavioral locomotor activity in phase with light/dark cycles (Supplementary Fig. S9f). This reinforces the notion that NAD+ can potentially synchronize the hepatic molecular clock, by reorganizing clock gene expression to adjust its phase to the time of the day when NAD+ bioavailability is higher. Collectively, these data support that boosting NAD+ levels is an effective treatment for HFD-induced metabolic disease, and demonstrate that a chronotherapeutic approach is significantly more beneficial when NAD+ increases at the onset of the active phase.

## Discussion

In the past decade, therapies oriented to increase endogenous NAD+ levels have received much attention as treatments for metabolic disorders. Mounting research in rodents demonstrate that pharmacological approaches using "NAD+ boosters" treat the physiopathology of diet and age-associated diabetes in mice, and reverse cardiovascular disease or muscle degeneration[85]. In humans, the NAD+ precursor

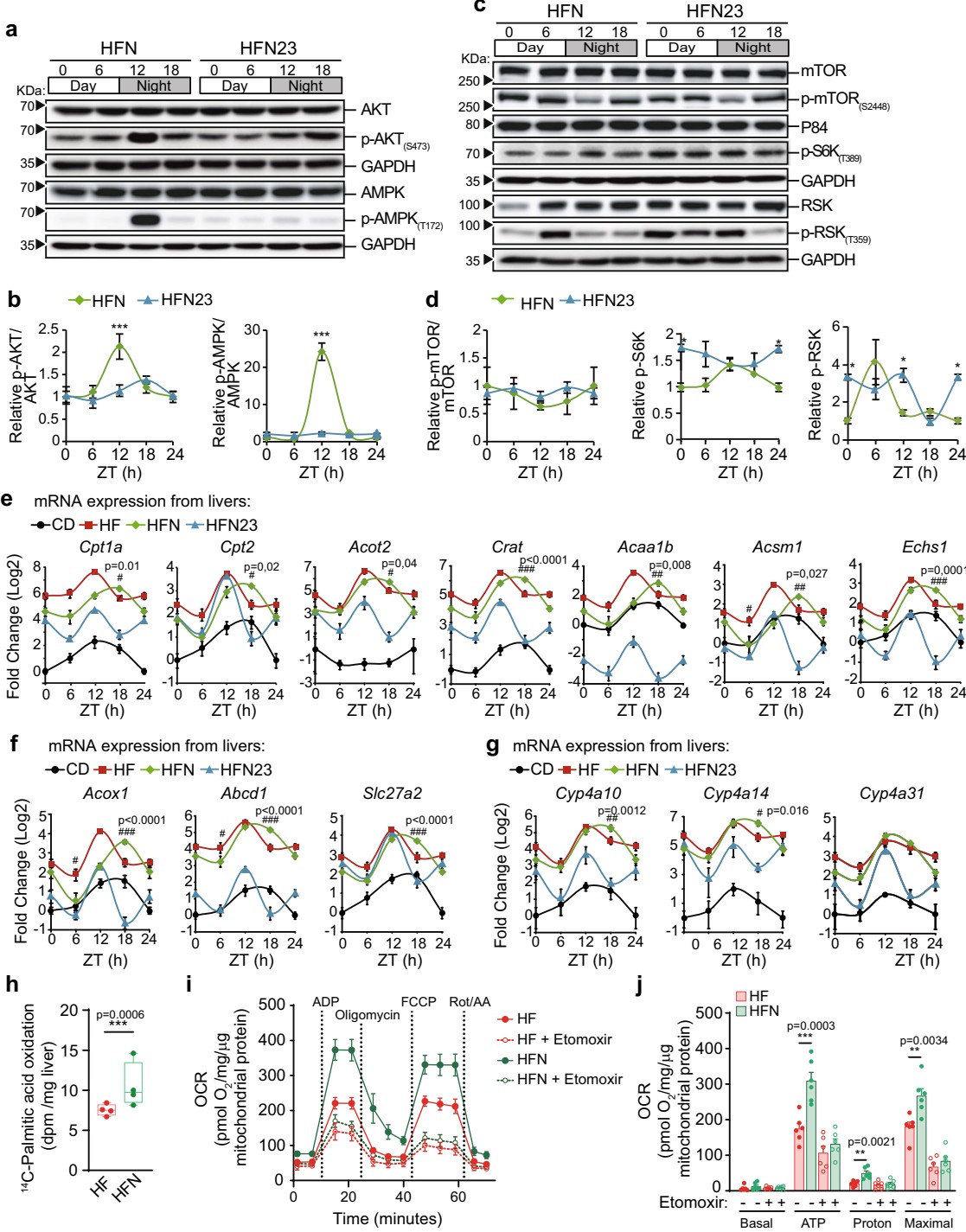

**Fig. 6 | Distinct impact of NAD+ treatment at ZT11 versus ZT23 on hepatic nutrient and insulin sensing pathways. a** Expression of AKT, p-AKT(S473), AMPK and p-AMPK(T172) along the day in the liver from mice treated with NAD+ either at ZT11 (HFN) or at ZT23 (HFN23) determined by western blot. Uncropped blots in Source Data. **b** Quantification of western blots from $n = 5$ biological samples, normalized to GAPDH loading control. Data from CD at ZT0 was set to 1. **c** Expression of proteins and phosphor-proteins in the mTOR pathway along the day in the liver from mice treated with NAD+ assessed by western blot. **d** Quantification of western blots from $n = 5$ biological samples. Measurements were normalized to p84 or GAPDH loading controls. Data from CD at ZT0 was set to 1. **e–g** RT-qPCR determined expression of rate-limiting and regulatory enzymes involved in mitochondrial (**e**) or peroxisomal (**f**) β-oxidation, and (**g**) ω-oxidation ($n = 5$ mice for CD and HF, 6 for HFN and 5 for HFN23 except at ZT6, where $n = 6$). **h** $^{14}$C-palmitic acid oxidation was assessed by quantifying $^{14}CO_2$ release ex vivo from livers of HF and

HFN mice ($n = 4$ mice per group with three independent measurements. Two experimental replicates were performed with comparable results). Two-way ANOVA **i** Oxygen consumption rate (OCR) was determined using Seahorse XF analyzer to assess CPT-1-dependent mitochondrial respiration from $n = 6$ mice with seven technical replicates each. **j** Mitochondrial bioenergetic parameters were calculated from extracellular flux analyses. One-way ANOVA with Siddak´s posttest. CD control diet-fed mice, HF high-fat diet-fed mice, HFN high-fat diet fed, NAD+ treated mice at ZT11, HFN23 high-fat diet fed, NAD+ treated mice at ZT23. Data points at ZT24 are duplicates from ZT0, replotted to show 24-h trends. The data represent means ± SE. *$p < 0.05$, **$p < 0.01$, ***$p < 0.001$, Two-way ANOVA followed by Bonferroni's (**b**, **d**) or Tukey's (**e–g**) posttest. Statistical details and exact $p$ values are provided in Supplementary Data 1.Symbol key for multiple comparisons: *HFN vs HFN23; #HF vs HFN. See also Supplementary Fig. S6.

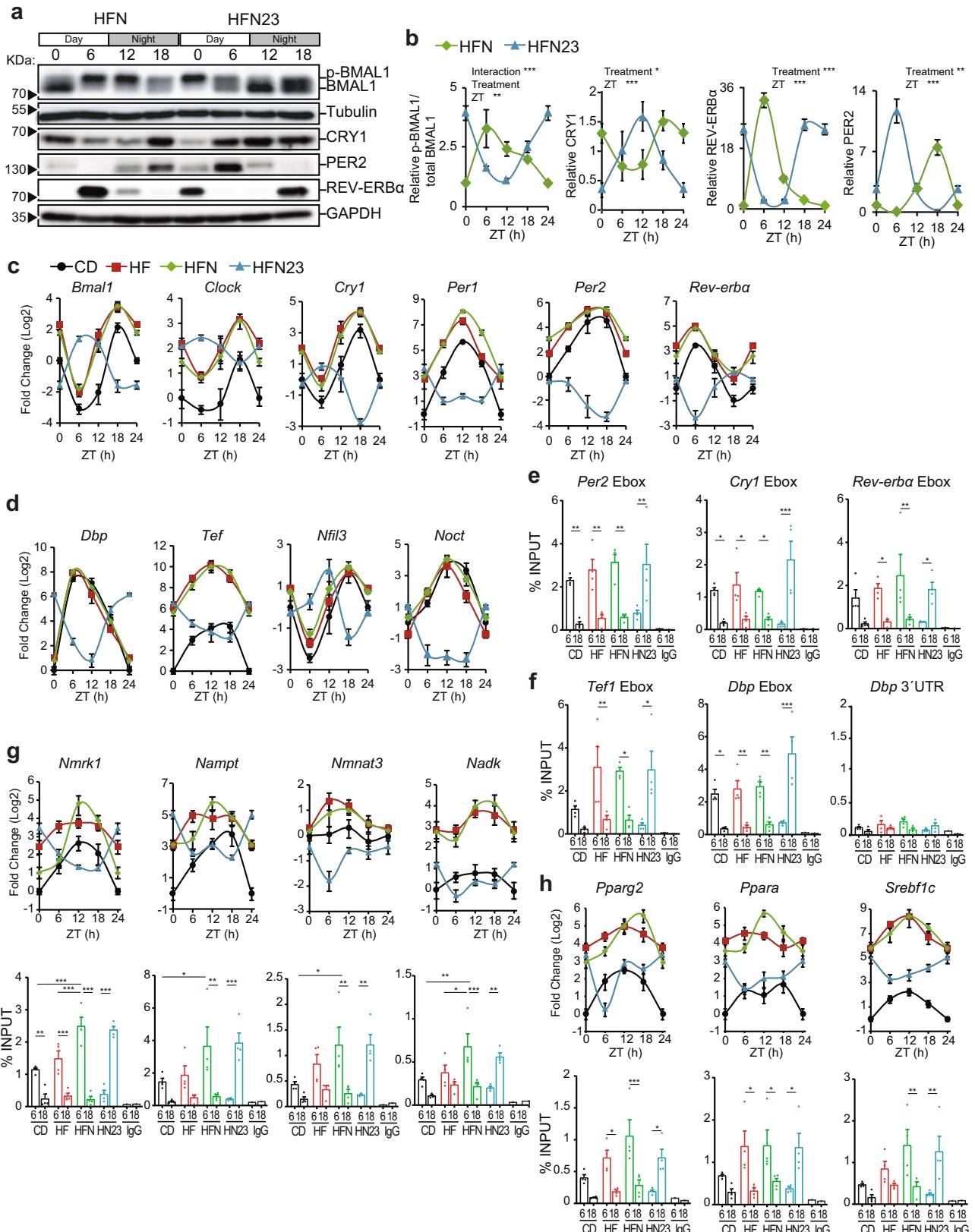

niacin has been largely used to treat dyslipidemia, and a number of clinical trials are ongoing for other NAD⁺ precursors[102]. However, all these studies and clinical protocols mostly disregard the reciprocal interactions between circadian rhythms and NAD⁺ metabolism. Here, we demonstrate that NAD⁺ can shift the phase of the hepatic molecular clock while preserving the SCN clock largely intact, and concomitantly,

the efficacy of increasing NAD⁺ levels to correct metabolic diseases depends on the time-of-day (Fig. 8f).

NAD⁺ and its phosphorylated and reduced forms, NADP⁺, NADH and NADPH, are fundamental compounds in intermediary metabolism as hydride-accepting or -donating coenzymes in redox reactions[103]. NAD⁺ is produced in all tissues from the salvageable metabolite NAM, or

**Fig. 7 | NAD$^+$ synchronizes the hepatic circadian clock. a** Circadian clock protein expression from liver whole cell extracts of obese mice treated with NAD$^+$ at ZT11 (HFN) or ZT23 (HFN23). Tubulin or GAPDH were used as loading control. Uncropped blots in Source Data. **b** Quantification of western blots from $n = 3$ mice, except for BMAL1 where n = 4 for HFN and 5 for HFN23. Measurements were normalized to the loading control, and data from CD at ZT0 was set to 1 **c** RT-qPCR determined circadian clock gene expression in the liver ($n = 5$ mice for CD and HF, 6 for HFN and 5 for HFN23 except at ZT6, where $n = 6$). **d** RT-qPCR determined rhythmic expression of clock-controlled genes in the liver ($n$ as in **c**). **e, f** Chromatin immunoprecipitation (ChIP qPCR) was performed in the liver from mice at ZT6 or ZT18, using anti-BMAL1 antibodies. ($n = 4$ biological and two technical replicates). **g, h** RT-qPCR determined rhythmic expression of genes related to NAD$^+$ metabolism (**g**) and genes regulating lipid metabolism (**h**) in the liver ($n$ as in **c**). BMAL1 ChIPs at ZT6 or ZT18 were analyzed by performing qPCR on BMAL1 binding sites at selected regulatory elements of these genes. CD Control diet-fed mice; HF High-fat diet-fed mice; HFN High-fat diet-fed, NAD$^+$ treated mice at ZT11; HFN23 High-fat diet fed, NAD$^+$ treated mice at ZT23. **b–d** Data points at ZT24 are duplicates from ZT0, replotted to show 24-h trends. The data represent means ± SE. *$p < 0.05$, **$p < 0.01$, ***$p < 0.001$, Two-way ANOVA followed by Sidak's posttest. $p$ values are provided in Supplementary Data 1. See also Supplementary Figs. S7, S8.

from precursors including nicotinamide riboside (NR), nicotinic acid (NA) or nicotinamide mononucleotide (β-NMN), while some tissues such a liver produce NAD$^+$ de novo from tryptophan, in a much less efficient biosynthetic pathway[30,104]. At this regard, the NAD$^+$ precursors NAM, NMN and NR have been preferentially used as NAD$^+$ boosters; however, we set up a therapy with NAD$^+$ because the limited data tracing metabolic fluxes suggest distinct, tissue-specific effects of NR and NMN[105]. Moreover, NAD$^+$ uptake appears fast and effective in cells, and a mitochondrial active transporter has been recently described[106–109]. Yet, to gain insights into the bioavailability of NAD$^+$ precursors in our study, it would be necessary to unravel the hepatic NAD$^+$ metabolome in all tested conditions, as for example, the possibility that time-dependent decline in NADPH and NADP$^+$ levels in livers from obese mice[27,35,110] contributes to differences between HFN and HFN23 mice cannot be ruled out, constituting a limitation of our study. However, we demonstrated that hepatic NAD$^+$ levels raised within an hour after IP injection in obese mice, and followed a circadian turnover when administered at ZT11, at the onset of the active phase (Fig. 1b). This chronotherapy recapitulated the metabolic improvements to a similar extent to the previously reported for the NAD$^+$ precursors NMN[36,111,112] and NR[32,35,40,43,44,113–115], mostly consisting of decreased weight gain, improved insulin sensitivity and glucose tolerance, decreased circulating leptin and triglycerides, and amelioration of NAFLD with decreased hepatic pro-inflammatory transcriptional signature (Figs. 1–3). At the molecular level, we demonstrated that, upon NAD$^+$ chronotherapy, daily rhythms were restored for hepatic insulin and nutrient signaling. This was evidenced by rhythms in AMPK-T172 and AKT-S473 phosphorylation, and mTORC1-directed pS6K phosphorylation, which became oscillatory with peaks during the active phase (ZT12-18; Fig. 4f, g). Accordingly, we also observed decreased phosphorylation of p90RSK-T359 (Fig. 4g), a positive effector of mTORC1 signaling and driver of NFκB activity[116,117]. It appears conflicting that the AMPK response to starvation and the mTORC1 nutrient sensing pathways became active at concurrent times during the day after restoring NAD$^+$ oscillations in obese mice, as they usually signal opposed nutritional states and engage into regulatory negative feedback loops[118]. However, recent research shows that specific activation of AMPK exists which does not lead to mTORC1 inhibition, but instead sustains ULK1 activity and autophagy to preserve protein homeostasis[119], which is in keeping with our findings (Fig. 4f, g, Supplementary Fig. S4c). Notably, a hepatic NAD$^+$-specific transcriptional signature emerged in treated mice related to intracellular trafficking, consisting of overexpression of the Rab GTPase network regulator of autophagy[70], further reinforcing the notion that NAD$^+$ preferably targets AMPK signaling to activate autophagy and possibly, translocation of membrane receptors. Along these lines, AMPK has been largely recognized as a therapeutic target for metabolic diseases[120,121], yet the well-known circadian fluctuations in its activity[61] have been fully overlooked for treatment.

We have demonstrated time-of-day dependent and independent responses to NAD$^+$ therapy. Clearly, as previously reported[122,123], rising NAD$^+$ levels elicit positive responses including correction of hepatic steatosis and the inflammatory environment in obese mice, and we found that these positive effects occur independently of time of NAD$^+$ supply (Fig. 5i–m). In fact, these two processes are interconnected,

since during obesity, inflammation in the liver happens and increased cytokines lead to overexpression of genes involved in de novo lipogenesis and ceramide biosynthesis[124,125]. However, we also observed different responses to NAD$^+$ between obese mice treated at ZT11 or at ZT23, where the latter did not fully recapitulate certain metabolic parameters, such as improvement in glucose tolerance, insulin sensitivity or circulating triglycerides. Concomitantly, NAD$^+$ therapy at ZT23 did not trigger AMPK phosphorylation neither corrected mTORC1 signaling in the liver of obese mice. Strikingly, the expression dynamics of the molecular clock were completely phase inverted in livers from HFN23 and HFNAM_23 mice, showing that at the onset of the active phase, NAD$^+$ can efficiently invert the phase of the hepatic clock (Figs. 7, 8, Supplementary Figs. S7–S9). These findings support earlier evidence that specific nutritional cues are potent zeitgebers for peripheral oscillators[16,47,97,126], and reinforce the existing notion of autonomous regulation of hepatic NAD$^+$ metabolism closely linked to the clock function[26]. Together with our findings, this suggests that the molecular clock acts as a key interface to induce timing-specific modulation of nutrient and insulin signaling by NAD$^+$.

Our analyses revealed substantial differences in expression from genes involved in fatty acid oxidation, with marked downregulation in obese mice treated with NAD$^+$ at ZT23 (Fig. 6e–g, Supplementary Fig. S6c). In mouse liver, these genes are oscillatory with a peak of expression at the end of the rest phase[86]. Their expression is to some extent clock-controlled; however, their transcriptional regulation mostly relies on nutritional cues integrated by intracellular signaling, multiple nuclear receptors and transcription factors such as PPARγ, PPARα or SREBP1, epigenetic regulators including MLL1 or SIRT1, and even neural circuits[16,86,127–129]. Untimed NAD$^+$ rise, through reorganizing the circadian machinery and the subsequent misalignment from feeding rhythms, might hinder the coordinated action between the clock and cooperative transcriptional regulators on chromatin, hereby obstructing the adequate control of specific transcriptional programs. At this regard, BMAL1 recruitment to chromatin was adjusted by timed NAD$^+$ treatment, and when administered at ZT23 leaded to phase-inverted transcription of direct CLOCK:BMAL1 targets, as expected for a pioneer-like transcription factor[130,131]. In this scenario, we found that several master regulators of rhythmic hepatic lipid and cholesterol metabolism including *Ppar*α, *Ppar*γ, *Srebp1c*, *Cebpa*, or *Hnf4a*[16,86,132,133] were subjected to this mechanism, and their phase inversion in HFN23 mice was accompanied by inverted rhythms in hepatic cholesterol and circulating triglycerides (Fig. 7, Supplementary Fig. S8). These results demonstrate that NAD$^+$ modulates BMAL1 recruitment to chromatin and shapes rhythmic transcription and metabolism.

NAD$^+$ is a coenzyme in redox reactions, but also serves as a substrate of NAD$^+$ consuming enzymes which cleave NAD$^+$ to produce NAM and an ADP-ribosyl product, such as ADP-ribose transferases, cADP-ribose synthases and sirtuins (SIRT1-SIRT7)[103,134,135]. Indeed, both NAD$^+$ consumers SIRT1[46,136,137] and SIRT3[21] provide reciprocal regulation to the clock machinery to modulate circadian transcription and metabolism in the liver. Furthermore, recent research shows that a NAD$^+$-SIRT1 interplay mediates deacetylation and nuclear translocation of PER2 and, in line with our results, shapes BMAL1 function, while this control is altered in livers from aged mice[46]. Through activation of SIRT1 and

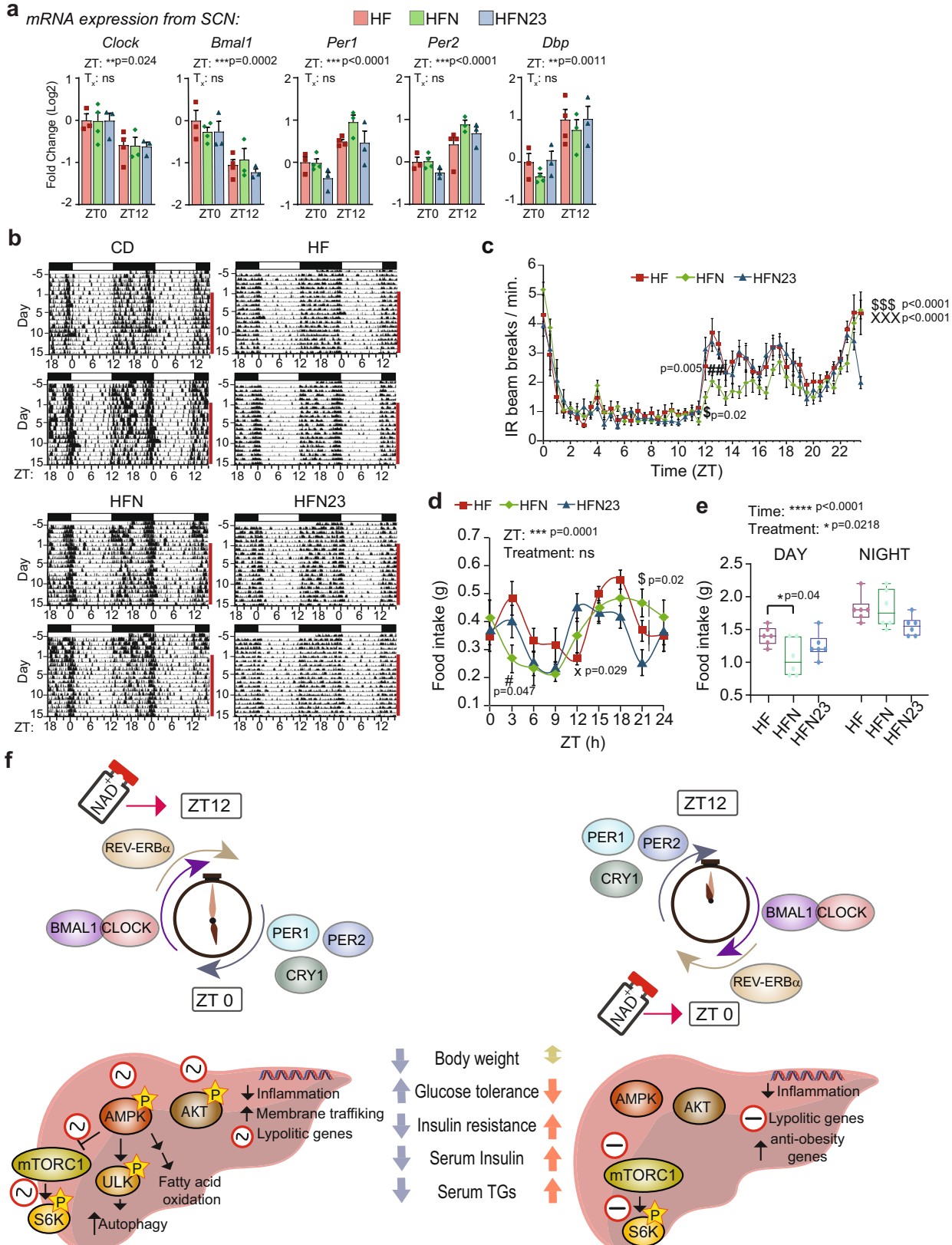

SIRT3, it is also possible that rising NAD⁺ at -ZT12 might contribute to rhythmic lipid oxidation and mitochondrial function driven by protein acetylation, including PPARγ[27,138], while keeping the hepatic clock aligned to the external time. Yet, the regulation of the circadian system by sirtuins in health and disease remains to be fully disentangled. Circadian misalignment imposed by antiphase NAD⁺ in our HFN23 and

HFNAM_23 mice might obstruct metabolic improvements, through uncoupling of the central light-synchronized and peripheral NAD⁺-synchronized clocks. Although hepatic neutral lipid content was reduced independently of time-of-treatment (Fig. 5i–k), significant improvement of glucose homeostasis and hepatic insulin signaling were apparent only in HFN mice. Indeed, circadian misalignment has

**Fig. 8 | Time-of-day dictates efficiency of NAD⁺ treatment of diet-induced metabolic disease through synchronizing the hepatic circadian clock and adjusting coordination between intracellular signaling and gene expression.** **a** RT-qPCR from clock genes in the SCN at ZT0 and ZT12 (*n* = 3 biological replicates and 2 technical replicates). **b** Representative double-plotted actograms of locomotion measured using infrared sensors in a 12-h light/12-h dark cycle. **c** Average 24-h activity profile from the indicated groups of mice. Average was calculated for five days after NAD⁺ treatment. *n* = 7 mice for HF and HFN, and 6 for HFN23. **d** Food intake was measured over 24 h **e** Average food intake during light phase (day) and dark phase (night). For the boxplots, the top and bottom lines of each box represent the 75th and 25th percentiles, respectively. The line inside each box represents the median. The lines above and below the boxes are the whiskers **f** At ZT12, NAD⁺ sustains the alignment of the hepatic molecular clock while reinforcing circadian oscillation in the activity of nutritional sensors such as AMPK, AKT or mTOR.

Transcriptional responses are adjusted accordingly to suppress inflammation probably through inhibition of NF-κB transcription factor, and to increase amplitude in lipolytic gene expression and fatty acid oxidation, with a peak during the active phase. Membrane trafficking and ULK activity are indicative of active autophagy as a specific response to NAD⁺ treatment. At ZT0, increased NAD⁺ in the liver inverts the phase of the molecular clock, imposing antiphase rhythms in clock gene and protein expression. This in turn inverts the phase of clock-controlled genes and uncouples transcriptional responses. CD control diet fed mice, HF high-fat diet-fed mice, HFN high-fat diet fed, NAD⁺ treated mice at ZT11, HFN23 high-fat diet fed, NAD⁺ treated mice at ZT23. The data represent means ± SE. *$p < 0.05$, **$p < 0.01$, ***$p < 0.001$, Two-way RM ANOVA followed by Sidak's posttest. Symbol key for multiple comparisons: #HF vs HFN, X HF vs HFN23, $HFN vs HFN23. See also Supplementary Fig. S8.

been extensively reported to drive metabolic dysfunction both in mouse and humans[139–141]. In this scenario, expression of clock genes in the SCN at two circadian times appeared largely intact upon NAD⁺ injection, and concomitantly, locomotor activity remained aligned to the light-dark cycles also in HFN23 mice (Fig. 8a–c). Although further analyses with higher resolution are essential to fully disentangle the extent of NAD⁺ influence on cyclic gene expression in the SCN, uncoupled liver and SCN clocks have been previously reported in mice when access to food is restricted to the light period[97,142,143]. However, our HFN23 mice did not show significant variations in eating behavior (Fig. 8d–e), evidencing that uncoupling the central and hepatic clocks is a time-dependent effect of NAD⁺ supply. Notably, abnormal metabolic signaling triggered by high-fat diets uncouples body clocks[15]; thus, it would be interesting to define which extra-hepatic oscillators are influenced by NAD⁺. At this regard, recent reports suggest that the brain blood barrier might be permeable to NAD⁺[144,145] in which case hypothalamic neurons could be influenced. Yet, further research is necessary to decipher the extent of the modulation of brain clocks by increased circulating NAD⁺ precursors. Additionally, our study is limited by the cellular heterogeneity in fatty liver, with for example, infiltration of proinflammatory macrophages which have been recently shown to limit NAD⁺ bioavailability through high expression of the NAD-consuming enzyme CD38[146,147]. Hence, it is possible that time-dependent cellular heterogeneity in liver[148] could contribute to the NAD⁺-dependent improvement of the metabolic phenotype.

In humans, clinical trials aiming to boost endogenous NAD⁺ for treatment of metabolic diseases are increasing, showing promising results; for example, in postmenopausal women with prediabetes, a daily dose of NMN increases muscle insulin sensitivity[45,134,149]. However, all these studies mostly overlook the time of drug intake, which is selected based on practicalities or attempting to displace side effects from the patient's active phase. Considering our results, we propose that time of treatment dictates the amplitude of metabolic benefits from rising NAD⁺ levels, which ideally outlines the basic strategy of chronobiology-based NAD⁺ therapy.

## Methods
### Animals and diets
Four-week-old C57Bl/6 J mice were obtained from the Biological Models Unit at the Instituto de Investigaciones Biomédicas (UNAM, Mexico). The mice were kept under a 12:12-h light:dark cycles. Food and water were provided ad libitum. Temperature and humidity were constantly monitored. Mice were randomly distributed to four groups. The control group was fed during 11 weeks with normal chow (CD, 2018S Teklad, ENVIGO), bearing 24% calories from protein, 18% calories from fat, and 58% calories from carbohydrates. The other three experimental groups were fed a high-fat diet (HFD, based on TD.160547 Teklad, ENVIGO), consisting of 15% calories from protein, 53% calories from fat and 38% calories from carbohydrates, and customized to match NAD⁺ dietary sources content to that of the CD

(0.2% tryptophan and 115 mg/kg nicotinic acid). Food intake and body weight were measured once a week. For daily food intake measurements, mice were single housed, and measurements were recorded for 1 week. Female mice were used for the experiments using NAM as a NAD⁺ precursor, and the rest of the experiments were performed in male mice.

All animal experimental procedures were reviewed and approved by the Internal Committee for the Care and Use of Laboratory Animals (CICUAL) at the Instituto de Investigaciones Biomédicas, (UNAM, Mexico), and are registered under protocol no. ID240.

### Chronotherapy with NAD⁺ and NAM
NAD⁺ and NAM were purchased from SIGMA (cat. no. N7004, N0636) and were dissolved in 0.9% NaCl isotonic saline solution and filter sterilized. To determine the NAD⁺ dose, we wanted to keep two premises: (1) to keep NAD⁺ levels into the physiological range, and (2) avoid undesirable secondary effects of high doses. To do so, we chose the range of tested doses based on previous reports[100,150,151], and treated mice with IP injection of 800, 100, 50, or 10 mg/kg body weight, while keeping a constant volume of approximately 180 μl. Control mice were injected with isotonic saline solution. C57Bl/6 J male mice (*n* = 3) were IP injected, and sacrificed one hour later. NAD⁺ was measured by HPLC as described below. Because we planned on a chronic treatment, the minimum dose inducing a statistically significant increase in hepatic NAD⁺ with respect to control livers was selected as the experimental dose (Supplementary Fig. S1a, 50 mg/Kg of body weight: $P < 0.0001$, One-way ANOVA with Tukey's posttest). Hence, for all experiments, mice were IP injected with 50 mg/kg of NAD⁺ for 20 consecutive days, either at ZT11 (one hour before lights off), or at ZT23 (one hour before lights on). Of note, we didn´t find differences in hepatic NAD⁺ at a dose of 10 mg/kg, a reason why we did not try lower concentrations. The dose for NAM treatment in female mice (200 mg/kg) was selected based on previous reports[101,152–154].

### Detection and quantification of NAD⁺ by HPLC
NAD⁺ measurements were performed according to Yoshino and Imai 2013[155], with subtle modifications. 100 mg of frozen tissue were processed in a final volume of 2 ml of 10% HClO₄ with a Polytron homogenizer (Kinematica CH-6010 Kiriens-Lu) and centrifuged at 12,000 × *g* for 5 min at 4 °C. The supernatant was neutralized adding a one-third volume of 3 M K₂CO₃, and vortexed. After 10 min of incubation on ice, samples were cleared by a 12,000 g centrifugation at 4 °C during 5 min. The supernatant was diluted at 30% with 50 mM phosphate buffer (3.85% of 0.5 M KH₂PO₄, 6.15% of 0.5 M K₂HPO₄, 90% HPLC grade water -v/v/v-, pH 7.0, filtered through a 0.22 μm filter and degassed). 50 μl of the samples were analyzed using a 1260 infinity quaternary LC VL HPLC system (Agilent) attached to a diode array detector. Analytes were separated on a ZORBAX Eclipse XDB-C18 4.6 × 150 mm, 5 μm column (Agilent p/n 993967-902). For the HPLC, the gradient mobile phase was delivered at a flow rate of 1 ml/min, and consisted of two solvents:

(A) 50 mM phosphate buffer pH 6.8 and (B) methanol 100%. The initial concentration of A was 100%, the solution was held into the column for 5 min and then B was progressively increased to 5% over 1 min, held at 5% B for 5 min, followed by an increase to 15% B over 2 min, held at 15% B for 10 min and returned to starting conditions of 100% A in 1 min, and held at 100% A for 6 min. $NAD^+$ was detected using a sample wavelength of 261 nm and reference wavelength of 360 nm. Adequate standards including $NAD^+$ were used for calibration and identification of the retention/migration time of $NAD^+$ within the samples. Instrument control, data acquisition, and analysis were performed using the Agilent ChemStation system for LC, according to manufacturer's instructions. $NAD^+$ levels were quantitated based on the peak area in the chromatograms compared to a standard curve and normalized to tissue weight.

### Glucose tolerance test (GTT) and insulin tolerance test (ITT)

At 8, 10, and 11 weeks of experimental paradigms, mice were subjected to either 12 h or 5 h of fasting, followed by a glucose tolerance test (GTT) or an insulin tolerance test (ITT) respectively. For the GTT, IP injection of D-glucose (SIGMA cat no. G7021) at 2 mg/kg was used, while for ITT, human insulin (Eli Lilly cat. HI0210) at 0.6 U/kg was IP injected. Circulating glucose was measured from a tail-tip blood drop, using an ACCU CHEK active glucometer (ROCHE) at time points 0 (before injection) and 15, 30, 60, and 120 min after IP injection of either glucose (GTT) or insulin (ITT). Experiments were performed per triplicate, using 5–6 mice per experiment.

### Metabolites and hormone analyses

Blood serum was collected postmortem by cardiac puncture. Triglycerides (TG) in serum and liver were measured using the Triglyceride Colorimetric Assay Kit (Cayman Chemical, cat. no. 10010303). Free fatty acid content was determined with the Free Fatty Acid Fluorometric Assay Kit (Cayman Chemical, cat. no. 700310). Serum insulin and leptin levels were measured by ELISA, using the Ultra-Sensitive Mouse Insulin ELISA Kit (Crystal Chem Inc, cat. no. 90080) and the Mouse Leptin ELISA Kit (Crystal Chem Inc, cat. no. 90030) according to the manufacturer's instructions. Hepatic cholesterol was determined using a Cholesterol Quantitation Kit (Sigma-Aldrich cat. no. MAK043, colorimetric). Absorbance/fluorescence was measured using a Synergy H1 microplate reader (BioTek).

### Temperature measurements

Rectal temperature in mice ($n = 10$ mice, and three technical replicates) was registered using a portable digital thermometer (BIOSEB) every 3 h throughout 24 h. For the acquisition of infrared thermography, mice were placed inside an acrylic box in darkness. Thermal images were acquired at ZT12 using an Inframetrics C2 Thermal Imaging System Compact Pocket-Size camera (FLIR Systems) with a frequency of 9 Hz, thermal sensitivity <0.10 °C, resolution $80 \times 60$ (4800 pixels) and temperature range of 14 to 302 ° F. ($n = 4$, with three technical replicates). Images processing was performed using FLIR-Tools software v 5.13.17214 (2015 FLIR® Systems).

### Oil-Red-O staining

Frozen OCT-embedded liver tissues were cut into 10-µm sections using a Leica cryostat and air dried for 10 min at room temperature. Slides were briefly washed with PBS and fixed for 2 min with 4% fresh paraformaldehyde. Preparation of Oil Red O (SIGMA, cat. no.O1391) working solution and staining of slides was performed according to Mehlem et al.[156]. Oil Red O working solution (3.75 mg/ml) was applied on OCT-embedded liver sections for 5 min at RT. Slides were washed twice during 10 min. in water, and mounted in vectashield mounting media (Vector Labs, cat. no. H-1000). The images were captured with the Olympus camera DP70 system using the DPController v 1.1.1.65 software, coupled to a Olympus BX51 microscope with the

DPManager software v. 1.1.1.71, using a ×40 magnification. The background was corrected by white balance and was selected as a blank area outside the section. For representative images, some sections were stained with Gil I haematoxylin. Surface of lipid droplets was quantified using the ImageJ software (v 1.53), by converting RGB to 8-bit grayscale images, and then using the "analyze particles" plug-in to measure the area and size of the lipid drops[157]. Three frames per biopsy were used for image analyses and quantification ($n = 3$ biological replicates with three technical replicates).

### SCN dissection

For gene expression analysis from the SCN, frozen brains were placed on ice, and the 1 $mm^3$ region above the optic chiasm was dissected out using microscissors. Tissues were placed in microcentrifuge tubes in 100 µl of Trizol and kept at −80 °C until use. Total RNA was subsequently extracted and resuspended in 12 µl of water.

### Total RNA extraction

20 mg of liver tissue or the dissected SCN, were homogenized (Benchmark Scientific, D1000 homogenizer) for 30 seconds with 0.5 ml of Trizol (TRIzol™ Reagent, Invitrogen, cat. no. 15596018). The homogenate was incubated for 5 min at RT, then 0.1 ml of chloroform was added, shaken, and incubated at RT for 3 min followed by a centrifugation during 15 min at $12,000 \times g$ at 4 °C. The upper phase was extracted, and 0.25 ml of isopropanol was added. After a 10 min incubation at RT, RNA was precipitated by centrifugation for 10 min at $12,000 \times g$ and 4 °C. The RNA was washed with 1 ml of 75% ethanol and resuspended in 20 µl of molecular biology grade water (Corning, cat. no. 46-000). 2 µl of the sample were used to quantify its concentration and assess its quality in a NanoDrop (Thermo Scientific).

### cDNA synthesis

It was performed using the kit iScript™ cDNA synthesis (Bio-Rad, cat. no 1708890). 500 ng of RNA were mixed with 2 µl of 5X iScript Reaction Mix and 0.5 µl of the enzyme iScript Reverse transcriptase in a volume of 10 µl. The thermal cycler (Axygen MaxyGeneTM II) was programmed as follows: Alignment for 5 min at 25 °C, reverse transcription for 20 min at 46 °C and inactivation for 1 min at 95 °C. The reaction was cooled to 4 °C and diluted to 5 ng/µl.

### Quantitative real-time polymerase chain reaction

The reactions were performed in a final volume of 10 µl, adding 5 µl of the Universal SYBR Green Super Mix reagent (Bio-Rad, cat. No. 1725121), 1 µl of 2.5 µM forward and reverse primers, and 7.5 ng of cDNA per reaction. The thermal cycler (Bio-Rad, CFX96 Touch Real-Time PCR Detection System) was set to the following program: 30 s at 95 °C followed by 40 cycles of 5 s at 95 °C and 30 s at 65 °C. Single-product amplification was verified by an integrated post-run melting curve analysis. Values were normalized to the housekeeping genes *B2m, Ppia,* and *Tbp.* The geometric mean was used to determine Ct values of the housekeeping genes and expression values for the genes of interest were calculated using ΔCT methodology. Primer sequences are available in Supplementary Data 5.

### mtDNA quantification by quantitative real-time PCR

10 mg of liver were used to extract DNA with the DNeasy Blood & Tissue Kit (QIAGEN, cat. no. 69506), according to the manufacturer's instructions. Quantitative PCR was performed using 7.5 ng of DNA and 2.5 µM of S18 and mtCOX1 primers as described for cDNA quantification, with a program of 20 min at 95 °C, followed by 50 to 55 cycles of 15 s at 95 °C, 20 s at 58 °C and 20 s at 72 °C. Single-product amplification was verified by an integrated post-run melting curve analysis. 5–6 mice were analyzed for each time point and condition, with two technical replicates. mtDNA content using the formula: $2 \times 2^{(\Delta CT)}$, where ΔCT is the difference of CT values between S18 gene and mtCOX1 gene[158].

## Transcriptional profiling from mouse livers

Liver RNA samples for microarray analysis were prepared using our previously described procedures, with slight modifications. Briefly, total RNA was first extracted with TRIzol Reagent (Invitrogen), then cleaned with RNeasy Mini purification Kit (QIAGEN cat. no. 74106) according to the manufacturer's RNA CleanUp protocol. RIN values (≥7.0) were validated with an Agilent Bioanalyzer 2100. 900 ng of total RNA per sample was used as a template to obtain cDNA with the GeneChip cDNA synthesis Kit (Affymetrix, Santa Clara, CA). Microarray experiments were conducted by the Microarray Unit at the National Institute of Genomic Medicine (INMEGEN, Mexico City) using the mouse Clariom™ D Assay (Applied Biosystems™), as per manufacturer's instructions. Microarray experiments were performed in triplicate ($n$ = 3 biological replicates). The Clariom™ D Array consists of 66100 genes (transcript clusters), 214900 transcripts, 498500 exons and 282500 exon-exon splice junctions from *Mus musculus*. Sequences are mapped to the National Center for Biotechnology Information (NCBI) UniGene database. The arrays were scanned in the GeneChip Scanner 3000 7 G (Affymetrix) and the GeneChip Command Console Software (v 4.0.3) was used to obtain the.CEL intensity files. Normalized gene expression data (.CHP files) were obtained with the Transcriptome Analysis Console (TAC v4.0.1.36) software using default parameters. Changes in gene expression (±1. fold-change; FDR-corrected $p$ value ≤0.05) were subjected to functional analyses using the "Compute Overlaps" tool to explore overlap with the CP (Canonical Pathways) and the GO:BP (GO biological process) gene sets at the MSigDB (molecular signature database) v7.0. The tool is available at: https://www.gsea-msigdb.org/gsea/msigdb/annotate.jsp, and estimates statistical significance by calculating the FDR q-value. This is the FDR analog of the hypergeometric P value after correction for multiple hypothesis testing according to Benjamini and Hochberg. Gene set enrichment analysis (GSEA) was performed using GSEA v. 4.0.3.[54] to determine the enrichment score within the Hallmark gene set collection in MSigDB v7.0[159], selecting the Signal2Noise as the metric for ranking genes. The findMotifs.pl program in the HOMER software v 2.0[160] was used for motif discovery and enrichment, searching within the genomic regions encompassing 300 Kb upstream and 50 Kb downstream the TSS, and selecting 6–8 bp for motif length. Motif enrichment is calculated by findMotifs.pl using the cumulative hypergeometric distribution.

All raw and processed data can be accessed at the GEO database, number: GSE163865.

## Protein carbonyl (PCO) content

The determination of the carbonyl content was performed from total hepatic protein extracts (0.5 mg/ml), following a previously published protocol[161]. PCO present in the samples were derivatized by reaction with a working solution of 2,4-dinitrophenylhydrazine (DNPH 10 mM diluted in 0.5 M $H_3PO_4$; SIGMA) for 10 min at RT. The reaction was stopped by adding a NaOH (6 M) for 10 min. The absorbance of the samples was read in a spectrophotometer (Jenway, 6305) at 370 nm and the mean absorbance of control tubes (RIPA buffer) was then subtracted. To calculate the PCO concentration expressed as nmol PCO/mg protein, we used the following equation:

$$\text{PCO concentration} = \frac{10^6 \times \left(\frac{\text{Abs}_{366nm}}{22000 \text{M}^{-1*}\text{cm}^{-1}}\right)}{[\text{protein}]_{mg/ml}}$$

## Western Blot

Livers were lysed in 1× RIPA buffer supplemented with a protease/phosphatase inhibitor cocktail (cOmplete mini ROCHE 1:25 v/v, PMSF 1 mM, $Na_3VO_4$ 1 mM, NaF 0.5 mM). Total protein was quantified with Bradford reagent (SIGMA, cat. no. B6916), and 25 µg of extract were suspended 1:6 (v/v) in 6× Laemmli buffer (60 mM Tris HCl pH 6.8, 12% SDS, 47% glycerol, 0.03% bromophenol blue, 1 M DTT), separated on sodium dodecyl sulfate–polyacrylamide gel electrophoresis (SDS-PAGE), and transferred onto PVDF membranes (Merck-Millipore), using the Mini-PROTEAN electrophoretic system (Bio-Rad). Membranes were blocked using non-fat milk in PBST buffer for one hour and incubated with the corresponding primary antibody overnight at 4 °C. Membranes were washed three times with PBST and incubated with the secondary antibody for 5 hrs at RT. Antibodies used in this study were: From Cell Signaling: PPARγ (2443), AKT (9272), Phospho-AKT$_{Ser473}$ (9271), AMPKα (5831), Phospho-AMPKα$_{Thr172}$ (50081), mTOR (2893),), Phospho-mTOR$_{Ser2448}$ (5536), Phospho-p70 S6K$_{Thr389}$ (9234), Phospho-4E-BP1$_{Thr37/46}$ (2855), RSK1/RSK2/RSK3 (9355), Phospho-p90RSK$_{Ser359}$ (8753), REV-ERBα (13418), ULK1 (8054), Phospho-ULK1$_{Ser555}$ (5869), all diluted 1:1000; from Santa Cruz: C/EBPα (SC-365318, 1:500); from Abcam: BMAL1 (Ab3350, 1:1000); from Alpha Diagnostics International: PER2 (PER21-A 1:2000); from Bethyl Laboratories: CRY1 (A302-614A 1:1000); from Sigma: α-Tubulin (T5168, 1:80000); from Genetex: GAPDH-HRP (GTX627408-01, 1:120000) and P84 (GTX70220-01, 1:1000) The secondary antibodies were Anti-rabbit IgG (Cell Signaling, 7074, 1:150000 for BMAL1, 1:10000 for Pparγ and 1:80000 for the rest) or Anti-mouse IgG (Sigma I8765, 1:80000), conjugated to horseradish peroxidase. For detection, the Immobilon Western Chemiluminescent HRP Substrate (Millipore, cat. no. WBKLS0100) was used and luminescence was visualized and documented in a Gel Logic 1500 Imaging System (KODAK). Protein bands were quantified by densitometric analysis using Image Studio Lite Version 5.0 software (LI-COR biosciences). 3–5 biological replicates were used for each quantification. Uncropped, unprocessed scans of the blots are available in the Source Data file.

## Determination of inflammatory cytokines in mouse liver

The Mouse Inflammation ELISA Strip (Signosis, cat. No. EA-1051) was used for profiling inflammation cytokines, following the manufacturer's instructions. Absorbance at 450 nm was measured using a Synergy H1 microplate reader (BioTek).

## Fatty acid oxidation assay

Fatty acid oxidation was quantitated ex vivo following our previous protocols[48], with subtle modifications. Briefly, 20–60 mg of fresh liver from mice sacrificed at ~ZT16 were weighed, and samples were minced and homogenized in 300 µl homogenization buffer (DMEM, 1 mM pyruvate, 1% BSA free fatty acid, and 0.5 mM palmitate) at 4 °C. Then, 5 µl palmitic acid [14 C] 100 µC/ml (Perkin Elmer) was added, and these lysates were incubated for 2 hr at 37 °C. Eppendorf tubes were prepared to contain small pieces of Whatman paper in the cap of the tube, which were wet with 20 µl NaOH 3 M, while 150 µl 70% perchloric acid was placed inside the tube. Lysates were added to these tubes and incubated 1 h at 37 °C. The trapped $^{14}CO_2$ was determined transferring the filter discs to a scintillation vial with 4 ml of scintillation liquid and measuring in a Beckman LS6500 scintillation counter.

## Palmitoyl-CoA oxidation assay in isolated liver mitochondria

The Palmitoyl-CoA oxidation depends on the activity of Carnitine palmitoyltransferase-1 (CPT-1), the rate-controlling enzyme for long-chain fatty acid oxidation. Measurements of CPT-1-mediated mitochondrial respiration was determined in mitochondria freshly isolated from mouse liver using the Seahorse XFe96 Extracellular Flux Analyzer (Agilent Technologies), as previously described[162] with modifications as follows. Mice were sacrificed by cervical dislocation at ~ZT16, and ~100 mg of liver was dissected and placed on ice on mitochondrial isolation buffer (MIB1, 210 mM d-Mannitol, 70 mM sucrose 5 mM HEPES, 1 mM EGTA and 0.5% free fatty acid BSA) and centrifuged at 800 $g$ for 10 min at 4 °C. The tissue was homogenized using the Polytron tissue homogenizer at low potency for 8 seconds and centrifuged

at 800 $g$ (10 min at 4 °C). The supernatant was collected in 15 ml falcon tubes and centrifuged at 8000 × $g$ (10 min at 4 °C). The resulting pellet containing the mitochondria was washed three times with 1 ml of MIB1 and resuspended on 0.5 ml of mitochondrial assay solution (MAS1, 220 mM d-Mannitol, 70 mM sucrose, 10 mM KH$_2$PO$_4$, 5 mM MgCl$_2$, 2 mM HEPES, 1 mM EGTA, 0.2% BSA, pH 7.2) with the addition of substrates (40 μM palmitoyl-CoA, 0.5 mM malate, and 0.5 mM carnitine). Total protein was determined using the Qubit® 3.0 Fluorometer and 14 μg of isolated mitochondria were diluted in MAS1 buffer with substrates with or without the presence of the CPT-1 inhibitor, etomoxir (3 μM) and loaded per well in the XFe96 plate. The plate containing mitochondria was centrifuged at 2000 $g$ for 20 min at 4 °C. The oxygen consumption rate (OCR) was measured with seven technical replicates for each mouse, as the following compounds were injected to final concentrations per well: ADP (4 mM), oligomycin (2.5 μM), carbonyl cyanide 4-(trifluoromethoxy)phenylhydrazone known as FCCP (2.0 μM) and antimycin A (1 μM) / rotenone (1 μM). OCR was measured in the absence or presence of etomoxir, with sequential addition of ADP (ATP precursor), oligomycin (complex V inhibitor), FCCP (a protonophore), and Rotenone/antimycin A (Rot/AA; complex III inhibitor). Four mitochondrial respiration states were calculated: basal respiration (respiration of mitochondria with substrates but without ADP), ATP production or phosphorylating respiration (rate of ATP formation from ADP and inorganic phosphate), proton leak or non-phosphorylating respiration (rate of oxygen consumption while ATP synthase is inhibited with oligomycin), and Maximal respiration state after the addition of FCCP. The instrument control, data analysis and file management were performed with the Agilent Seahorse Wave v2.6 software.

## Chromatin immunoprecipitation (ChIP)

100–200 mg of liver tissue were homogenized with a pestle in PBS. Dual crosslinking was performed in a final volume of 1 ml using 2 mM of DSG (Disuccinimidyl glutarate, ProteoChem, CAS: 79642-50-5) for 10 min at RT on a rotary shaker. DSG was washed out and a second crosslink was performed using 1% formaldehyde (Sigma-Aldrich, F8775) in PBS for 15 min at RT on a rotary shaker. Crosslinking was stopped with 0.125 M glycine for 5 min at 4 °C. After two washes with ice-cold PBS, nuclei were isolated by resuspending the tissue in 600 μL of ice-cold nuclei preparation buffer (NPB: 10 mM HEPES, 10 mM KCl, 1.5 mM MgCl2, 250 mM sucrose, 0.1% IGEPAL CA-630) and incubating at 4 °C for 5 min in rotation. Nuclei were collected by centrifugation at 1,500 g for 12 min at 4 °C and, and resuspended in 600 μL of cold nuclear lysis buffer (10 mM Tris pH 8, 1 mM EDTA, 0.5 mM EGTA, 0.3% SDS, 1× cOmplete™ Protease Inhibitor Cocktail, Roche) for 30 min on ice. Nuclear lysates were stored at −80 °C. 300 μL of lysates were sonicated using a Bioruptor Pico Sonicator (Diagenode) for 15 cycles (30 s ON/30 s OFF). Chromatin fragments (100–500 bp) were evaluated on agarose gels using 10 μL of sonicated chromatin for DNA purification using the phenol method. 600 μL of ice-cold ChIP-dilution buffer (1% Triton X-100, 2 mM EDTA, 20 mM Tris pH 8, 150 mM NaCl, 1 mM PMSF, 1× cOmplete™ Protease Inhibitor Cocktail, Roche) was added to the fragmented chromatin, and 10% volume was recovered as the Input. Immunoprecipitation was set up overnight at 4 °C, by adding 20 μL of magnetic beads (Magna ChIP Protein G Magnetic Beads C #16-662, Sigma-Aldrich) and a combination of two anti-BMAL1 antibodies: 1.25 μL rabbit anti-BMAL1 (ab3350, Abcam) and 2.5 μL rabbit anti-BMAL1 (ab93806, Abcam) in 900 μL final volume. Immunoprecipitations with 4 μL of normal mouse IgG (Sigma-Aldrich, Cat. No. 18765) were performed simultaneously in 900 μL final volume. Sequential washes of the magnetic beads were performed for 10 min at 4 °C, as follows: Wash buffer 1 (20 mM Tris pH 8, 0.1% SDS, 1% Triton X-100, 150 mM NaCl, 2 mM EDTA), Wash buffer 2 (20 mM Tris pH 8, 0.1% SDS, 1% Triton X-100, 500 mM NaCl, 2 mM EDTA), Wash buffer 3 (10 mM Tris pH 8, 250 mM LiCl, 1% IGEPAL CA-630, 1% sodium deoxycholate)

and TE buffer (10 mM Tris pH 8, 1 mM EDTA). Chromatin was eluted by adding 400 μL of fresh elution buffer (10 mM Tris pH 8, 0.5% SDS, 300 mM NaCl, 5 mM EDTA, 0.05 mg/mL proteinase K) to the magnetic beads and incubating overnight at 65 °C. A treatment with RNase A at 0.1 mg/ml for 30 min at 37 °C was performed. The DNA was purified from the IPs and Inputs by adding one volume of phenol: chloroform:isoamamyl alcohol (25:24:1). After mixing and centrifugation, the aqueous phase was recovered, and DNA was precipitated by adding 1/10 volumes of sodium acetate (0.3 M pH 5.2), 20 μg of glycogen (10901393001, Roche) and 2 volumes of ice-cold ethanol, at −80 °C overnight. DNA was pelleted by centrifugation at 12,000 × $g$ for 30 min at 4 °C. The DNA was washed with 70% ethanol, and resuspended in 50 μL of molecular-grade water. 1.5 μl were used for subsequent qRT-PCR reactions with specific primers designed using Primer3web v4.1.0, within regulatory regions previously identified as BMAL1 binding sites in mouse liver, as reported in the ChIP-Atlas database[163]. Primer sequences are available in Supplementary Data 5.

## Assessment of locomotor behavior

Mice were individually housed in a light-tight, ventilated cabinet, under a 12 h light:12 h dark cycle, and ad libitum access to food and water. At the appropriate time for each treatment, animals were removed from their cages to receive IP injections for less than 2 min each. Cages were equipped with two infrared motion sensors (OASPAD system, OMNIALVA). Beam break data were continuously recorded and compiled with the OASPAD20 (OMNIALVA) software, v2019, and files containing the number of beam breaks per 6-min bin were exported. Double-plotted actograms were generated using RhythmicAlly[164]. Activity profiles were obtained averaging 5 consecutive days prior to the NAD$^+$ treatment, and 5 consecutive days after the start of the treatment. Activity profile data from 30 min were averaged for statistical comparisons.

## Statistics and reproducibility

All data were presented as the mean ± standard error of the mean, and two-way analysis of variance (ANOVA) followed by Tukey's test for multiple comparisons was used for statistical analyses except when otherwise noted in the Figure legends. Differences between groups were rated as statistically significant at $P < 0.05$. GraphPad Prism version 8.4.2.679 for Windows (GraphPad Software Inc., San Diego, CA, USA) and Excel (Microsoft Office 360, v2301) were used for statistical analyses and plotting. 24-h period rhythms were assessed employing CircWave version 1.4[165]. CircWave uses a forward linear harmonic regression to calculate the profile of the wave fitted into a 24 h period. Daily rhythms were confirmed when the null amplitude hypothesis was rejected by running an $F$ test that produced a significant value ($P < 0.05$). CircWave provides the calculation of the Centre of Gravity (CoG), representing the acrophase of the curve, with SD. Double-plotted data (ZT24) for visualization proposes are indicated in Figure legends, and were not included in the statistical analyses. Data from live mice were replicated in two independent experiments with similar results. Figures were assembled using Adobe Illustrator CC 2015 (Adobe Inc., San José, CA, USA).

## Reporting summary

Further information on research design is available in the Nature Portfolio Reporting Summary linked to this article.

## Data availability

All data generated or analyzed during this study are included in this article (and its supplementary information files). Source data are provided with this paper. All gene expression data that support the findings of this study have been deposited in the National Center for Biotechnology Information Gene Expression Omnibus (GEO) and are accessible through the GEO Series accession number: GSE163865. The

ChIP-Atlas database can be accessed at: https://chip-atlas.org/. The Investigate Mouse Gene Sets tool to compute overlaps with gene sets in MSigDB can be accessed at: http://www.gsea-msigdb.org/gsea/msigdb/mouse/annotate.jsp. Source data are provided with this paper.

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

## Acknowledgements

We thank all members of Aguilar-Arnal and Orozco-Solis laboratories for helpful discussions and advice. We thank Alicia González-Manjárrez, PhD, from the Instituto de Fisiología Celular, UNAM, México, and Rudolf M. Buijs, PhD, from the Instituto de Investigaciones Biomédicas (IIB), UNAM, México, for their suggestions and comments on this research. We thank Victor Daniel Garzón Cortés, PhD, and the Unidad de Modelos Biológicos (UMB) at the IIB for their support with animal care and maintenance. We are thankful to the Programa de Investigación de Cáncer de Mama and Dr. Alfonso Leon-del-Rio laboratory at the IIBo, particularly to Rafael Cervantes MSc, for kindly sharing reagents and equipment. We also thank the Microarray Unit at the Instituto Nacional de Medicina Genómica (INMEGEN, Mexico City) for assistance. We also thank Alfonso González-Noriega, PhD, for his generous gift of equipment, reagents, and laboratory space. Research in L.A.-A. lab was supported by grants PAPIIT IN210619, IN208022 from Universidad Nacional Autónoma de México (UNAM), the Early Career Return Grant CRP/ MEX16-05_EC from The International Center for Genomic Engineering and Biotechnology (ICGEB), Human Frontiers Science Program Young Investigators' Grant RGY0078/2017 and the National Council of Science and Technology (CONACyT) FORDECYT-PRONACES/15758/2020, all to LAA-A. R.O.-S. lab was supported by CONACyT grants FC 2016/2672 and FOSISS 272757, and the INMEGEN (09/2017/I), to R.O.-S. Q.E.-C. acknowledges the reception of PhD fellowship from CONACyT, and a fellowship from DGAPA-PAPIIT IN210619. L.M.-V. was a recipient of a postdoctoral fellowship from DGAPA-UNAM.

## Author contributions

L.A.-A. and R.O.-S. conceived and designed the study. Q.E.-C. designed and conducted all experiments. L.M.-V., R.S.-O., F.B.-P., I.P.-B., EC-V, PM-S, L.A.-A., R.O.-S., assisted with the in vivo experiments and tissue collection. M.G.-S. and L.A.-A. performed ChIP experiments. L.A.V.-V. performed extracellular flux analyses, L.M.-V and M.B.-Z. provided technical assistance. Q.E.-C., L.M.-V., R.O.-S., and L.A.-A. analyzed and interpreted the data. Q.E.-C. and L.A.-A. wrote the manuscript. All authors reviewed the manuscript and discussed the work.

## Competing interests

The authors declare no competing interests.
