## [Peer Review File · Nature Communications]

Time-of-day defines NAD⁺ efficacy to treat diet-induced metabolic disease by synchronizing the hepatic clock in miceReviewers' comments:

Reviewer #1 (Remarks to the Author):

This article by Escalante-Covarrubias and colleagues describes the time-dependent effect of NAD on obesity and type 2 diabetes in mice. Because NAD decreases in obesity, they restore a normal rhythmic NAD level in diet induced obese mice through the daily injection of NAD at ZT11, just before the feeding/active period. Their treatment leads to a decrease in body weight, an increase in insulin sensitivity, a decrease in inflammation, and a reduced steatohepatitis. These observations are supported by analysis of gene expression and activation of key signalling pathways. Interestingly, this effect is reduced when the treatment is performed at ZT23, at the end of the active period. Astonishingly, authors associate this difference to the complete inversion of the liver circadian clock and the mice are injected at ZT23. While very interesting if true, several key results or experiments are missing to support this conclusion:

- To what extent NAD concentration is impacted by injection at ZT23? This key information is missing.
- Peripheral circadian clocks and several signalling pathways (mTOR, AMPK...) are known to be entrained by feeding rhythms, including in the liver. NAD and some NAD-dependent enzymes like SIRT1 impact food intake regulation in the hypothalamus (PMID 20020036, 31900990, 33788980, 25763637, 30230244). To what extent do the injections of NAD at ZT11 and ZT23 impact feeding rhythms? Astonishingly, this crucial information is missing on figure S1C, D.
- On the same idea, authors propose that injection of NAD at ZT23 inverts the liver circadian clock. At present, only an inversion of feeding rhythm has a similar effect. Is this result due to an effect on feeding rhythms? Is it specific to the liver? Is the central clock in the suprachiasmatic nucleus also impacted and shows inverted rhythms? And the circadian locomotor and feeding activity as well?
- If this liver-specific effect of NAD is true, do injections at other time points (ZT5 or 17) have similar entrainment effects on the circadian clock?
- Authors should refrain from double plotting the data at ZT0/24. This is misleading and often artificially increases the number of statistically significant time points.

Reviewer #2 (Remarks to the Author):

In this manuscript, the authors investigate the chronotherapeutic effects of NAD⁺ on diet-induced obesity by restoring NAD⁺ oscillations with a daily timed intraperitoneal (IP) injection. NAD⁺ treatment at ZT11 ameliorates metabolic markers of diet-induced obesity such as body weight, glucose and insulin tolerance and restores hepatic gene expression related to inflammatory response and lipid metabolism. The authors demonstrated that the same treatment at ZT23 severely compromises these beneficial responses. Further analyses demonstrated that lipid oxidative pathways and the molecular clock are central mediators for phase-dependent, differential effects of NAD⁺.

These findings seem to be interesting, and most of the materials and experimental procedures were described in sufficient detail. However, some substantial concerns are identified as the major conclusions cannot be supported by the experimental data.

1. The authors demonstrate that time-of-day determines the efficacy of NAD⁺ as a therapy for diet-induced metabolic disease in mice. However, in Figure 5A/B/C/G/H, this reviewer does not see the effect of chronotherapy, because there are no significant differences in metabolic markers such as body weight, glucose, triglycerides between the two therapy groups (HFN vs HFN23). As the effect of chronotherapy is not evident, it cannot support the conclusion as noted in the title "time-of-day defines the efficacy of NAD⁺ to treat diet-induced metabolic disease ...".

2. In addition, the authors proposed in the title "... by synchronizing the hepatic circadian clock". The authors thought that NAD⁺ is a potent time-giver (or a synchronizer) for the hepatic molecular clock,

However, in Figure 7C/D, the data showed that supplementing NAD⁺ at ZT11 hardly had effect on the amplitude and period of clock genes compared to HF group (HFN vs HF). Moreover, in Figure 3C, rhythms in the core clock proteins BMAL1, CRY1, PER2 and REV-ERB α in HFN group (with NAD⁺ treatment) were not significantly different (or enhanced), compared to HF group. It is problematic to say the mechanism is to synchronize the hepatic circadian clock.

3. The authors described in the article that diet-induced obesity is known to present decreased, non-rhythmic levels of NAD⁺. While in Figure 1C, the data seems to be contradictory, with most cases are elevated (HF vs CD group). It is hard to understand the phenomena.

4. Line 105, "with antiphase increase of NAD⁺, at the onset of the rest phase, which showed only mild recovery of metabolic health". As shown in Figure 7C/D/E, the data revealed a remarkable impact of NAD⁺ treatment at ZT23 in the dynamic expression of the clock proteins CRY1, PER2 and REV-ERB α , which displayed an almost complete antiphase dynamic. Therefore, how to explain the treatment at ZT23 was to alleviate the disease rather than aggravate it.

Reviewer #3 (Remarks to the Author):

The manuscript of Escalante-Covarrubias and co-workers is original, well written, and makes an important contribution to the dissection of the role of NAD in metabolic dysregulation. With some revision, it will be meritorious of publication in Nature Communications.

Major points:

- 1) Lines 79-83 and elsewhere: There's a prevailing bias in the NAD literature that comes across in these lines as well as in the references cited that the redox functions of NAD are not very important and that the major mediators of dysfunction or NAD-based improvements in function are sirtuins. Frankly, this is wrong and it is incumbent upon authors to better understand the coenzymatic functions of NAD and the nature of evidence for sirtuins as important mediators. NAD coenzymes (NAD⁺, NADH, NADP⁺ and NADPH) perform essential functions in every cell in every tissue of the body. In contrast, in most mouse strains, you can make quite fully functioning mice without sirt1 or sirt3. They have mitochondria. They do most of the things we expect of mice. The vast majority of experiments showing increased sirt activities with NR are totally correlative and not particularly informative. More importantly, we've shown that NAD⁺ coenzymes come under attack in many conditions of metabolic stress including heart failure, obesity, type 2 diabetes, central and peripheral neurodegeneration, etc. Some of the relevant data are in ref 32, for example. Indeed, based on ref 32's observation that hepatic NADPH (in particular) is under attack in mice fed a high fat diet, the authors set out to determine whether NAD repletion at the onset of activity or NAD repletion at the onset of rest would be sufficient to improve metabolic health. The authors got a nice answer to this question but should be under no obligation to attribute that effect to sirtuin activation.
- 2) NAD⁺ was injected intact. The problem here is that while there are experiments in which NAD⁺ is applied extracellularly or to a living animal intact, NAD⁺ has to break down to NR, NAM or NA in order to enter cells. The fact that there is a mitochondrial NAD⁺ importer doesn't change this. The authors need to repeat some experiments with NR or NAM at mole-equivalent doses. Given that these are chrono experiments, I think it is okay to dose by IP.
- 3) Because NAD⁺ is such an abundant metabolite, the LC method of Yoshino is probably okay but there should be no reasonable expectation for such methods to quantify NMN. There are innumerable small molecules that coelute with NMN on LC that contribute to peak areas. They are really missing a lot of information by not doing quantitative targeted analysis of the NAD metabolome. They certainly should look at NADP⁺, which might be quantified by LC and is easily quantified by LC-MS.
- 4) The authors should understand that once mice stop gaining weight, lots of excellent things happen simultaneously. They don't have sufficient mechanistic insight to state that all of the things they are observing are direct results of repleting NAD⁺. They should also be conscious that there are entirely

different cell types that are targets including immune cells that might be calming down with NAD+ repletion, hepatocytes, that might be increasing beta oxidation programs, etc. Once one of these things starts, then a lot of other things are follow-on consequences, so they authors don't need to attribute everything to direct NAD targets. In the future, the authors may want to identify which cells are responding in which ways.

5) Lines 502-516 are highly problematic. NAD+, NADH, NADP+ and NADPH are not substrates, they are coenzymes. The sirtuin discussion is credulous and noncritical. NAD+ itself does not circulate. Many problems here.

Minor point:

Line 73: Not "the NAD salvage pathway" but "the NAM salvage pathway to NAD"

Reviewers' comments:

Reviewer #1 (Remarks to the Author):

R1: *This article by Escalante-Covarrubias and colleagues describes the time-dependent effect of NAD on obesity and type 2 diabetes in mice. Because NAD decreases in obesity, they restore a normal rhythmic NAD level in diet induced obese mice through the daily injection of NAD at ZT11, just before the feeding/active period. The treatment leads to a decrease in body weight, an increase in insulin sensitivity, a decrease in inflammation, and a reduced steatohepatitis. These observations are supported by analysis of gene expression and activation of key signalling pathways. Interestingly, this effect is reduced when the treatment is performed at ZT23, at the end of the active period. Astonishingly, authors associate this difference to the complete inversion of the liver circadian clock and the mice are injected at ZT23. While very interesting if true, several key results or experiments are missing to support this conclusion:*

A: We thank this Reviewer for the positive comments on the major findings of our research.

R1: - *To what extent NAD concentration is impacted by injection at ZT23? This key information is missing.*

A: This point is well taken. We now provide the HPLC quantitative measurements of NAD⁺ in the livers from the HFN23 group of mice at all tested time points in figure S5A and S5B. As expected, in this group of mice (HFN23) hepatic NAD⁺ reaches significantly high concentrations at ZT24, imposing circadian rhythms to hepatic NAD⁺, with an inverted phase compared to CD and HFN11 mice.

Added to text (lines 305-307):

“In this group of mice, circadian oscillations of hepatic NAD⁺ were induced with antiphase respect to CD and HFN mice, showing a peak at ZT0 and decreasing at ZT12-18 (Figure S5A, S5B)”

R1: - *Peripheral circadian clocks and several signalling pathways (mTOR, AMPK...) are known to be entrained by feeding rhythms, including in the liver. NAD and some NAD-dependent enzymes like SIRT1 impact food intake regulation in the hypothalamus (PMID 20020036, 31900990, 33788980, 25763637, 30230244). To what extent the injection of NAD at ZT11 and ZT23 impact feeding rhythms? Astonishingly, this crucial information is missing on figure S1C, D.*

A: This is an important observation. We agree with the Reviewer that to better understand the effect of NAD⁺ treatment in the control of nutrient signaling pathways it is essential to disentangle feeding patterns around the day. To approach this question, first we quantified the amount of food intake during the three weeks of treatment for all obese mice, treated with NAD⁺ at ZT11 (HFN), at ZT23 (HFN_23), and non-treated (HF) (Figure 5C), showing that all three groups eat comparable amounts of food before and after the treatment, indicating that weight loss (Figure 5B) is not due to differences in total food intake. So, as precisely suggested by this Reviewer, the next question is whether these timed NAD⁺ therapies affect the 24-hour pattern of food intake. Thus, we quantified food intake around the day and found that all groups of HFD-fed mice, treated and non-treated, ate mostly during the dark phase (Figure 7H, I). Hence, circadian oscillations in food intake were overall maintained in all three groups of mice. Importantly, the NAD⁺ therapy at ZT23 did not invert the phase of feeding cycles, hereby the phase inverted expression of the hepatic clock is not due to an inverted eating pattern, as reported before[1-4].

Added text (lines 313-314):

“Notably, total food intake was comparable between all high-fat fed mice (Figure 5C).”

Added text (lines 432-438):

“We next questioned whether feeding cycles might be altered by NAD⁺ treatment because as previously reported, this is a cause for uncoupled central and peripheral clocks^{88,89}, while NAD⁺ itself can influence feeding behavior through implicated hypothalamic circuits^{90,91}. Notably, daily food intake appeared circadian and aligned to light/dark cycles for all groups of HF diet fed mice

(Figures 7H, 7I), showing a more robust day-to-night difference for the obese mice treated with NAD⁺ at ZT11 (Figure 7I)”

R1: - *On the same idea, authors propose that injection of NAD at ZT23 invert the liver circadian clock. At present, only an inversion of feeding rhythm has a similar effect. Is this result due to an effect on feeding rhythms? Is it specific to the liver? Is the central clock in the suprachiasmatic nucleus also impacted and show inverted rhythms? And the circadian locomotor and feeding activity as well?*

A: We thank this reviewer for this important observation. We also find remarkable that a sole timed injection of NAD⁺ at ZT23 effectively inverts the phase of the hepatic clock which is clearly reminiscent of the effect described for inverted feeding cycles[1, 2]. To investigate whether timed NAD⁺ treatment could (1) invert the phase of the central clock and/or (2) reorganize feeding rhythms, we first dissected the SCN from treated and non-treated mice and determined the transcriptional levels of the clock genes *Clock*, *Bmal1*, *Per1*, *Per2* and the CCG *Dbp*, all of them known to oscillate in the SCN with distinct phases (Figure 7E). We found that timed NAD⁺ treatment did not alter the expression of these genes in the SCN. Furthermore, to obtain insights on the locomotor circadian behavior, which is robustly dictated by the SCN, we used infrared motion sensors with constant recording of beam break data from all groups of mice during five days before and 15 days after the NAD⁺ treatment. The resulting double-plotted actograms (Figure 7F) show that for all groups of mice, locomotor activity remained adjusted to light/dark cycles before and after the treatment. The quantification of the data in 30 minutes bins (five days and 5-6 mice were averaged), reveals that before the treatment, all groups of HF-fed mice were equivalent, and as such, their locomotor behavior was comparable (Figure S6F). Five days after the treatment, NAD⁺ treated mice show reduced locomotion for ~90 min after NAD⁺ injection (ZT12-13.5) in mice treated at ZT11, and ~30 min (ZT23.5-24) in mice treated at ZT23; Figure 7G). However, while these differences in locomotor activity were observed, rhythms in food intake were overall maintained and even reinforced in HFN mice, as we related in the answer for the previous question from this Reviewer (Figure 7H, I). We conclude that despite observing some differences in locomotor and feeding behavior, their extent is limited to 1-1.5 hours, keeping these activities largely aligned to light/dark cycles, and thus are not strong enough to explain the complete inverted phase on the hepatic molecular clock, showing a 12-hour phase shift in mice treated with NAD⁺ at ZT23.

Added text (lines 424-432):

“This effect on the hepatic clock gene expression has been shown for mice with inverted feeding rhythms, where the SCN clock remains aligned to light-dark cycles^{88,89}. At this regard, in all tested groups of mice, clock gene expression in the SCN remained largely intact after NAD⁺ treatment (Figure 7E, Two-way ANOVA), and locomotor behavior analyses showed that overall, NAD⁺ treatment preserved alignment between light/dark and rest/activity patterns (Figure 7F). Quantification of locomotion in 30 minutes bins revealed that after NAD⁺ treatment, mice became significantly less active for either 90 minutes (HFN) or 30 minutes (HFN_23) windows (Figures 7G, S6F, Two-way ANOVA followed by Sidak’s posttest).”

Added text to methods sections (Lines 691-697 and 800-810):

“SCN dissection

For gene expression analysis from the SCN, frozen brains were placed on ice, and the 1mm³ region above the optic chiasm was dissected out using microscissors. Tissues were placed in microcentrifuge tubes in 100 µl of Trizol and kept at -80°C until use. Total RNA was subsequently extracted and resuspended in 12 µl of water.”

“Assessment of locomotor behavior

Mice were individually housed in a light-tight, ventilated cabinet, under a 12h light:12h dark cycle, and ad libitum access to food and water. At the appropriate time for each treatment, animals were removed from their cages to receive IP injections for less than 2 minutes each. Cages were equipped with two infrared motion sensors (OASPAD system, OMNIALVA). Beam break data was continuously recorded and compiled with the OASPAD20 (OMNIALVA) software, and files containing the number of beam breaks per 6-minute bin were exported. Double-plotted actograms were generated using RhythmicAlly¹⁴³. Activity profiles were obtained averaging 5 consecutive days prior to the NAD⁺ treatment, and 5 consecutive days after the start of the treatment. Activity profile data from 30 minutes were averaged for statistical comparisons.”

Finally, we agree with the Reviewer that more peripheral clocks could be impacted by the treatment, hence this specific mention is included in the discussion section with references which facilitate contextualizing our results.

Added text (Lines 536-546)

“We have uncovered that expression of the clock genes in the SCN is largely intact upon NAD⁺ injection, and consequently, locomotor activity remains aligned with the light-dark cycles (Figure 7E-G). Uncoupled liver and SCN clocks have been previously reported in mice when access to food is restricted to the light period^{88,124,125}; however, our HFN23 mice did not show significant variations in eating behavior (Figure 7H-I). Notably, abnormal metabolic signaling triggered by high-fat diets uncouples body clocks¹⁵; thus, it would be interesting to define which extra-hepatic oscillators are reset by NAD⁺. At this regard, recent reports suggest that the brain blood barrier might be permeable to NAD⁺^{126,127} in which case hypothalamic neurons could be influenced. Yet, further research is necessary to decipher the extent of the modulation of brain clocks by increased circulating NAD⁺ precursors”

R1: *If this liver-specific effect of NAD is true, do injections at other time points (ZT5 or 17) have similar entrainment effect on the circadian clock?*

A: This is an interesting question. Since we addressed that hepatic NAD⁺ can rise one hour after IP injection, while the acrophase from its circadian oscillation is at ZT12, it is conceivable that increasing the amplitude in alignment with its natural rhythms is best to elicit beneficial therapeutic effects. Based on our results, it would be interesting to evaluate the time window around the day where NAD⁺ therapy retains and/or loses its efficacy against diet-induced

metabolic disorder, and particularly, how the molecular clock is altered along the day. At this regard, we consider that these questions were substantially addressed in the manuscript, by choosing two time points separated by 12 hours. These ZT were selected because they are either aligned (ZT11) or inverted (ZT23) with natural hepatic NAD⁺ oscillations, and this was included in the hypothesis driving our research. Thereby, we demonstrated that a NAD⁺ therapy must be aligned with the internal clock to be effective against metabolic diseases. Accordingly, the NAD⁺ injection at ZT23, did not produce significant beneficial effects and importantly, rewired the hepatic clock. Furthermore, to reinforce our study, we analyzed the effects of another NAD⁺ precursor, nicotinamide (NAM). Previous reports demonstrated that NAM supplementation in diet or water prevents diet-induced hepatic steatosis, oxidative stress and inflammation, hereby improving glucose metabolism and adiposity[5, 6]. Also, IP injection of NAM raises hepatic NAD⁺ levels in one hour[7]. Hence, we performed IP injections with NAM either at ZT11 or at ZT23 in obese mice, for three weeks. As shown in the additional supplementary information (Figure S7), the results were comparable to those obtained for the NAD⁺ treatment.

We consider that these data add additional support on the chronotherapeutic effects caused by modulating the NAD⁺ pathway, and indeed, that these new set of experiments may provide enough and convincing proof to this Reviewer on the time-dependent effects of NAD⁺-based therapies to treat obesity-associated pathologies. This is described in the new version of the manuscript as follows:

Aded text (Lines 438-449)

“Furthermore, we found similar observations when applying a therapy with the NAD⁺ precursor nicotinamide (NAM), which was previously described to boost hepatic NAD⁺ after IP injection in one hour⁹². Hence, three weeks with NAM chronotherapy performed best when applied at ZT11 to improve body weight, GTT and ITT (Figure S7A-D). As shown for NAD⁺, the NAM treatment at ZT23 inverted the expression of the hepatic molecular clock (Figure S7E), while keeping behavioral locomotor activity in phase with light/dark cycles (Figure S7F). This reinforces the notion that NAD⁺ can potentially synchronize the hepatic molecular clock, by resetting clock gene expression to adjust its phase to the time of the day when NAD⁺ bioavailability is higher. Collectively, these data support that boosting NAD⁺ levels is an effective treatment for HFD-induced metabolic disease, and demonstrate that a chronotherapeutic approach is significantly more beneficial when NAD⁺ increases at the onset of the active phase.”

Finally, as this Reviewer may notice, we have toned down our interpretation of NAD⁺ as a synchronizer of the hepatic clock across the text, while the title of the manuscript has been modified accordingly to:

“Time-of-day defines the efficacy of NAD⁺ to treat diet-induced metabolic disease by adjusting oscillations of the hepatic circadian clock”.

R1: *Authors should refrain from double plotting the data at ZT0/24. This is misleading and often artificially increase the number of statistically significant time points.*

A: We understand this reviewer’s concerns about the double plotting, and as such we want to stress the fact that we didn’t double the data to run the statistical analyses. To reinforce this, we

have now deleted any statistical information in ZT24 time point, and when the data showed statistically significant differences, these are now indicated exclusively at ZT0. Also, estimation of cyclic parameters using CircaWave do not show ZT24. Still, we strongly believe that deleting ZT24 from most of the graphs would be detrimental for the visualization of the data. Notably, double plotting ZT0 and ZT24 is a common practice, as it facilitates viewing of daily cycling. This is generally accepted by the field and can be found at publications from specialized journals such as the Journal of Biological Rhythms or Sleep, amongst others. Some examples of research showing double plotted ZT0 and ZT24 can be found on references for this response to the Reviewer: **3, and 8-23** [3, 8-23]

We are aware that this practice might raise concerns about the statistical significance of the data, hence we disclosed the double plotted data not only in all figure legends, but also in the methods section:

Added text (lines 822-824):

“Double-plotted data (ZT24) for visualization purposes are indicated in figure legends and were not included in the statistical analyses.”

Reviewer #2 (Remarks to the Author):

R2: *In this manuscript, the authors investigate the chronotherapeutic effects of NAD⁺ on diet-induced obesity by restoring NAD⁺ oscillations with a daily timed intraperitoneal (IP) injection. NAD⁺ treatment at ZT11 ameliorates metabolic markers of diet-induced obesity such as body weight, glucose and insulin tolerance and restores hepatic gene expression related to inflammatory response and lipid metabolism. The authors demonstrated that the same treatment at ZT23 severely compromises these beneficial responses. Further analyses demonstrated that lipid oxidative pathways and the molecular clock are central mediators for phase-dependent, differential effects of NAD⁺.*

These findings seem to be interesting, and most of materials and experimental procedures were described in sufficient detail. However, some substantial concerns are identified as the major conclusions cannot be supported by the experimental data.

A: We appreciate the comments from this Reviewer and thank him/her for these valuable insights.

R2: *1. The authors demonstrate that time-of-day determines the efficacy of NAD⁺ as a therapy for diet-induced metabolic disease in mice. However, in Figure 5A/B/C/G/H, this reviewer does not see the effect of chronotherapy, because there are no significant differences in metabolic markers such as body weight, glucose, triglycerides between the two therapy groups (HFN vs HFN23). As the effect of chronotherapy is not evident, it cannot support the conclusion as noted in the title “time-of-day defines the efficacy of NAD⁺ to treat diet-induced metabolic disease ...”.*

A: This point is well taken. We wish to apologize for the confusion created in the way we presented our data, particularly on Figure 5. We agree that comparisons were difficult to follow in the first version of the manuscript. We have now amended that and included more data as well. This has very much improved readability and in the new version of the manuscript, the data has been analyzed and adequately presented to avoid confusion. Additionally, we wish to state that we only consider differences based on statistical analyses, “tendencies” in our data are not considered as differences unless adequate statistical tests reveal this with a P value <0.05 . All statistical analyses are indicated in the figure legends. Hence, Figure 5B shows now the % change in body weight calculated per mice for the total period of treatment in all groups, and as can be seen, the only group losing weight after the three weeks of treatment are those supplied with NAD^+ at ZT11 (HFN). In figure 5D, we also clearly plotted serum levels of insulin at two representative time points of light (ZT6) and dark (ZT18) phases together for all groups of mice, evidencing that the HFN group corrected the hyperinsulinemia while in the HFN23 group no improvement was observed. Furthermore, differences in glucose homeostasis are now illustrated in Figure 5G, where using the AUC values from the GTT and ITT for each mouse (paired samples), we calculated the relative delta to the obese, non-treated mice, proving that at day 0, these were comparable between HF, HFN and HFN23 groups, while after 10 days on treatment improved GTT and ITT was only observed in the HFN group. After 20 days on treatment, the HFN group showed significantly higher ameliorations in GTT and ITT than those pertaining to the HFN23 group. In the case of the circulating triglycerides, we formatted the graph on Figure 5H to allow visibility of the data (mean with SEM), showing that the mice treated at ZT11 had significantly fewer circulating triglycerides than those treated at ZT23, and comparable to those detected for the lean mice ($P<0.05$, One-way ANOVA with Tukey posttest). Although steatohepatitis was reduced irrespective of the time of treatment, the previously listed differences in metabolic markers (Figure 5 and S5) strongly support the notion that a chronotherapy with NAD^+ is optimal when supplied at the onset of the active phase (ZT11).

Added text (lines 303-338):

“To investigate if the beneficial effects of pharmacological restitution of NAD^+ oscillations depend on the time of the day, we supplied NAD^+ in opposite phase to its circadian oscillation, hereby at the end of the active phase in mice, ZT23 (HFN23 group). In these HFN23 mice, circadian oscillations of hepatic NAD^+ were induced with antiphase respect to CD and HFN mice, showing a peak at ZT0 and decreasing at ZT12-18 (Figure S5A, S5B). As shown in mice treated at ZT11 (HFN), these also showed mild, albeit non-significant, weight loss after one week of treatment (Figure 5A, week 9). Contrary to the HFN group, mice supplied at ZT23 gained weight during weeks 10 and 11 (Figure 5A). Instead, after three weeks of treatment, mice treated with NAD^+ at ZT11 had lost ~5% of body weight, while those treated at ZT23 were ~2% heavier (Figure 5B), illustrating significant differences on the efficacy of the treatment depending on the time of administration. Notably, total food intake was comparable for all high-fat fed mice (Figure 5C). Serum insulin was significantly higher in mice injected at ZT23 particularly during the dark phase (Figure 5D, S5C), indicating insulin resistance in these mice, although NAD^+ therapy was effective to reduce fasting serum glucose independently of the time of supply (Figure S5D). These results indicate that in obese mice treated with NAD^+ at ZT23, insulin clearance or the feedback inhibition of insulin secretion are impaired, which is a sign of persistent metabolic dysfunction in these mice⁷⁹. Along these lines, we performed GTT and ITT at ZT4, because the effects of NAD^+ supply at ZT11 tended to be more pronounced during the

light phase (Figures 1F-I). We found that at the end of the treatment with NAD⁺ at ZT23 (day 20, HFN23), glucose and insulin tolerance showed mild, albeit non-significant, improvement compared to the HF-fed mice (Figures 5E, 5F, S5E, S5F; One-way ANOVA followed by Tukey's posttest). To quantify the relative improvement to the obese non-treated mice, we calculated the relative delta with the AUC for each mouse during the treatment (Figure 5G), showing that after 10 days of treatment, NAD⁺ was effective to improve GTT and ITT only when supplied at ZT11, but not at ZT23. At the end of the treatment (day 20), NAD⁺ supply at ZT11 showed still significantly better performance than at ZT23 (Figure 5G, Two-way ANOVA followed by Tukey's posttest).

Circulating triglycerides, largely known to be reduced by the NAD⁺ precursor niacin^{80,81}, were decreased along the day to normal levels by NAD⁺ only when the treatment was performed at ZT11, but not at ZT23 (Figure 5H). Interestingly, serum triglycerides were rhythmic for all groups; yet specific to the HFN23 group was that highest levels appeared at daytime, thus presenting antiphase circadian oscillations (Figure 5H). We found a very significant reduction in serum triglycerides only when NAD⁺ was injected at ZT11, while injection at ZT23 kept serum triglyceride levels significantly higher than injection at ZT11 (Figure 5H, P<0,05; One-way ANOVA with Tukey post-test). Besides, hepatic steatosis was reduced to a similar extent in HFN and HFN23 groups (Figure 5I-K, S5G-I)."

Finally, we thank this Reviewer for pointing this out, as this has been a very valuable comment helping us to improve this manuscript. We hope that this Reviewer is now convinced about the differential effects of NAD⁺ treatment depending on the time of administration.

R2 *2. In addition, the authors proposed in the title "... by synchronizing the hepatic circadian clock". The authors thought that NAD⁺ is a potent time-giver (or a synchronizer) for the hepatic molecular clock, However, in Figure 7C/D, the data showed that supplementing NAD⁺ at ZT11 hardly had effect on the amplitude and period of clock genes compared to HF group (HFN vs HF). Moreover, in Figure 3C, rhythms in the core clock proteins BMAL1, CRY1, PER2 and REV-ERB α in HFN group (with NAD⁺ treatment) were not significant different (or enhanced), compared to HF group. It is problematic to say the mechanism is to synchronize the hepatic circadian clock.*

A: We understand this reviewer's concerns about the synchronization of the clock by NAD⁺, as the intervention at ZT11 largely keeps the clock in place. Notably, NAD⁺ oscillates with an acrophase at ZT12; hence it is reasonable to find that when we supply NAD⁺ in the obese mice, restoring the oscillations and phase to ZT12, the clock will not generally be affected. The effects were observed when NAD⁺ is supplied at ZT23, when the hepatic clock phase is inverted. In the absence of daily NAD⁺ fluctuations (i.e., obese, non-treated mice), the molecular clock is generally kept in phase by means of additional signaling, including the SCN-registered light inputs and rhythmic food intake (see new data on Figure 7E-I), for example.

In Figure 3C, there is a robust and reproducible effect that we could detect in the expression of the CRY1 protein, where NAD⁺ treatment significantly decreased CRY1 levels specifically at ZT12 (P<0.05, Two-Way ANOVA with Tukey Posttest). This effect shaped CRY1 oscillatory pattern, altering the phase and the amplitude as revealed by CircaWave analyses as follows:

	CD	HF	HFN
Acrophase (hour)	19.898	22.053	23.445
SE	1.391	1.747	1.234
Amplitude	0.54	0.8	1.13

Hence, in CRY1 protein, NAD⁺ treatment has a visible effect consisting of a phase delay of 1.4±1.2 hours (compared to HF group), or 3.4±1.2 hours (compared to CD group), and increased amplitude. This is probably due to increased activity of AMPK at ZT12 in NAD⁺ treated mice (Figure 4F), which is known to phosphorylate CRY1 in mouse liver and destabilize it[24]. This observation is described in the results section, lines 249-253.

We could not detect changes in the transcriptional profiles of clock genes in the HFN mice. At this regard, it has been described that obese mice present a 4-hour phase advance in the transcriptional oscillatory profile from clock genes [25], however our sampling resolution was of 6 hours; hereby, a 4-hour phase shift could be undetectable. It would be interesting to dissect with high resolution (i.e. 2-3 hours), the effects of NAD⁺ injections at ZT11 or other ZTs, on the hepatic clock machinery, as well as other peripheral clocks, but we consider that this would require many mice and samples, constituting a new avenue that is now out of the scope of the research presented in this manuscript.

Following this Reviewer comment, we have systematically and consistently toned down the conclusions of the study by removing the role of NAD⁺ as a general “potent synchronizer” as this might depend on the time of administration. We have modified the title as follows:

“Time-of-day defines the efficacy of NAD⁺ to treat diet-induced metabolic disease by adjusting oscillations of the hepatic circadian clock”

R2: *3. The authors described in the article that diet-induced obesity is known to present decreased, non-rhythmic levels of NAD⁺. While in Figure 1C, the data seems to be contradictory, with most cases are elevated (HF vs CD group). It is hard to understand the phenomena.*

This point is well taken. We observed that within our HPLC data from HF mice at ZT0, there was one mouse whose hepatic NAD⁺ content was unusually high, close to 20mM/mg. We have now included additional liver samples from 3 obese mice sacrificed at ZT0, and analyzed them by HPLC, showing values for their hepatic NAD⁺ content between 14-16 mM/mg. Hence, we didn't exclude the data from the mouse with highest NAD⁺ levels. Instead, including the new measurements (7 mice were analyzed per triplicate), the average NAD⁺ content at ZT0 decreased (Figure 1C). It is possible that the mice with high hepatic NAD⁺ content altered his eating behavior just before the sacrifice or suffered some health condition that we couldn't detect; however, as previously described, we chose not to exclude the data because we don't have a strong argument for that. We thank the Reviewer for this observation, as taking additional measurements improved the quality of this data-point.

Additionally, we measured by HPLC the hepatic NAD⁺ content from mice injected with NAD⁺ at ZT23 (Supplementary Figure S5A, S5B), which was absent in the first version of our

manuscript. We found that circadian NAD⁺ oscillations, as determined by CircaWave, were lacking only in the livers from the obese, non-treated mice, and as expected, the HFN23 group presented inverted phase.

R2: *4.Line 105, “with antiphase increase of NAD⁺, at the onset of the rest phase, which showed only mild recovery of metabolic health”. As shown in Figure 7C/D/E, the data revealed a remarkable impact of NAD⁺ treatment at ZT23 in the dynamic expression of the clock proteins CRY1, PER2 and REV-ERB α , which displayed an almost complete antiphase dynamic. Therefore, how to explain the treatment at ZT23 was to alleviate the disease rather than aggravate it.*

A: We understand this Reviewer’s concerns about the interpretation of the data, because as discussed in the first comment, the way we organized the results in Figure 5 could had been confusing. We hope that after carefully re-analyzing the data in Figure 5, we have now convinced this Reviewer that time of day defines the efficacy of a NAD⁺ therapy, as obese mice treated at ZT11 showed significant improvement in many metabolic parameters, while those treated at ZT23 did not. Indeed, ameliorations in ZT23-treated mice were observed for hepatic steatosis, but this constituted an exception. We think that some beneficial pathways might be enhanced in ZT23-treated mice oriented to decrease hepatic lipid content, however a major and detrimental side effect was precisely inverting the phase of the hepatic clock. This is because it has been extensively demonstrated that circadian misalignment leads to metabolic disease in both mouse models and humans[26-29]. For example, restricted feeding entrains the liver clock while the SCN clock remains intact [1, 4, 30], and when access to food is limited to the light period in mice (i.e. their resting phase), misalignment between central (SCN, light entrained) and peripheral (liver, food entrained) clocks occurs, leading to metabolic dysfunction [31-33]. So far, misalignment between the SCN and the hepatic clock has only been described by means of inverted feeding (or other interventions such as shift-work which generally end up inverting feeding behavior). Here we describe for the first time that a pharmacological intervention at the onset of the rest phase (ZT23) uncouples the central and peripheral clocks as a side effect, which opposes the beneficial effects of NAD⁺ to treat obesity. Indeed, on the long term, chronic treatment at ZT23 could potentially worsen metabolic disease.

This rationale has now been strengthened with additional data shown in Figures 7E-I. Precisely, Figure 7E shows that the SCN clock remains largely intact upon NAD⁺ treatment, while locomotor behavior (Figure 7F, 7G) is also aligned with the light/dark cycles. Additionally, food intake still follows a circadian pattern, with most of the feeding occurring during the dark, active phase in all groups of mice. These data demonstrates that the uncoupling of central and peripheral clocks occurs in the HFN23 mice, which is a hallmark of metabolic disruption[34]. This has been described in the new version of the manuscript as follows:

Added to lines 424-438

“This effect on the hepatic clock gene expression has been shown for mice subjected to inverted feeding rhythms, where the SCN clock remains aligned to light-dark cycles^{88,89}. At this regard, in all tested groups of mice, clock gene expression in the SCN remained largely intact after NAD⁺ treatment (Figure 7E, Two-way ANOVA), and locomotor behavior analyses showed that overall, NAD⁺ treatment preserved alignment between light-dark and rest-activity patterns

(Figure 7F). Quantification of locomotion in 30 minutes bins revealed that after NAD⁺ treatment, mice became significantly less active for either 90 minutes (HFN) or 30 minutes (HFN_23) windows (Figures 7G, S6F, Two-way ANOVA followed by Sidak's posttest). We next questioned whether feeding cycles might be altered by NAD⁺ treatment because as previously reported, this is a cause for uncoupled central and peripheral clocks^{88,89}, while NAD⁺ itself can influence feeding behavior through implicated hypothalamic circuits^{90,91}. Notably, daily food intake appeared circadian and aligned to light-dark cycles for all groups of HF diet fed mice (Figures 7H, 7I), showing a more robust day-to-night difference for the obese mice treated with NAD⁺ at ZT11 (Figure 7I)."

Added to the discussion, Lines 534-546:

"At ~ZT0, circadian misalignment imposed by antiphase NAD⁺ in our HFN23 and HFNAM_23 mice, might obstruct metabolic improvements, through uncoupling of the central light-synchronized and peripheral NAD⁺-synchronized clocks. We have uncovered that expression of the clock genes in the SCN is largely intact upon NAD⁺ injection, and consequently, locomotor activity remains aligned with the light-dark cycles (Figure 7E-G). Uncoupled liver and SCN clocks have been previously reported in mice when access to food is restricted to the light period^{88,124,125}; however, our HFN23 mice did not show significant variations in eating behavior (Figure 7H-I). Notably, abnormal metabolic signaling triggered by high-fat diets uncouples body clocks¹⁵; thus, it would be interesting to define which extra-hepatic oscillators are reset by NAD⁺. At this regard, recent reports suggest that the brain blood barrier might be permeable to NAD⁺^{126,127} in which case hypothalamic neurons could be influenced. Yet, further research is necessary to decipher the extent of the modulation of brain clocks by increased circulating NAD⁺ precursors."

Finally, it is possible that on the long term, NAD⁺ treatment at ZT23 would aggravate metabolic disease; yet two to three weeks of uncoupling central/peripheral clocks as a side effect of the treatment are not sufficient to impose deterioration over improvement. In any case, our data show that time of treatment is crucial to maximize the benefits of rising NAD⁺ levels.

Reviewer #3 (Remarks to the Author):

R3: *The manuscript of Escalante-Covarrubias and co-workers is original, well written, and makes an important contribution to the dissection of the role of NAD in metabolic dysregulation. With some revision, it will be meritorious of publication in Nature Communications.*

A: We are pleased that this Reviewer gives these positive observations and recommends publication of our manuscript in Nature Communications after some revision. Thank you for your thoughtful comments.

R3: Major points:

1) Lines 79-83 and elsewhere: There's a prevailing bias in the NAD literature that comes across in these lines as well as in the references cited that the redox functions of NAD are not very important and that the major mediators of dysfunction or NAD-based improvements in function are sirtuins. Frankly, this is wrong and it is incumbent upon authors to better understand the coenzymatic functions of NAD and the nature of evidence for sirtuins as important mediators. NAD coenzymes (NAD⁺, NADH, NADP⁺ and NADPH) perform essential functions in every cell in every tissue of the body. In contrast, in most mouse strains, you can make quite fully functioning mice without sirt1 or sirt3. They have mitochondria. They do most of the things we expect of mice. The vast majority of experiments showing increased sirt activities with NR are totally correlative and not particularly informative. More importantly, we've shown that NAD⁺ coenzymes come under attack in many conditions of metabolic stress including heart failure, obesity, type 2 diabetes, central and peripheral neurodegeneration, etc. Some of the relevant data are in ref 32, for example. Indeed, based on ref 32's observation that hepatic NADPH (in particular) is under attack in mice fed a high fat diet, the authors set out to determine whether NAD depletion at the onset of activity or NAD depletion at the onset of rest would be sufficient to improve metabolic health. The authors got a nice answer to this question but should be under no obligation to attribute that effect to sirtuin activation.

A: This point is well taken; we agree with this Reviewer that sirtuins as major effectors of NAD⁺ fluctuations has been generally overemphasized in the literature. Yet, in our manuscript, we present interesting observations which are not necessarily related to sirtuins' activity, including overexpression of genes related to membrane trafficking and autophagy after NAD⁺ treatment, and a strong effect on the mTOR pathway. Indeed, we performed some SIRT1 protein analyses by western blot in our mice, and found that this protein was overexpressed in livers from obese mice, which contradicts a number of reports[35, 36], although is aligned with others[5, 37], as one of the many examples of the existing controversy around sirtuins in the field. Hence, we have been very careful in our research to not let this bias (i.e, sirtuins as unique effectors of NAD⁺ fluctuations) affect the development of the project and as such, we refrained to include our sirtuin-related data or to attribute the NAD⁺ effects to sirtuins' activity, as we found that this was not adding new hints to our research or advancing the field. We wanted to tackle our question/hypothesis, which nicely addressed this Reviewer in the comment above, from a broader perspective.

However, in the circadian field, sirtuins attracted a lot of attention after two major papers appeared in 2008, contributed by leaders in the field[38, 39]. These two papers demonstrate that the molecular clock proteins PER2 and BMAL1, are substrates for SIRT1 deacetylation, hence altering their function with an impact in the oscillations' properties, such as period length or amplitude. A year later, two papers by two different groups, also leaders in the field, described daily NAD⁺ oscillations in mouse liver directed by the transcriptional oscillations of the clock-controlled gene *Nampt*, which in turn "modulate the activity of SIRT1 feeding back into the circadian clock"[40, 41]. In 2013, another seminal paper described that cyclic NAD⁺ imposes oscillations on the activity of the mitochondrial sirtuin SIRT3, which in turn, rhythmically deacetylates mitochondrial proteins to drive rhythms in oxidative metabolism[42]. In 2015, some of the authors of this manuscript under review described that SIRT1 in the presence of NAD⁺ could deacetylate and modulate the activity of the epigenetic modifier MLL1, which collaborates

with the clock machinery to generate a cyclic epigenetic environment in the promoters of clock genes, however most of these findings were obtained “*in vitro*”[43]. Interestingly, a recent research using liver specific *Sirt1* and *Sirt6* KO mice showed that the impact of sirtuins deletion on the hepatic circadian transcriptome, and particularly on clock genes, is surprisingly modest[44], raising the hypothesis that maybe sirtuins are largely dispensable for circadian function under normal conditions. At this regard, last year it was shown that decreased NAD⁺ in livers from aged mice is accompanied by increased PER2 acetylation, and BMAL1 loss of function, which is corrected by NR supplementation in drinking water of old mice[45]. Surprisingly, in this research, using NMN IP injections as NAD⁺ precursor, Levine *et al.* showed that mice lacking hepatic *Sirt1* no longer reconstitute the expression of NMN-responsive genes[45], concluding that SIRT1 is the effector of NAD⁺ at the transcriptional level through regulating PER2 deacetylation; however these experiments were not performed in aged mice or using a circadian paradigm, while the rest of evidence were obtained “*in vitro*”, raising additional questions. Importantly, in calorie restricted mice, hepatic cyclic protein acetylation is altered showing constant protein hyperacetylation during the circadian cycle, while the hepatic NAD⁺ metabolome appears with increased NAD⁺ and NADP⁺ levels[46]. At this regard, the role of sirtuins in regulating circadian rhythms in response to NAD⁺ levels remain largely unclear, although we think that is still important for the field and as such, cannot be overlooked. Thus, we are convinced that this groundbreaking research should be cited in our manuscript, as is important for the circadian field, and some of our data can be contextualized in the discussion section considering these previous findings.

With this background, as we still agree with this Reviewer that NAD⁺ *in vivo* is much more relevant than the attributed function as a substrate for sirtuins and other NAD⁺ consuming enzymes, we have modified the introduction and the discussion sections and included new references which give a global, less sirtuin-centric view, as follows:

Added text (Lines 76-86):

“NAD⁺ and its phosphorylated and reduced forms NADH, NADP⁺ and NADPH, are coenzymes for hydride transfer enzymes, crucial to biological redox reactions. NAD⁺/NADH ratio is a basic determinant of the rate of catabolism and energy production^{29,30}. In fed state or nutrient overload NAD⁺/NADH ratio falls, and a prolonged redox imbalance potentially leads to metabolic pathologies, such as diabetes³¹. Along these lines, extensive research demonstrates that NAD⁺ levels significantly decline in metabolic tissues of obese mice and humans³²⁻³⁷. NAD⁺ decay itself may contribute to metabolic dysfunction by distinct mechanisms, including increased oxidative stress and ROS production, disbalance in the oxidative-reductive capacity, disrupted Ca²⁺ homeostasis or reduced activity of sirtuins^{38,39}; a class of deacetylase enzymes using NAD⁺ as cofactor and known to influence mitochondrial function and metabolism.”

R3: 2) *NAD⁺ was injected intact. The problem here is that while there are experiments in which NAD⁺ is applied extracellularly or to a living animal intact, NAD⁺ has to break down to NR, NAM or NA in order to enter cells. The fact that there is a mitochondrial NAD⁺ importer doesn't change this. The authors need to repeat some experiments with NR or NAM at mole-equivalent doses. Given that these are chrono experiments, I think it is okay to dose by IP.*

A: This is an excellent observation. We agree with this Reviewer that NAD⁺ treatment might appear notorious for the reasons mentioned in this comment. As such, following this Reviewer's suggestion, we performed a new experimental paradigm where we treated obese mice with nicotinamide (NAM) for three weeks either at ZT11 or at ZT23, obtaining comparable results to the NAD⁺ treatment, which are shown in supplementary figure S7. At ZT11, NAM treatment performed significantly better than at ZT23 to reduce body weight and improve GTT and ITT (Figure S7A-D). Importantly, the hepatic clock was altered by NAM treatment at ZT23 to the same extent than the NAD⁺ treatment did (Figure S7E), while preserving the SCN-dependent locomotor activity (Figure S7F). In this case, we IP injected 200mg/kg of NAM, representing more than an equimolar dose suggested by the Reviewer, which corresponds to 9 mg/kg. We selected this dose based on the available literature, where we found that IP injections with NAM have been tested in rodents at doses ranging between 100-500 mg/kg. An important evidence for selecting this dose came from a previous report showing that a single IP injection with 500 mg/kg NAM doubled hepatic NAD⁺ levels in just one hour, and these were 4 times higher after 4 hours [7]. Also, IP injection of 200mg/kg [47] and 500 mg/kg [48] of NAM for 15 days have been previously reported to improve certain pathological conditions in rodents.

We have described these results as follows (Lines 438-446):

“Furthermore, we found similar observations when applying a therapy with the NAD⁺ precursor nicotinamide (NAM), which was previously described to boost hepatic NAD⁺ after IP injection in one hour⁹². Hence, three weeks with NAM chronotherapy performed best when applied at ZT11 to improve body weight, GTT and ITT (Figure S7A-D). As shown for NAD⁺, the NAM treatment at ZT23 inverted the expression of the hepatic molecular clock (Figure S7E), while keeping behavioral locomotor activity in phase with light/dark cycles (Figure S7F). This reinforces the notion that NAD⁺ can potentially synchronize the hepatic molecular clock, by resetting clock gene expression to adjust its phase to the time of the day when NAD⁺ bioavailability is higher”

R3: 3) *Because NAD⁺ is such an abundant metabolite, the LC method of Yoshino is probably okay but there should be no reasonable expectation for such methods to quantify NMN. There are innumerable small molecules that coelute with NMN on LC that contribute to peak areas. They are really missing a lot of information by not doing quantitative targeted analysis of the NAD metabolome. They certainly should look at NADP⁺, which might be quantified by LC and is easily quantified by LC-MS.*

A: We understand this Reviewer's comment about the importance of the NAD⁺ metabolome. Indeed, we tried to do this, but with very limited success. During the last few months, the facilities at our University have remained mostly closed due to the widespread infection by the Delta variant. Despite this, we could get access to a mass-spec facility once a week for limited hours and with limited technical assistance. The equipment details were:

Agilent 1200 SL System with autosampler liquid chromatograph with a Zorbax Eclipse Plus 100 x 2.1 mm, 3.5 μm; coupled with a Bruker Daltonics Esquire 6000 ESI ion trap mass spectrometer with the following parameters:

LC: The mobile phase was prepared from a 5mM ammonium formate solution and methanol. The flow rate was 0.2 ml/min at RT.

MS: Mode Tuns SPS; Target Mass 663 m/z; Compound Stability 90 %; Trap Drive Level 100 %; Optimize Normal; Smart Parameter Setting active; Mass Range Mode Std/Normal; Ion Polarity Negative; Ion Source Type ESI; Alternating Ion Polarity off; Current Alternating Ion Pol Positive; Dry Temp (Set) 300 °C; Nebulizer (Set) 10.00 psi; Dry Gas (Set) 5.00 l/min; HV Capillary 4000 V; HV End Plate Offset -500 V; Rolling on; Rolling, Averages 2 cts; Scan Begin 50 m/z; Scan End 1000 m/z; Averages 5 Spectra; Max. Accu Time 200000 μs; ICC Target 10000

With this set up, we could detect and measure NAD⁺, however other components of the NAD⁺ metabolome were not detected, since many of these share similar structure and close mass. Indeed, we had a limitation with the equipment, as the ion trap mass analyzer might not be sufficient to quantify these metabolites, and a QQQ is optimal. In this scenario, we strongly believe that to identify the components of the NAD⁺ metabolome we should arrange a collaboration with experts in the field, located abroad[49]. However, it has been really challenging to keep up with collaborations due to the pandemic situation and especially, travel bans. Additionally, shipping samples abroad in dry ice has become a problem, as these services have been halted in our University during the COVID19 pandemic. Hereby, the only available resource that we can think off to measure the NAD⁺ metabolome are commercial kits, however these are less accurate than LC-MS measurements, and quite expensive.

We understand that these data would be of general interest to our research, however we are convinced that the results would not affect the most relevant conclusions of the paper: 1) the efficacy of NAD⁺ treatment for obesity depends on the time of administration, and 2) the hepatic clock oscillations are shaped by NAD⁺ levels, while the SCN-driven behavioral rhythms are overall unaffected. Also, we would like to highlight that unraveling and quantifying the NAD⁺ metabolome from 80 liver samples would require a substantial effort, coordination between distinct collaborators and administrative staff, as well data analyses and interpretation, which indeed, would probably involve additional experiments to resolve questions arising from this set of data. For these reasons, we conclude that these measurements constitute grounds for novel avenues and are now out of the scope of the present research. Certainly, we will for sure pursue this in the future. We would like to propose that if this Reviewer still considers so, we could perform measurements of NADP⁺/NADPH ratio with a commercial kit in our liver samples, although to our opinion, the information of these measurements would be of limited reach compared to the metabolome. This is in part due to that both NADP⁺ and NADPH are decreased in livers from obese mice[50], NADP⁺ appears upregulated in livers from calorie-restricted mice[46], whereas the PPP pathway, a major source of NADPH, regulates circadian oscillations in a NADPH-dependent manner[51]. These temporal variations in NADP⁺ and NADPH themselves would be complicated to discern by measuring NADP⁺/NADPH ratio.

Finally, we have added a note in the discussion about the limitations of our study by lacking these measurements, as follows:

Added to text (lines 473-477):

“Yet, to gain insights into the bioavailability of NAD⁺ precursors in our study, it would be necessary to unravel the hepatic NAD⁺ metabolome in all tested conditions, as for example, the possibility that time-dependent decline in NADPH and NADP⁺ levels in livers from obese mice^{27,35,103} contributes to differences between HFN and HFN23 mice cannot be ruled out, constituting a limitation of our study.”

R3: 4) *The authors should understand that once mice stop gaining weight, lots of excellent things happen simultaneously. They don't have sufficient mechanistic insight to state that all of the things they are observing are direct results of repleting NAD⁺. They should also be conscious that there are entirely different cell types that are targets including immune cells that might be calming down with NAD⁺ repletion, hepatocytes, that might be increasing beta oxidation programs, etc. Once one of these things starts, then a lot of other things are follow-on consequences, so they authors don't need to attribute everything to direct NAD targets. In the future, the authors may want to identify which cells are responding in which ways.*

A: This is an interesting point; we agree with the Reviewer that new and appealing avenues come with our research. As suggested, we have toned down some of our statements about the NAD⁺ direct effects in the new version of the manuscript, and stated the need for more profound analyses in the discussion, as follows:

Added text (lines 546-551):

“Additionally, our study is limited by the cellular heterogeneity in fatty liver, with for example, infiltration of pro-inflammatory macrophages which have been recently shown to limit NAD⁺ bioavailability through high expression of the NAD-consuming enzyme CD38^{128,129}. Hence, it is possible that time-dependent cellular heterogeneity in liver¹³⁰ could contribute to the NAD⁺-dependent improvement of the metabolic phenotype.”

Nonetheless, it is interesting to note that mice treated with NAD⁺ or NAM at ZT23 lost some weight after the first week of treatment. However, weight loss was not sustained during weeks 2 and 3. We collected new measurements on food intake for this revised version of the manuscript (new data added to Figures 5C, 7H, 7I), showing that all groups of HF-fed mice ate comparable amounts of food, and their rhythms in food intake were overall maintained after the treatment. Collectively, these data indicate that the NAD⁺ treatment has a positive effect on weight loss, the extent of which is largely determined by the time of administration. As this Reviewer points out, many beneficial effects simultaneously happen after the NAD⁺ treatment, including weight loss, hence trying to address a unique molecular mechanism responsible for all these interlocked “chain reactions” might be ineffective and probably flawed, due to the complexity of the system. Instead, we address a previously unappreciated role for the hepatic circadian clock machinery, which in HFN23 mice, adjust its phase to the time of NAD⁺ treatment, and consequently, oscillations of the hepatic clock and clock-controlled transcriptional oscillations become misaligned with environmental cues, including food intake and light-dark cycles (new data added to Figures 7E-I), and consequently, oppose the potential of NAD⁺ to treat metabolic dysfunction. We agree that liver heterogeneity should be considered and probably, advancing spatially resolved single-cell metabolomics techniques might provide novel tools to address this.

R3: 5) *Lines 502-516 are highly problematic. NAD⁺, NADH, NADP⁺ and NADPH are not substrates, they are coenzymes. The sirtuin discussion is credulous and noncritical. NAD⁺ itself does not circulate. Many problems here.*

A: This is well taken. We have modified these points in the discussion, we apologize for the lack of precision and thank this Reviewer for pointing this out. We indeed made a mistake by swapping the role of NAD⁺ as substrate of NAD⁺-consuming enzymes and coenzyme in redox reactions, and now has been corrected with the adequate references[52, 53] included as follows:

Added text (Lines 523-525):

“NAD⁺ is a coenzyme in redox reactions, but also serves as a substrate of NAD⁺ consuming enzymes which cleave NAD⁺ to produce NAM and an ADP-ribosyl product, such as ADP-ribose transferases, cADP-ribose synthases and sirtuins (SIRT1-SIRT7)^{94,120.}”

Regarding the affirmation that NAD⁺ itself does not circulate, we also have corrected this sentence to “circulating NAD⁺ precursors” as is much more reasonable based on the current evidence: extracellular NAD⁺ is catabolized to NR or NAM to enter the cell. Interestingly some research quantifying the human plasma NAD⁺ metabolome shows that plasma NAD⁺ levels decline with age[54], while other quantitative analyses of serum NAD⁺ in mammals exists[55], showing that although much lower than in tissues, plasma (or what we would understand for “circulating”) NAD⁺ can be quantified. Also, NAD⁺ incorporation into cells has been quantified and shown to be fast[56], and although the breakdown of NAD⁺ cannot be ruled out from these experiments, the possibility of a NAD⁺ transporter in the plasma membrane has not been completely discarded yet[57]. Indeed, last year a mitochondrial transporter for NAD⁺ was described[58, 59], constituting, to our opinion, a groundbreaking finding for the field. These are very exciting questions that remain to completely understand our data; however, we consider that they are out of the scope of this research. We have included some hints and references by modifying the discussion section as follows

Added text (lines 463-473):

“NAD⁺ and its phosphorylated and reduced forms, NADP⁺, NADH and NADPH, are fundamental compounds in intermediary metabolism as hydride-accepting or -donating coenzymes in redox reactions⁹⁴. NAD⁺ is produced in all tissues from the salvageable precursors NAM, nicotinamide riboside (NR) or niacin, while some tissues such a liver produce NAD⁺ *de novo* from tryptophan, in a much less efficient biosynthetic pathway^{30,95}. It is generally accepted that NR or NAM enter the cell⁹⁶, while extracellular NAD⁺ and NMN are converted to NR⁹⁷. At this regard, the NAD⁺ precursors NAM, NMN and NR have been preferentially used as NAD⁺ boosters; however, we set up a therapy with NAD⁺ due to that the limited data tracing metabolic fluxes suggest distinct, tissue-specific effects of NR and NMN⁹⁸. Moreover, NAD⁺ uptake appears fast and effective in cells and a mitochondrial active transporter has been recently described^{99-102.}”

This Reviewer indicates that “the sirtuin discussion is credulous and noncritical”. We agree that sirtuins are hyped in the literature as the major NAD⁺ effectors, and accordingly, the arguments around sirtuins in the discussion conforms less than 10% of the section. As mentioned before, (see response to major point 1), the regulation of the clock machinery through deacetylation by the NAD⁺ -dependent sirtuins still constitutes a remarkable finding for the field. Although some specific data regarding to either BMAL1 or PER2 or both as SIRT1 deacetylation targets, might be conflicting, the general role of sirtuins as rheostats of the clock machinery has not been disproved yet, and based on our results, we don’t have grounds to challenge this view in the discussion of our manuscript, where the length is also limiting, while simultaneously these

previous findings constitute direct background for our research. It would be of interest to recapitulate all these evidence with a critical view in a dedicated review or perspective article. Indeed, a nice, well balance review has been recently published relating many intriguing links between redox metabolism and circadian transcription beyond sirtuins[60]. However, we have now toned down the statements related to sirtuins as we completely agree with this Reviewer that still, many questions remain open. In the new version of our manuscript, reference to sirtuins in the discussion remains less than 10% and has been revised, as follows:

Revised text (lines 525-534)

“Indeed, both NAD⁺ consumers SIRT1^{46,121,122} and SIRT3²¹ provide reciprocal regulation to the clock machinery to modulate circadian transcription and metabolism in the liver. Furthermore, recent research shows that a NAD⁺-SIRT1 interplay mediate deacetylation and nuclear translocation of PER2 and shape BMAL1 function, and this control is altered in livers from aged mice⁴⁶. Through activation of SIRT1 and SIRT3, it is possible that rising NAD⁺ at ~ZT12 might contribute to rhythmic lipid oxidation and mitochondrial function driven by protein acetylation, including PPAR γ ^{27,123}, while keeping the hepatic clock aligned to the external time. Yet, the regulation of the circadian system by sirtuins in health and disease remains to be fully disentangled”

Minor point:

R3: Line 73: Not “the NAD salvage pathway” but “the NAM salvage pathway to NAD”

A: This sentence was corrected as suggested.

REFERENCES

1. Damiola, F., et al., *Restricted feeding uncouples circadian oscillators in peripheral tissues from the central pacemaker in the suprachiasmatic nucleus*. *Genes Dev*, 2000. **14**(23): p. 2950-61.
2. Saini, C., et al., *Real-time recording of circadian liver gene expression in freely moving mice reveals the phase-setting behavior of hepatocyte clocks*. *Genes Dev*, 2013. **27**(13): p. 1526-36.
3. Girotti, M., M.S. Weinberg, and R.L. Spencer, *Diurnal expression of functional and clock-related genes throughout the rat HPA axis: system-wide shifts in response to a restricted feeding schedule*. *Am J Physiol Endocrinol Metab*, 2009. **296**(4): p. E888-97.
4. Hara, R., et al., *Restricted feeding entrains liver clock without participation of the suprachiasmatic nucleus*. *Genes Cells*, 2001. **6**(3): p. 269-78.
5. Mitchell, S.J., et al., *Nicotinamide Improves Aspects of Healthspan, but Not Lifespan, in Mice*. *Cell Metabolism*, 2018. **27**(3): p. 667-676.e4.
6. Mendez-Lara, K.A., et al., *Nicotinamide Protects Against Diet-Induced Body Weight Gain, Increases Energy Expenditure, and Induces White Adipose Tissue Beiging*. *Mol Nutr Food Res*, 2021. **65**(11): p. e2100111.
7. Schein, P.S. and S. Loftus, *Streptozotocin: depression of mouse liver pyridine nucleotides*. *Cancer Res*, 1968. **28**(8): p. 1501-6.
8. Saderi, N., et al., *Differential Recovery Speed of Activity and Metabolic Rhythms in Rats After an Experimental Protocol of Shift-Work*. *J Biol Rhythms*, 2019. **34**(2): p. 154-166.

9. Vancura, P., et al., *Rhythmic Regulation of Photoreceptor and RPE Genes Important for Vision and Genetically Associated With Severe Retinal Diseases*. Invest Ophthalmol Vis Sci, 2018. **59**(10): p. 3789-3799.
10. van den Berg, R., et al., *A Diurnal Rhythm in Brown Adipose Tissue Causes Rapid Clearance and Combustion of Plasma Lipids at Wakening*. Cell Rep, 2018. **22**(13): p. 3521-3533.
11. Seaton, D.D., et al., *Dawn and photoperiod sensing by phytochrome A*. Proc Natl Acad Sci U S A, 2018. **115**(41): p. 10523-10528.
12. Lam, V.H., et al., *CK1alpha Collaborates with DOUBLETIME to Regulate PERIOD Function in the Drosophila Circadian Clock*. J Neurosci, 2018. **38**(50): p. 10631-10643.
13. Barca-Mayo, O., et al., *Astrocyte deletion of Bmal1 alters daily locomotor activity and cognitive functions via GABA signalling*. Nat Commun, 2017. **8**: p. 14336.
14. Petsakou, A., T.P. Sapsis, and J. Blau, *Circadian Rhythms in Rho1 Activity Regulate Neuronal Plasticity and Network Hierarchy*. Cell, 2015. **162**(4): p. 823-35.
15. Curie, T., et al., *In Vivo Imaging of the Central and Peripheral Effects of Sleep Deprivation and Suprachiasmatic Nuclei Lesion on PERIOD-2 Protein in Mice*. Sleep, 2015. **38**(9): p. 1381-94.
16. Curie, T., et al., *Homeostatic and circadian contribution to EEG and molecular state variables of sleep regulation*. Sleep, 2013. **36**(3): p. 311-23.
17. Guerra, P.A., et al., *Discordant timing between antennae disrupts sun compass orientation in migratory monarch butterflies*. Nat Commun, 2012. **3**: p. 958.
18. Delezie, J., et al., *The nuclear receptor REV-ERBalpha is required for the daily balance of carbohydrate and lipid metabolism*. FASEB J, 2012. **26**(8): p. 3321-35.
19. Hampp, G., et al., *Regulation of monoamine oxidase A by circadian-clock components implies clock influence on mood*. Curr Biol, 2008. **18**(9): p. 678-83.
20. Merlin, C., et al., *An antennal circadian clock and circadian rhythms in peripheral pheromone reception in the moth Spodoptera littoralis*. J Biol Rhythms, 2007. **22**(6): p. 502-14.
21. Spanagel, R., et al., *The clock gene Per2 influences the glutamatergic system and modulates alcohol consumption*. Nat Med, 2005. **11**(1): p. 35-42.
22. Carr, A.J., et al., *Photoperiod differentially regulates circadian oscillators in central and peripheral tissues of the Syrian hamster*. Curr Biol, 2003. **13**(17): p. 1543-8.
23. Field, M.D., et al., *Analysis of clock proteins in mouse SCN demonstrates phylogenetic divergence of the circadian clockwork and resetting mechanisms*. Neuron, 2000. **25**(2): p. 437-47.
24. Lamia, K.A., et al., *AMPK Regulates the Circadian Clock by Cryptochrome Phosphorylation and Degradation*. Science, 2009. **326**(5951): p. 437-440.
25. Eckel-Mahan, K.L., et al., *Reprogramming of the circadian clock by nutritional challenge*. Cell, 2013. **155**(7): p. 1464-78.
26. Scheer, F.A., et al., *Adverse metabolic and cardiovascular consequences of circadian misalignment*. Proc Natl Acad Sci U S A, 2009. **106**(11): p. 4453-8.
27. Parameswaran, G. and D.W. Ray, *Sleep, circadian rhythms, and type 2 diabetes mellitus*. Clin Endocrinol (Oxf), 2021.
28. Leproult, R., U. Holmback, and E. Van Cauter, *Circadian misalignment augments markers of insulin resistance and inflammation, independently of sleep loss*. Diabetes, 2014. **63**(6): p. 1860-9.

29. Arble, D.M., et al., *Circadian disruption and metabolic disease: findings from animal models*. Best Pract Res Clin Endocrinol Metab, 2010. **24**(5): p. 785-800.
30. Stokkan, K.A., et al., *Entrainment of the circadian clock in the liver by feeding*. Science, 2001. **291**(5503): p. 490-3.
31. Mukherji, A., A. Kobiita, and P. Chambon, *Shifting the feeding of mice to the rest phase creates metabolic alterations, which, on their own, shift the peripheral circadian clocks by 12 hours*. Proc Natl Acad Sci U S A, 2015. **112**(48): p. E6683-90.
32. Yoon, J.A., et al., *Meal time shift disturbs circadian rhythmicity along with metabolic and behavioral alterations in mice*. PLoS One, 2012. **7**(8): p. e44053.
33. Fonken, L.K., et al., *Light at night increases body mass by shifting the time of food intake*. Proc Natl Acad Sci U S A, 2010. **107**(43): p. 18664-9.
34. Dyar, K.A., et al., *Atlas of Circadian Metabolism Reveals System-wide Coordination and Communication between Clocks*. Cell, 2018. **174**(6): p. 1571-1585.e11.
35. Niu, B., et al., *SIRT1 upregulation protects against liver injury induced by a HFD through inhibiting CD36 and the NFkappaB pathway in mouse kupffer cells*. Mol Med Rep, 2018. **18**(2): p. 1609-1615.
36. Gao, Z., et al., *Sirtuin 1 (SIRT1) protein degradation in response to persistent c-Jun N-terminal kinase 1 (JNK1) activation contributes to hepatic steatosis in obesity*. J Biol Chem, 2011. **286**(25): p. 22227-34.
37. Caron, A.Z., et al., *The SIRT1 deacetylase protects mice against the symptoms of metabolic syndrome*. FASEB J, 2014. **28**(3): p. 1306-16.
38. Asher, G., et al., *SIRT1 Regulates Circadian Clock Gene Expression through PER2 Deacetylation*. Cell, 2008. **134**(2): p. 317-328.
39. Nakahata, Y., et al., *The NAD⁺-dependent deacetylase SIRT1 modulates CLOCK-mediated chromatin remodeling and circadian control*. Cell, 2008. **134**(2): p. 329-40.
40. Nakahata, Y., et al., *Circadian control of the NAD⁺ salvage pathway by CLOCK-SIRT1*. Science, 2009. **324**(5927): p. 654-7.
41. Ramsey, K.M., et al., *Circadian clock feedback cycle through NAMPT-mediated NAD⁺ biosynthesis*. Science, 2009. **324**(5927): p. 651-654.
42. Peek, C.B., et al., *Circadian clock NAD⁺ cycle drives mitochondrial oxidative metabolism in mice*. Science, 2013. **342**(6158): p. 1243417-1243417.
43. Aguilar-Arnal, L., et al., *NAD(+)-SIRT1 control of H3K4 trimethylation through circadian deacetylation of MLL1*. Nat Struct Mol Biol, 2015. **22**(4): p. 312-8.
44. Masri, S., et al., *Partitioning Circadian Transcription by SIRT6 Leads to Segregated Control of Cellular Metabolism*. Cell, 2014. **158**(3): p. 659-672.
45. Levine, D.C., et al., *NAD⁺ Controls Circadian Reprogramming through PER2 Nuclear Translocation to Counter Aging*. Molecular Cell, 2020. **78**(5): p. 835-849.e7.
46. Sato, S., et al., *Circadian Reprogramming in the Liver Identifies Metabolic Pathways of Aging*. Cell, 2017. **170**(4): p. 664-677.e11.
47. Zheng, M., et al., *Nicotinamide reduces renal interstitial fibrosis by suppressing tubular injury and inflammation*. J Cell Mol Med, 2019. **23**(6): p. 3995-4004.
48. Zhen, X., et al., *Nicotinamide Supplementation Attenuates Renal Interstitial Fibrosis via Boosting the Activity of Sirtuins*. Kidney Diseases, 2021. **7**(3): p. 186-199.

49. Trammell, S.A. and C. Brenner, *Targeted, LCMS-based Metabolomics for Quantitative Measurement of NAD(+) Metabolites*. Comput Struct Biotechnol J, 2013. **4**: p. e201301012.
50. Trammell, S.A.J., et al., *Nicotinamide Riboside Opposes Type 2 Diabetes and Neuropathy in Mice*. Scientific Reports, 2016. **6**(1): p. 26933.
51. Rey, G., et al., *The Pentose Phosphate Pathway Regulates the Circadian Clock*. Cell Metab, 2016. **24**(3): p. 462-473.
52. Belenky, P., K.L. Bogan, and C. Brenner, *NAD+ metabolism in health and disease*. Trends in Biochemical Sciences, 2007. **32**(1): p. 12-19.
53. Bogan, K.L. and C. Brenner, *Nicotinic acid, nicotinamide, and nicotinamide riboside: a molecular evaluation of NAD+ precursor vitamins in human nutrition*. Annu Rev Nutr, 2008. **28**: p. 115-30.
54. Clement, J., et al., *The Plasma NAD(+) Metabolome Is Dysregulated in "Normal" Aging*. Rejuvenation Res, 2019. **22**(2): p. 121-130.
55. O'Reilly, T. and D.F. Niven, *Levels of nicotinamide adenine dinucleotide in extracellular body fluids of pigs may be growth-limiting for Actinobacillus pleuropneumoniae and Haemophilus parasuis*. Can J Vet Res, 2003. **67**(3): p. 229-31.
56. Billington, R.A., et al., *Characterization of NAD uptake in mammalian cells*. J Biol Chem, 2008. **283**(10): p. 6367-74.
57. Felici, R., et al., *Insight into molecular and functional properties of NMNAT3 reveals new hints of NAD homeostasis within human mitochondria*. PLoS One, 2013. **8**(10): p. e76938.
58. Kory, N., et al., *MCART1/SLC25A51 is required for mitochondrial NAD transport*. Science Advances, 2020. **6**(43): p. eabe5310.
59. Luongo, T.S., et al., *SLC25A51 is a mammalian mitochondrial NAD+ transporter*. Nature, 2020.
60. Sato, T. and C.M. Greco, *Expanding the link between circadian rhythms and redox metabolism of epigenetic control*. Free Radic Biol Med, 2021. **170**: p. 50-58.

REVIEWER COMMENTS

Reviewer #1 (Remarks to the Author):

Authors did a great job to convincingly answer reviewers' queries. Nevertheless, there is still one question that need to be answer, particularly for the impact of NAD on the circadian clock. While authors provide in this revised article convincing evidence of the impact of the timing of NAD injection on the rhythmic NAD concentration in the liver, they did not consider the hypothesis that this change of NAD redox state can directly impact BMAL1 DNA binding (PMID: 11441146). An analysis of rhythmic DNA binding of BMAL1 on several bona fide BMAL1 target genes will provide important insight into the mechanism by which timed NAD injection can rewire the circadian clock. One another minor issue is the utilization of the word "circadian" on many instances while all experiments have been performed on a light/dark cycle. The words "rhythmic" or "daily rhythms" are more appropriate in that case.

Reviewer #4 (Remarks to the Author):

Escalante-Covarrubias et al.

"Time-of-day defines the efficacy of NAD⁺ to treat diet-induced metabolic disease by adjusting oscillations of the hepatic circadian clock"

Although the authors addressed Reviewers' criticisms and concerns pretty well, the mechanism(s) by which NAD⁺ supplementation ameliorates HFD-induced changes in transcriptional regulation and metabolic signaling still remain unclear in this revised manuscript. Indeed, whereas it is absolutely correct that NAD⁺-related coenzymes are impacted by NAD⁺ supplementation, it would be difficult to directly connect those changes in coenzymes to changes in transcriptional regulation and metabolic signaling which the authors described. This is exactly the place where NAD⁺-dependent sirtuins play important roles, and many groups, including Paolo Sassone-Corsi, Ueli Schibler, Joe Bass, Shin Imai, and others, have already firmly established this important connection between sirtuins and circadian transcriptional regulation. It is true that there are still many questions for details, but the foundation has already been established with a significant number of publications. In this regard, what Reviewer #3 argued to the authors seems to be biased towards one extreme view, and this reviewer believes that the authors have already answered to this Reviewer's criticisms adequately.

The important point that the authors should think about is NOT which one is more important between NAD⁺-dependent sirtuins and NAD⁺-related coenzymes in general, BUT what exactly can give a reasonable mechanistic explanation to the phenomena (the timed effects of NAD⁺ supplementation) that the authors observed in this study. This important point is still weak in this revised manuscript, and without addressing this critical point, a potential scientific advancement that this study could make would not be enough for the sake of the field.

This reviewer is not suggesting the authors that they should address the involvement of sirtuins or other related coenzymes, but they should figure out, at least in part, how exactly NAD⁺ supplementation brought these effects on HFD-induced obese, diabetic mice.

Reviewer #5 (Remarks to the Author):

This manuscript from Escalante-Covarrubias et al describes a time-of-day efficacy of NAD to treated diet induced metabolic disease. The authors provide an experimental paradigm where mice are fed HFD and then supplemented with NAD at ZT 11 or 23. The authors claim that NAD given at ZT 11 improves metabolic health and this is not seen at ZT 23. Also, the authors observe an inversion of the

hepatic clock with ZT 23 NAD supplementation. Unfortunately, these statements are not supported by the data presented. Below, an overview of the concerns are highlighted:

Body weight drop is very severe in Fig 1B which immediately raises the question regarding food intake. The authors state: "Notably, total food intake was comparable for all high fat fed mice (Figure 5C)." Also, a statement regarding the maintained circadian nature of feeding was made for Fig. 7H and 7I. The major concern is not the rhythmicity of feeding, but the overall difference in total food intake. The statement of the authors that no difference in food intake is seen is not supported by the data presented (Fig 5C and Fig 7H). Indeed, there is a difference in food intake between the HF and HFN/HFN23 groups. The overall relative decrease is evident in HFN and HFN23 groups and this is likely the biggest contributor to the differences in nutrient sensing pathways and transcriptional signature that are seen by the authors. This is a major concern regarding the findings presented and the conclusions the authors reach. This would mean that improved metabolic parameters are simply due to a decrease in high fat diet food intake, and not NAD.

Indirect calorimetry is the true test of changes in food intake and this experiment was not performed. However, it would be clear from this data that indirect calorimetry would confirm a decrease in RER and therefore dampened food intake. Again, this would mean that improved metabolic parameters are simply due to a decrease in HFD intake.

The authors state that "circulating leptin was significantly decreased in NAD+ chrono-treated mice, specifically at ZT12." The authors indicate this is a marker for improved metabolic health, along with circulating insulin levels. Leptin is only decreased at ZT12 and this is because of the NAD injection at ZT 11. There is no improvement overall of leptin, which remains very high. This data does not support the conclusion from the authors regarding improved metabolic health.

Similarly, the authors use ITT and GTT as a readout for improved metabolic health with NAD injection. Given the time of injection at ZT 11, the ITT and GTT performed at ZT 16 is critical. Fig 1G shows slight improvement of GTT at day 10 and no impressive difference in GTT at day 20. After 3 weeks of treatment with NAD, no impressive differences in GTT are seen at the end of the experiment. The same statement applies for the ITT data in Fig 1I. The differences in ITT are marginal between the groups. This does not support the claim of the authors that NAD supplementation for 3 weeks is beneficial for metabolic health.

Data presented in Fig 2A are very striking, but they don't reflect triglyceride levels shown in Fig 2D which illustrates that fats are still very high in HF and HFN groups, actually over 4x higher than control group. This data is contradictory and does not support the conclusion that HFN improves metabolic health parameters. Triglyceride levels are still incredibly high, and if they are being broken down, you would see decrease of TGs and increase free fatty acids. The authors claim that in Fig 6 that fats are being oxidized which leads to weight loss. This is not supported by Triglyceride levels presented in Fig 2D. The authors are referring to small changes in TG levels between HF and HFN in Fig 2D at ZT 12, but these small fluctuations are not reflective of the overall picture. Overall, Triglycerides remain very high in HF and HFN, which does not support the claim that NAD is improving metabolism.

Insulin levels are shown in Fig 1D where control diet and HFN are remarkably similar. The authors claim that HFN mice have a re-sensitization of insulin signaling (AKT phosphorylation) and nutrient sensing (AMPK). If insulin levels are exactly the same in CD and HFN (Fig 1D), why are the activated insulin signaling pathways not similar between CD and HFN in Fig 4F and 4G? This is contradictory data and very unclear. What is even more surprising is Fig 5D where insulin is presented as day/night data. Insulin is rhythmic in CD, which is expected. However, there is no rhythmicity in HFN group at all. This data does not support the hypothesis of the authors that HFN is improving metabolic health.

The most concerning data is presented in Figure 6. Metabolic gene expression presented in Fig 6E/F/G does not support the hypothesis that NAD supplementation improves metabolic markers, especially

the gene expression profiles presented here. For example, Cpt1, Acot2, etc are all highly expressed and metabolic gene expression of HFN and HFN23 all resemble gene expression profiles similar to HF. The authors state that increasing NAD does the following: "favoring a transcriptional program of genes involved in fatty acid oxidation, which contribute to weight loss and decreased hepatic and circulating triglycerides in these mice." This claim is not correct. The gene expression of fatty acid oxidation genes is high in all HF fed groups, with or without NAD. Therefore, the argument of the authors that FA oxidation contributes to weight loss cannot be true as the HF group gains weight.

The data presented in Figure 7 is concerning and unclear. How does NAD invert the circadian cycle? Inversion of the circadian cycle in the liver would cause serious problems and therefore would have a negative impact on liver physiology. This data, if true, does not fit with the argument that HFN23 has no major impact. It would be expected that HFN23 mice would also have inverted or disrupted metabolic gene expression profiles. Again, this would argue that HFN23 is detrimental to metabolic health and this is counter to the argument of the authors.

The authors also claim that HFN is beneficial while HFN23 is not. The ORO staining does not support this claim in Fig 5I and S5G, ORO looks exactly the same between the 2 groups. Therefore, how is HFN only beneficial? This data does not support the claim that time-dependent NAD supplementation is beneficial.

POINT BY RESPONSES TO REVIEWERS

Reviewer #1 (Remarks to the Author):

R#1: *Authors did a great job to convincingly answer reviewers' queries. Nevertheless, there is still one question that need to be answer, particularly for the impact of NAD on the circadian clock. While authors provide in this revised article convincing evidence of the impact of the timing of NAD injection on the rhythmic NAD concentration in the liver, they did not consider the hypothesis that this change of NAD redox state can directly impact BMAL1 DNA binding (PMID: 11441146). An analysis of rhythmic DNA binding of BMAL1 on several bona fide BMAL1 target genes will provide important insight into the mechanism by which timed NAD injection can rewire the circadian clock.*

A: We thank this Reviewer for these positive comments and constructive observations, which substantially contributed to improve our manuscript. We agree on the proposed hypothesis, consisting of differential BMAL1 binding to the genome shaped by NAD⁺ oscillations. Hence, to approach this hypothesis, we have now performed chromatin immunoprecipitation (ChIP)

experiments using BMAL1 antibodies to investigate its recruitment to know genomic targets, as follows:

Added to lines 432-461:

“Because redox rhythms regulate DNA binding of CLOCK:BMAL1 heterodimers *in vitro*⁸⁹, and the NAD⁺ precursor NR increases BMAL1 recruitment to chromatin in livers from aged mice⁴⁶, we hypothesized that inverted expression of clock genes in HFN23 might be driven by time-specific recruitment of BMAL1 to chromatin. To test this, we performed ChIP analyses to measure BMAL1 binding to regulatory E-boxes of clock and clock-controlled genes (Figures 7E and 7F). As described⁹⁰, we observed increased recruitment of BMAL1 at ZT6 in livers from CD, HF and HFN groups of mice for all tested E-boxes. Notably, in livers from HFN23 mice, BMAL1 binding appeared significantly increased at ZT18, consistent with inverted expression (Figures 7E, 7F; P<0.05, Two-way ANOVA with Tukey’s post-test). A non-related region at the 3’ UTR region of *Dbp* gene was used as a negative control. We further evaluated the effect of NAD⁺ supplementation in the expression of NAD⁺ biosynthesis and salvage genes *Nmrk1*, *Nampt*, *Nmnat3* and *Nadk* which also showed inverted phase specifically in HFN23 mice (Figure 7G). Accordingly, BMAL1 binding to their regulatory elements was increased at ZT18 in HFN23 mice, yet specific to these group of genes was that NAD⁺ treatment significantly potentiated BMAL1 recruitment to chromatin. Finally, we explored the expression from TFs collaborating with the clock machinery to sustain a rhythmic transcriptional reprogramming in obesity^{16,84}: *Pparg2*, *Ppara* and *Srebf1c*. Transcription for these genes was phase-inverted specifically in HFN23 mice, which was also accompanied by differential BMAL1 chromatin recruitment (Figure 7H). Also, expression levels of additional TFs related to hepatic lipid metabolism *Hnf4a*, *Foxa2*, *Foxo1* and *Cebpa* were altered to a similar extent (Figure S7B). Antiphase expression of key transcription factors regulating hepatic lipid metabolism might underlie the inverted pattern of circulating triglycerides in HFN23 mice (Figure 5H), but other lipids synthesized in the liver might be affected. Accordingly, hepatic cholesterol levels also showed a phase inverted pattern in the liver of HFN23 mice (Figure S7C, S7D), reinforcing the idea that NAD⁺-mediated synchronization of transcriptional rhythms in the liver inverts hepatic lipid metabolism. Together, this data demonstrates a time-dependent transcriptional response to NAD⁺ therapy in the liver of obese mice, through the synchronization of BMAL1 recruitment to chromatin and rhythmic transcription of clock and clock-controlled genes. Hereby, BMAL1 plays a pivotal role translating fluctuations in NAD⁺ levels to shape circadian transcription.”

Added to Figure 7E, 7F, 7G and 7H:

Added to Figure S7B, S7C, S7D:

Added to lines 558-570 (discussion):

“Untimed NAD⁺ rise, trough resetting the circadian machinery and the subsequent misalignment from feeding rhythms, might hinder the coordinated action between the clock and cooperative

transcriptional regulators on chromatin, hereby obstructing the adequate control of specific transcriptional programs. At this regard, BMAL1 recruitment to chromatin was adjusted by timed NAD⁺ treatment, and when administered at ZT23 led to phase-inverted transcription of direct CLOCK:BMAL1 targets, as expected for a pioneer-like transcription factor^{123,124}. In this scenario, we found that several master regulators of rhythmic hepatic lipid and cholesterol metabolism including *Pparα*, *Pparg*, *Srebp1c*, *Cebpa*, or *Hnf4a*^{16,84,125,126} were subjected to this mechanism, and their phase inversion in HFN23 mice was accompanied by inverted rhythms in hepatic cholesterol and circulating triglycerides. These results demonstrate that NAD⁺ modulates BMAL1 recruitment to chromatin and shapes rhythmic transcription and metabolism.”

Added to the Methods section:

“Chromatin immunoprecipitation (ChIP)

100-200 mg of liver tissue were homogenized with a pestle in PBS. Dual crosslinking was performed in a final volume of 1ml using 2 mM of DSG (Disuccinimidyl glutarate, ProteoChem, CAS: 79642-50-5) for 10 min at RT on a rotary shaker. DSG was washed out and a second crosslink was performed using 1% formaldehyde (Sigma-Aldrich, F8775) in PBS for 15 min at RT on a rotary shaker. Crosslinking was stopped with 0.125 M glycine for 5 min at 4°C. After two washes with ice-cold PBS, nuclei were isolated by resuspending the tissue in 600 µL of ice-cold nuclei preparation buffer (NPB: 10 mM HEPES, 10 mM KCl, 1.5 mM MgCl₂, 250 mM sucrose, 0.1% IGEPAL CA-630) and incubating at 4°C for 5 min in rotation. Nuclei were collected by centrifugation at 1,500g for 12 min at 4°C and, and resuspended in 600 µL of cold nuclear lysis buffer (10 mM Tris pH 8, 1 mM EDTA, 0.5 mM EGTA, 0.3% SDS, 1X cComplete™ Protease Inhibitor Cocktail, Roche) for 30 min on ice. Nuclear lysates were stored at -80°C. 300 µL of lysates were sonicated using a Bioruptor Pico Sonicator (Diagenode) for 15 cycles (30 s ON/30 s OFF). Chromatin fragments (100-500 bp) were evaluated on agarose gels using 10 µL of sonicated chromatin for DNA purification using the phenol method. 600 µL of ice-cold ChIP-dilution buffer (1% Triton X-100, 2 mM EDTA, 20 mM Tris pH 8, 150 mM NaCl, 1 mM PMSF, 1X cComplete™ Protease Inhibitor Cocktail, Roche) was added to the fragmented chromatin, and 10% volume was recovered as the Input. Immunoprecipitation was set up overnight at 4°C, by adding 20 µL of magnetic beads (Magna ChIP Protein G Magnetic Beads C #16-662, Sigma-Aldrich) and a combination of two anti BMAL1 antibodies: 1.25 µL rabbit anti-BMAL1 (ab3350, Abcam) and 2.5 µL rabbit anti-BMAL1 (ab93806, Abcam). Immunoprecipitations with 4 µL of

normal mouse IgG (Sigma-Aldrich, Cat. No. 18765) were performed simultaneously. Sequential washes of the magnetic beads were performed for 10 min at 4°C, as follows: Wash buffer 1 (20 mM Tris pH 8, 0.1% SDS, 1% Triton X-100, 150 mM NaCl, 2 mM EDTA), Wash buffer 2 (20 mM Tris pH 8, 0.1% SDS, 1% Triton X-100, 500 mM NaCl, 2 mM EDTA), Wash buffer 3 (10 mM Tris pH 8, 250 mM LiCl, 1% IGEPAL CA-630, 1% sodium deoxycolate) and TE buffer (10 mM Tris pH 8, 1 mM EDTA). Chromatin was eluted by adding 400 µL of fresh elution buffer (10 mM Tris pH 8, 0.5% SDS, 300 mM NaCl, 5 mM EDTA, 0.05 mg/mL proteinase K) to the magnetic beads and incubating overnight at 65°C. A treatment with RNase A at 0.1 mg/ml for 30 min at 37 °C was performed. The DNA was purified from the IPs and Inputs by adding one volume of phenol:chloroform:isoamyl alcohol (25:24:1). After mixing and centrifugation, the aqueous phase was recovered, and DNA was precipitated by adding 1/10 volumes of sodium acetate (0.3 M pH 5.2), 20 µg of glycogen (10901393001, Roche) and 2 volumes of ice-cold ethanol, at -80°C overnight. DNA was pelleted by centrifugation at 13500 rpm for 30 min at 4°C. The DNA was washed with 70% ethanol, and resuspended in 50 µL of molecular grade water. 1.5 µl were used for subsequent qRT-PCR reactions with specific primers designed using Primer3web, within regulatory regions previously identified as BMAL1 binding sites in mouse liver, as reported in the ChIP-Atlas database¹⁴⁶. Primer sequences are available in Supplementary Table 4.”

R#1: *One another minor issue is the utilization of the word “circadian” on many instances while all experiments have been performed on a light/dark cycle. The words “rhythmic” or “daily rhythms” are more appropriate in that case.*

A: This point is well taken, we have now corrected the term “circadian” when this expression was used to indicate daily rhythms mostly across the results section, but also within the discussion.

We thank Reviewer #1 for the provided thorough comments, which have critically contributed to improve the quality and reach of our research in this manuscript.

Reviewer #4 (Remarks to the Author):

Escalante-Covarrubias et al.

"Time-of-day defines the efficacy of NAD⁺ to treat diet-induced metabolic disease by adjusting oscillations of the hepatic circadian clock"

R#4: *Although the authors addressed Reviewers' criticisms and concerns pretty well, the mechanism(s) by which NAD⁺ supplementation ameliorates HFD-induced changes in transcriptional regulation and metabolic signaling still remain unclear in this revised manuscript. Indeed, whereas it is absolutely correct that NAD⁺-related coenzymes are impacted by NAD⁺ supplementation, it would be difficult to directly connect those changes in coenzymes to changes in transcriptional regulation and metabolic signaling which the authors described. This is exactly the place where NAD⁺-dependent sirtuins play important roles, and many groups, including Paolo Sassone-Corsi, Ueli Schibler, Joe Bass, Shin Imai, and others, have already firmly established this important connection between sirtuins and circadian transcriptional regulation. It is true that there are still many questions for details, but the foundation has already been established with a significant number of publications. In this regard, what Reviewer #3 argued to the authors seems to be biased towards one extreme view, and this reviewer believes that the authors have already answered to this Reviewer's criticisms adequately.*

A: We are pleased that Reviewer #4 considers that we adequately addressed Reviewer #3 criticism, and thank Reviewer #4 for these positive comments. During the previous round of revisions, a mechanism was not requested by the reviewers, and this new request is addressed in this revised version of the manuscript, as described below.

R#4: *The important point that the authors should think about is NOT which one is more important between NAD⁺-dependent sirtuins and NAD⁺-related coenzymes in general, BUT what exactly can give a reasonable mechanistic explanation to the phenomena (the timed effects of NAD⁺ supplementation) that the authors observed in this study. This important point is still weak in this revised manuscript, and without addressing this critical point, a potential scientific advancement that this study could make would not be enough for the sake of the field.*

This reviewer is not suggesting the authors that they should address the involvement of sirtuins or other related coenzymes, but they should figure out, at least in part, how exactly NAD⁺ supplementation brought these effects on HFD-induced obese, diabetic mice.

A: We thank this Reviewer for these interesting comments. As noticed, our manuscript is advancing the field in several ways: 1) because most of the research in the field has been performed using NAD⁺ boosters (NMN, NR, NAM), here we demonstrate that NAD⁺ effects are comparable to these other molecules in obese mice. 2) We introduce new pathways involved in the beneficial effects of NAD⁺ supplementation in obese mice, including activation of membrane trafficking and autophagia. 3) Importantly, we demonstrate for the first time a strong time-of-day dependent effect of NAD⁺ supplementation, where the therapy is more effective when supplemented at the onset of the active phase. 4) We determine for the first time that NAD⁺ is a strong synchronizer of the hepatic clock, in a way that while the SCN clock and behavioral rhythms are largely resilient to the therapy, the hepatic molecular clock becomes aligned to the time of NAD⁺ supplementation. This mechanism itself has not been described before and indeed it can challenge the view on how NAD⁺-related therapies are understood and designed. We think this is important as it can raise awareness on potential detrimental effects of NAD⁺ therapies when these are not adequately planned around the circadian cycle.

We agree that there are molecular mechanisms responsible for all these effects, which ultimately converge on improved metabolic parameters after NAD⁺ supplementation particularly at the onset of the active phase. Indeed, most of these molecular mechanisms have been previously described; some of them are also explored in our manuscript: rewiring of mTOR pathway, activation of AMPK and insulin signaling, silencing of inflammatory pathways in the liver. Other mechanisms have been extensively explored in the literature, such as sirtuins activation as this reviewer pointed out, or even stimulation of lipid oxidative pathways; hereby we think that confirming these would not add novel or crucial insights to the manuscript. Hence, we decided to focus on disentangling distinctive transcription-related mechanisms responsible for the effects of rising hepatic NAD⁺ either at ZT12 or ZT24. To approach this question, we reasoned that the cause for inverted transcription may be driven by distinct binding of the CLOCK:BMAL1 pioneer transcription factor to chromatin[1]. To test this hypothesis, we performed ChIP-qPCR to explore BMAL1 recruitment to regulatory elements of selected genes. We describe these data in the new version of the manuscript, as follows:

Added to lines 432-461:

“Because redox rhythms regulate DNA binding of CLOCK:BMAL1 heterodimers *in vitro*⁸⁹, and the NAD⁺ precursor NR increases BMAL1 recruitment to chromatin in livers from aged mice⁴⁶, we hypothesized that inverted expression of clock genes in HFN23 might be driven by time-specific recruitment of BMAL1 to chromatin. To test this, we performed ChIP analyses to

measure BMAL1 binding to regulatory E-boxes of clock and clock-controlled genes (Figures 7E and 7F). As described⁹⁰, we observed increased recruitment of BMAL1 at ZT6 in livers from CD, HF and HFN groups of mice for all tested E-boxes. Notably, in livers from HFN23 mice BMAL1 binding appeared significantly increased at ZT18, consistent with inverted expression (Figures 7E, 7F; $P < 0.05$, Two-way ANOVA with Tukey's post-test). A non-related region at the 3' UTR region of *Dbp* gene was used as a negative control. We further evaluated the effect of NAD⁺ supplementation in the expression of NAD⁺ biosynthesis and salvage genes *Nmrk1*, *Nampt*, *Nmnat3* and *Nadk* which also showed inverted phase specifically in HFN23 mice (Figure 7G). Accordingly, BMAL1 binding to their regulatory elements was increased at ZT18 in HFN23 mice, yet specific to these group of genes was that NAD⁺ treatment significantly potentiated BMAL1 recruitment to chromatin. Finally, we explored the expression from TFs collaborating with the clock machinery to sustain a rhythmic transcriptional reprogramming in obesity^{16,84}: *Pparg2*, *Ppara* and *Srebf1c*. Transcription for these genes was phase-inverted specifically in HFN23 mice, which was also accompanied by differential BMAL1 chromatin recruitment (Figure 7H). Also, expression levels of additional TFs related to hepatic lipid metabolism *Hnf4a*, *Foxa2*, *Foxo1* and *Cebpa* were altered to a similar extent (Figure S7B). Antiphase expression of key transcription factors regulating hepatic lipid metabolism might underlie the inverted pattern of circulating triglycerides in HFN23 mice (Figure 5H), but other lipids synthesized in the liver might be affected. Accordingly, hepatic cholesterol levels also showed a phase inverted pattern in the liver of HFN23 mice (Figure S7C, S7D), reinforcing the idea that NAD⁺-mediated synchronization of transcriptional rhythms in the liver inverts hepatic lipid metabolism. Together, this data demonstrates a time-dependent transcriptional response to NAD⁺ therapy in the liver of obese mice, through the synchronization of BMAL1 recruitment to chromatin and rhythmic transcription of clock and clock-controlled genes. Hereby, BMAL1 plays a pivotal role translating fluctuations in NAD⁺ levels to shape circadian transcription."

Added to Figure 7E, 7F, 7G and 7H:

Added to Figure S7B, S7C, S7D:

Added to lines 558-570 (discussion):

“Untimed NAD⁺ rise, through resetting the circadian machinery and the subsequent misalignment from feeding rhythms, might hinder the coordinated action between the clock and cooperative transcriptional regulators on chromatin, hereby obstructing the adequate control of specific transcriptional programs. At this regard, BMAL1 recruitment to chromatin was adjusted by timed NAD⁺ treatment, and when administered at ZT23 led to phase-inverted transcription of direct CLOCK:BMAL1 targets, as expected for a pioneer-like transcription factor^{123,124}. In this scenario, we found that several master regulators of rhythmic hepatic lipid and cholesterol metabolism including *Pparα*, *Pparg*, *Srebp1c*, *Cebpa*, or *Hnf4a*^{16,84,125,126} were subjected to this mechanism, and their phase inversion in HFN23 mice was accompanied by inverted rhythms in hepatic cholesterol and circulating triglycerides. These results demonstrate that NAD⁺ modulates BMAL1 recruitment to chromatin and shapes rhythmic transcription and metabolism.”

Added to the Methods section:

“Chromatin immunoprecipitation (ChIP)

100-200 mg of liver tissue were homogenized with a pestle in PBS. Dual crosslinking was performed in a final volume of 1ml using 2 mM of DSG (Disuccinimidyl glutarate, ProteoChem, CAS: 79642-50-5) for 10 min at RT on a rotary shaker. DSG was washed out and a second crosslink was performed using 1% formaldehyde (Sigma-Aldrich, F8775) in PBS for 15 min at RT on a rotary shaker. Crosslinking was stopped with 0.125 M glycine for 5 min at 4°C. After two washes with ice-cold PBS, nuclei were isolated by resuspending the tissue in 600 µL of ice-cold nuclei preparation buffer (NPB: 10 mM HEPES, 10 mM KCl, 1.5 mM MgCl₂, 250 mM sucrose, 0.1% IGEPAL CA-630) and incubating at 4°C for 5 min in rotation. Nuclei were collected by centrifugation at 1,500g for 12 min at 4°C and, and resuspended in 600 µL of cold nuclear lysis buffer (10 mM Tris pH 8, 1 mM EDTA, 0.5 mM EGTA, 0.3% SDS, 1X cOmplete™ Protease Inhibitor Cocktail, Roche) for 30 min on ice. Nuclear lysates were stored at -80°C. 300 µL of lysates were sonicated using a Bioruptor Pico Sonicator (Diagenode) for 15 cycles (30 s ON/30 s OFF). Chromatin fragments (100-500 bp) were evaluated on agarose gels using 10 µL of sonicated chromatin for DNA purification using the phenol method. 600 µL of ice-cold ChIP-dilution buffer (1% Triton X-100, 2 mM EDTA, 20 mM Tris pH 8, 150 mM NaCl, 1 mM PMSF, 1X cOmplete™ Protease Inhibitor Cocktail, Roche) was added to the fragmented chromatin, and 10% volume was recovered as the Input. Immunoprecipitation was set up overnight at 4°C, by

adding 20 μ L of magnetic beads (Magna ChIP Protein G Magnetic Beads C #16-662, Sigma-Aldrich) and a combination of two anti BMAL1 antibodies: 1.25 μ L rabbit anti-BMAL1 (ab3350, Abcam) and 2.5 μ L rabbit anti-BMAL1 (ab93806, Abcam). Immunoprecipitations with 4 μ L of normal mouse IgG (Sigma-Aldrich, Cat. No. 18765) were performed simultaneously. Sequential washes of the magnetic beads were performed for 10 min at 4°C, as follows: Wash buffer 1 (20 mM Tris pH 8, 0.1% SDS, 1% Triton X-100, 150 mM NaCl, 2 mM EDTA), Wash buffer 2 (20 mM Tris pH 8, 0.1% SDS, 1% Triton X-100, 500 mM NaCl, 2 mM EDTA), Wash buffer 3 (10 mM Tris pH 8, 250 mM LiCl, 1% IGEPAL CA-630, 1% sodium deoxycolate) and TE buffer (10 mM Tris pH 8, 1 mM EDTA). Chromatin was eluted by adding 400 μ L of fresh elution buffer (10 mM Tris pH 8, 0.5% SDS, 300 mM NaCl, 5 mM EDTA, 0.05 mg/mL proteinase K) to the magnetic beads and incubating overnight at 65°C. A treatment with RNase A at 0.1 mg/ml for 30 min at 37 °C was performed. The DNA was purified from the IPs and Inputs by adding one volume of phenol:chloroform:isoamyl alcohol (25:24:1). After mixing and centrifugation, the aqueous phase was recovered, and DNA was precipitated by adding 1/10 volumes of sodium acetate (0.3 M pH 5.2), 20 μ g of glycogen (10901393001, Roche) and 2 volumes of ice-cold ethanol, at -80°C overnight. DNA was pelleted by centrifugation at 13500 rpm for 30 min at 4°C. The DNA was washed with 70% ethanol, and resuspended in 50 μ L of molecular grade water. 1.5 μ l were used for subsequent qRT-PCR reactions with specific primers designed using Primer3web, within regulatory regions previously identified as BMAL1 binding sites in mouse liver, as reported in the ChIP-Atlas database¹⁴⁶. Primer sequences are available in Supplementary Table 4.”

Of note, modulation of CLOCK:BMAL1 binding in response to redox rhythms was reported *in vitro* [2], as indicated by Reviewer #1, and a recent report from J. Bass laboratory provided further advance on this effect *in vivo*, using the NAD⁺ precursor NR, which showed that supplementation with NR in aged mice increases by at least ~25% the BMAL1 binding density in the genome and argues that under normal conditions, “NAD⁺ drives circadian transcription by stabilizing BMAL1 chromatin binding and facilitating the recruitment of collaborative TFs”[3]. According to this report, this effect relies on previously described molecular interplay between the NAD⁺-dependent deacetylase SIRT1 and its clock protein targets such as PER2[3, 4]. Yet, BMAL1 itself is also a target for SIRT1 deacetylation[5], and furthermore, we previously described in a very detailed manner a molecular mechanism involving the histone methyltransferase MLL1 as a target for SIRT1-mediated deacetylation, which shapes

fluctuations in the circadian epigenome[6]. As indicated, all these molecular mechanisms are influenced by NAD⁺ levels, and it is conceptually apparent that they also contribute to coordinate the transcriptional transitions described in our NAD⁺ treated mice. Yet, they mostly converge in regulating BMAL1:CLOCK function, which is explored as described in this new version of the manuscript.

Added to lines 553-571 (Discussion)

“Our analyses revealed substantial differences in expression from genes involved in fatty acid oxidation, with marked downregulation in obese mice treated with NAD⁺ at ZT23 (Figure 6E-G, S6C). In mouse liver, these genes are oscillatory with a peak of expression at the end of the rest phase⁸⁴. Their expression is to some extent clock-controlled; however, their transcriptional regulation mostly relies on nutritional cues integrated by intracellular signaling, multiple nuclear receptors and transcription factors such as PPAR_γ, PPAR_α or SREBP1, epigenetic regulators including MLL1 or SIRT1, and even neural circuits^{16,84,120-122}. Untimed NAD⁺ rise, through resetting the circadian machinery and the subsequent misalignment from feeding rhythms, might hinder the coordinated action between the clock and cooperative transcriptional regulators on chromatin, hereby obstructing the adequate control of specific transcriptional programs. At this regard, BMAL1 recruitment to chromatin was adjusted by timed NAD⁺ treatment, and when administered at ZT23 led to phase-inverted transcription of direct CLOCK:BMAL1 targets, as expected for a pioneer-like transcription factor^{123,124}. In this scenario, we found that several master regulators of rhythmic hepatic lipid and cholesterol metabolism including *Ppara*, *Pparγ*, *Srebp1c*, *Cebpa*, or *Hnf4a*^{16,84,125,126} were subjected to this mechanism, and their phase inversion in HFN23 mice was accompanied by inverted rhythms in hepatic cholesterol and circulating triglycerides. These results demonstrate that NAD⁺ modulates BMAL1 recruitment to chromatin and shapes rhythmic transcription and metabolism.”

We hope that these new mechanistic insights contribute to satisfy this Reviewer’s concerns.

Reviewer #5 (Remarks to the Author):

R#5: *This manuscript from Escalante-Covarrubias et al describes a time-of-day efficacy of NAD to treated diet induced metabolic disease. The authors provide an experimental paradigm where mice are fed HFD and then supplemented with NAD at ZT 11 or 23. The authors claim that NAD given at ZT 11 improves metabolic health and this is not seen at ZT 23. Also, the authors observe an inversion of the hepatic clock with ZT 23 NAD supplementation. Unfortunately, these statements are not supported by the data presented. Below, an overview of the concerns are highlighted:*

A: We appreciate this Reviewer's comments, below we include a point-by-point response to all the raised concerns in full.

R#5: *Body weight drop is very severe in Fig 1B which immediately raises the question regarding food intake. The authors state: "Notably, total food intake was comparable for all high fat fed mice (Figure 5C)." Also, a statement regarding the maintained circadian nature of feeding was made for Fig. 7H and 7I. The major concern is not the rhythmicity of feeding, but the overall difference in total food intake. The statement of the authors that no difference in food intake is seen is not supported by the data presented (Fig 5C and Fig 7H). Indeed, there is a difference in food intake between the HF and HFN/HFN23 groups. The overall relative decrease is evident in HFN and HFN23 groups and this is likely the biggest contributor to the differences in nutrient sensing pathways and transcriptional signature that are seen by the authors. This is a major concern regarding the findings presented and the conclusions the authors reach. This would mean that improved metabolic parameters are simply due to a decrease in high fat diet food intake, and not NAD.*

Indirect calorimetry is the true test of changes in food intake and this experiment was not performed. However, it would be clear from this data that indirect calorimetry would confirm a decrease in RER and therefore dampened food intake. Again, this would mean that improved metabolic parameters are simply due to a decrease in HFD intake.

A: We thank Reviewer #5 for these observations, and for pointing out that our results could potentially be understood in this way. We have now carefully reviewed our results and we conclude that the collected data in our manuscript do not support the statement that the phenotypic and metabolic improvements identified after NAD⁺ treatment in obese mice are due

to decreased food intake. Below, we carefully describe the data and reasons behind this conclusion, and how to reliably measure food intake in mice.

First, indirect calorimetry is very useful to measure energy expenditure, using oxygen consumption and carbon dioxide production to calculate it. These systems measure the volume of O₂ (VO₂) consumed and volume of CO₂ (VCO₂) produced by an animal. In this system, the respiratory exchange rate (RER) is the ratio between VCO₂ and VO₂ (VCO₂/VO₂) and is an approach used to estimate the substrate utilization. A value of 0.7 indicates a nearly complete dependence on fatty acid metabolism, while a value of 1.0 indicates primary dependence on carbohydrate metabolism. Hereby, mice fed high fat diets for several weeks show their RER mostly stable across the light/dark cycle in between 0.7-0.8 [7]. In specific animal models where energy balance can be established (i.e. the quantity of expended energy equals the consumed energy), the RER might be a good indicator of energy intake[7]. However, in general there are many physiological and behavioral factors strongly altering the RER, such as *de novo* lipogenesis[8], physical activity[9], stress[10], environmental temperature[11], thermogenesis by brown adipose tissue activation or browning processes [12, 13], and importantly, energy balance. In mouse models for obesity using pharmacological or genetic interventions oriented to weight loss, decreased RER has been consistently observed even with paired feeding[14-26]. Decreased RER in these obesity models is generally understood as greater fat oxidation causing weight loss, but also other physiological processes could contribute such as thermogenesis[12, 27]. Furthermore, time of feeding has a striking impact on the RER in obese mice[28]. Importantly, SIRT1 activation, a direct consequence of NAD⁺ supplementation, significantly decreases RER in mice[29, 30], which also happens in obese mice treated with SIRT1 activators, without reducing food intake[31-33]. Hereby, with all these confounding variables which undoubtedly apply to our model, a decreased RER is expected in NAD⁺ treated obese mice, but would not indicate changes in overall food intake, but rather an increase in energy expenditure through fat oxidation or other processes[34]. With all due respect, we consider that determining the source of potential RER differences between our groups of mice among all these confounding variables would require extensive amount of work and resources and might be the subject for future research.

While this Reviewer clearly conveys concerns about the food intake data, we still consider that the most reliable approach to assess energy intake in our mouse model is directly measuring food consumption. We approached this question through weekly measurements on food intake or every three / four hours around the day. We analyzed the data with two-way ANOVA using

matched values as these are repeated measurements from the same mice every week, and we did not find significant differences between groups ($P < 0.5$). From the comments, we understand that this Reviewer might be concerned about non-significant trends in the data, yet we consider that use of trends to describe differences that have been found to be 'almost' but not quite statistically significant is generally misleading. Non-significant P values supporting trends, by being 'almost significant' are generally considered an error at least in non-clinical research. However, we agree that not all researchers understand *trends* in this way and many times this is a subject of extensive debate. To solve this discrepancy, we have now included additional measurements on food intake, increasing from 6-7 (20 total) to 11 (33 mice total). As shown in fig. 5C, there are no differences, based on the application of Two-way ANOVA with matched values, and a Tukey post-test to analyze differences between groups for each week was also performed, and statistical differences were not detected. We also analyzed the average food intake per mice for 3 weeks before and after the treatment, using data from 37 - 48 mice as follows: HF: 48; HFN: 37; HFN23: 37. Two-way ANOVA with Bonferroni post-test analyses demonstrated that there were no statistical differences in food intake within groups before or after the treatments (Fig. S5C), and as such, any potential difference or trend between groups could be attributed to randomization, but not to the treatment. Furthermore, specifically HFN group of mice showed significant weight loss during the three weeks of treatment, while maintaining their caloric intake, indicating that calories/kg of body weight ratio is increased, supporting that the NAD⁺ treatment is counteracting the detrimental effects of high fat feeding. To give clarity to the food intake data in Figures 5, besides providing the source data, we have introduced these modifications to the new version of the manuscript:

Added to lines 313-315

“Notably, total food intake was comparable for all high-fat fed mice, and before and after the treatment no significant differences were found (Figure 5C, S5C, Two-way ANOVA with post-test)”

Added to Figure 5C:

Added to Figure S5C:

Modified to figure legend, Figure 5C:

“(C) Weekly food intake from the indicated groups of mice is shown during three weeks before and after the NAD⁺ treatment (n=33 mice).”

Added to the figure legend, Figure S5C:

“(C) Measurements from three weeks of food intake before and after the treatment were averaged for n= HF: 48; HFN: 37; HFN23: 27. Two-way ANOVA with Bonferroni post-test was applied, and the data are means ± SD.”

Along the same lines, this Reviewer argues that “*the major concern is not the rhythmicity of feeding*”. Overall, we disagree with this argument, as it has been extensively demonstrated that time of feeding shapes hepatic circadian rhythms, and as such, it is a major contributor to metabolic disease. As suggested by Reviewers #1 and #2 for the first version of this manuscript, these data are critical to understand whether NAD⁺ treatment at ZT23 inverted the hepatic clock through misaligning feeding and light/dark cycles. To address this, experiments on Fig. 7 H-I (or Fig. 8D-E in the new version of the manuscript) were performed. In these experiments, food intake was monitored every three hours, and the time series were plotted in the cartesian plot (Fig. 8D), showing that food intake was not inverted in HFN23 mice. To alleviate this Reviewer’s concerns on this set of data, here we show the average daily energy intake (Kcal., n=10 mice per group) calculated from these measurements:

For these data, differences are also non-significant by one-way ANOVA and Tukey's multiple comparisons test.

Of note, we decided not to include this graph in the new version of the manuscript to avoid redundancies, however, if this Reviewer considers so we can add it.

We find that food intake is no longer a concern, and the new version of the manuscript has now improved with more measurements and data analyses and descriptions, which strengthen the notion that decreased food intake is not a consequence of NAD⁺ treatment. We thank this reviewer for raising this question, as the manuscript is much clearer now.

R#5 *The authors state that “circulating leptin was significantly decreased in NAD⁺ chronotreated mice, specifically at ZT12.” The authors indicate this is a marker for improved metabolic health, along with circulating insulin levels. Leptin is only decreased at ZT12 and this is because of the NAD injection at ZT 11. There is no improvement overall of leptin, which remains very high. This data does not support the conclusion from the authors regarding improved metabolic health.*

A: This point is well taken, and we agree that the data on circulating leptin might be unclear and inconclusive as leptin signaling is not a subject of research in our manuscript. Following Reviewer #5 suggestion, we decided to remove this data from our manuscript and consider exploring the potential variations in leptin signaling and neural control of metabolism by NAD⁺ in future studies.

R#5 *Similarly, the authors use ITT and GTT as a readout for improved metabolic health with NAD injection. Given the time of injection at ZT 11, the ITT and GTT performed at ZT 16 is critical. Fig 1G shows slight improvement of GTT at day 10 and no impressive difference in GTT*

at day 20. After 3 weeks of treatment with NAD, no impressive differences in GTT are seen at the end of the experiment. The same statement applies for the ITT data in Fig 11. The differences in ITT are marginal between the groups. This does not support the claim of the authors that NAD supplementation for 3 weeks is beneficial for metabolic health.

A: This Reviewer indicates that when the NAD⁺ is supplied at ZT11, stronger effects on ITT and GTT should be identified at ZT16 compared to ZT4. Indeed, ZT16 is closer to the injection time than ZT4; however, it has been known for a long time that glucose tolerance and insulin sensitivity are rhythmic in mammals. In mice, highest glucose tolerance occurs around ZT15-17 which coincides with improved insulin sensitivity[35], and conversely, during the rest period these tests perform poorer [35]. Rhythms in glucose homeostasis, which are also observed in humans[36], are to a large extent controlled by the central clock in the SCN, and sustain daily adaptation to feeding rhythms[37]. As such, for all groups of mice, ITT and GTT show better results at ZT16 than at ZT4, and importantly, this is observed even in HF fed, non-treated mice (Fig.1). This is the main reason why differences in GTT and ITT between HF and CD fed mice are more prominent at ZT4, when they naturally have less metabolic flexibility and hereby, glucose or insulin challenges are more prone to exceed their homeostatic capacity. Concomitantly, this supports that the beneficial effects of NAD⁺ treatment on glucose homeostasis are more evident at ZT4, and the reason why we selected ZT4 to investigate GTT and ITT in HFN23 mice, as at ZT16 mild effects could be masked.

This reviewer argues that “*After 3 weeks of treatment with NAD, no impressive differences in GTT are seen at the end of the experiment*” or “*differences in ITT are marginal between the groups*” Overall, we cannot agree with this statement, mainly because improvements in GTT and ITT shown for HFN mice in figure 1 at ZT16 are statistically significant ($P < 0.05$ for the AUC) and largely comparable to CD mice. Furthermore, considering these differences impressive, marginal or not is a subjective criterion that to our opinion cannot be supported with data beyond statistical analyses. It is important to note that most of the reported data in the literature for rodents, GTT and ITT are performed during the light phase, that is, when mice are predisposed to show differences between diets/treatments/condition as explained above. While this is generally due to practical reasons (i.e, there is light in the room, more adequate working hours, restricted access to the animal facility at night etc.), the information related to changes in glucose homeostasis during the dark, active phase remains unknown with this approach. Nevertheless, previous reports using obese mice which are pharmacologically or genetically intervened to rise NAD⁺ levels show GTT/ITT improvements of a magnitude comparable to the

effects shown in our NAD⁺ treated mice. We respectfully invite this Reviewer to please refer to Figure 2A-F in Ref. [38], Figures 3H, 3J in Ref. [39], Figure 1A in Ref. [40], Figure 1G in Ref. [41], Figure 1H, 1I in Ref.[42] , Fig. 1G in Ref.[43] , Figures 1A, 2F, 2G in Ref. [44], amongst others. It can be noted that the magnitude of improvements in GTT and/or ITT after NAD⁺ treatment in our HFN mice is similar and equivalent to those previously reported for comparable mouse models, and significant according to the generally applied statistical tests. For these reasons, we conclude that this data in Figure 1 is sound and supports the idea that NAD⁺ supplementation for three weeks improves glucose homeostasis in obese mice. Additionally, to firmly demonstrate that NAD⁺ treatment performs better when supplied at ZT11 than at ZT23, we included a direct comparison between these two groups of mice in Figure 5, as follows:

Added to lines 322-326:

“We found that at the end of the treatment with NAD⁺ at ZT23 (day 20, HFN23), glucose and insulin tolerance showed non-significant improvement compared to the HF-fed mice. Actually, the NAD⁺ treatment at ZT11 was significantly more favorable to improve glucose homeostasis than at ZT23 (Figures 5E, 5F, S5F, S5G; One-way ANOVA followed by Tukey’s posttest).”

Added to Figure 5E, F:

R#5 *Data presented in Fig 2A are very striking, but they don't reflect triglyceride levels shown in Fig 2D which illustrates that fats are still very high in HF and HFN groups, actually over 4x higher than control group. This data is contradictory and does not support the conclusion that HFN improves metabolic health parameters. Triglyceride levels are still incredibly high, and if they are being broken down, you would see decrease of TGs and increase free fatty acids. The authors claim that in Fig 6 that fats are being oxidized which leads to weight loss. This is not supported by Triglyceride levels presented in Fig 2D. The authors are referring to small changes in TG levels between HF and HFN in Fig 2D at ZT 12, but these small fluctuations are not reflective of the overall picture. Overall, Triglycerides remain very high in HF and HFN, which does not support the claim that NAD is improving metabolism.*

A: We thank Reviewer #5 for these observations, drawing attention that our results could potentially be understood in this way. Certainly, we have reasons to interpret results in Fig 2A-C and Fig 2D in a different way, as follows:

- 1) ORO stains neutral lipids, a group of lipids including cholesteryl ester, triglycerides, fatty acids and wax esters. In the HFN group, we observed a dramatic reduction in the ORO stain, but we cannot discern which kind of lipids are being impacted. Opposite, in Fig. 2D, measurements are specific for TGs.
- 2) Technically, ORO stain is a qualitative or semi-quantitative approach. Indeed, our data clearly indicates that fatty liver is qualitatively corrected by NAD⁺ therapy (Fig. 2A), and this is further confirmed by semiquantitative measurements on the intensity and the length of lipid droplets (Fig 2B, 2C). Particularly, we chose to analyze triglycerides through a quantitative colorimetric assay, because they constitute the major form of storage and transport of fatty acids within cells. These quantitative measurements show that overall, triglycerides are significantly reduced in HFN group of mice compared to their HF littermates, demonstrating that NAD⁺ therapy at ZT12 is accompanied by decreased hepatic triglycerides (Fig 2D).
- 3) Regarding hepatic TGs in HFN mice, indeed these are still significantly higher than in the CD mice. However, it cannot be ignored that they are also significantly decreased compared to the HF group. Importantly, both HF and HFN mice are being fed a diet containing 60% Kcal from fat; hence, if diet composition is mostly lipidic, it is not surprising that TGs, the main form of lipid storage, remain high. In these experiments, we consider a relevant/novel/unexpected result that after NAD⁺ treatment, hepatic TGs significantly decrease.

- 4) Other types of lipids might be more impacted by NAD⁺ treatment, resulting in the striking ORO data, which this Reviewer noted. Yet also TGs were significantly decreased. Of note, ORO data was obtained from livers at ZT12, which also show important differences in TGs content.

We hope we can further convince this reviewer by showing the raw data side by side:

ORO stain (AU)				Triglycerides (mg/ml) ZT12		
	CD	HF	HFN	CD	HF	HFN
	0.93	16.02	4.62	8.4102108	58.6857143	30.9714286
	0.7	12.72	7	10.1142857	54.1142857	41.8285714
	1.17	32.86	7.89	9.54285714	59.8285714	42.4010467
	1.13	42.86	4.54	9.82857143	46.4001321	34.9714286
	1.06	14.99	1.84	12.1142857	61.2571429	38.1142857
	1.04	22.31	3.66	9.44615385	78.6769231	26.0615385
	0.91	16.82	7.23	10.3692308	73.0615385	36.9846154
	1.06	21.47	7.19	10.6769231	64.2153846	64.8307692
	1	34.38	4.78	12.2153846	42.6769231	38.3692308
				10.3692308	88.8307692	47.6000193
Mean	1	23.83	5.417	10.31	62.77	40.21
Std. Error of Mean	0.04679	3.485	0.6744	0.3694	4.49	3.332

As shown, differences in ORO stain are not strictly or linearly conserved in the TGs content. For example, there is a 23-fold difference in the ORO stain between CD and HF mice, but only a 6-fold difference in the TGs content between these two groups of mice. Indeed, for the HFN the ratio is better conserved, as there is a 5-fold difference between CD and HFN in the ORO stain, and this difference is of 4-fold in the TG content. As a result, ORO and TGs measurements are not necessarily correlated, as they are completely different techniques. Hence, we don't find grounds to conclude that these two sets of data are contradictory. Furthermore, statistical analyses (One-way ANOVA followed by Tukey's post-test) demonstrate that differences between HF and HFN are very significant (**** $P < 0.0001$) for both sets of data (ORO and TGs). We consider this an important result validating NAD⁺ therapy as an effective treatment for diet-induced fatty liver.

In an effort to prevent confusion, we modified the text as follows:

Added to lines 163-168:

“Histological staining with Oil-Red-O (ORO) was used to semi-quantitatively assess hepatic steatosis (Figure 2A), revealing that obese mice treated with NAD⁺ significantly decreased hepatic neutral lipid content (Figure 2B, 2C, One-way ANOVA, Tukey’s posttest). Furthermore, a quantitative assay specific for hepatic triglycerides, the major form of fatty acids storage, revealed that these were globally reduced in obese mice after restoring hepatic NAD⁺ oscillations (Figure 2D, Two-way ANOVA, Tukey’s posttest).”

R#5 *Insulin levels are shown in Fig 1D where control diet and HFN are remarkably similar. The authors claim that HFN mice have a re-sensitization of insulin signaling (AKT phosphorylation) and nutrient sensing (AMPK). If insulin levels are exactly the same in CD and HFN (Fig 1D), why are the activated insulin signaling pathways not similar between CD and HFN in Fig 4F and 4G? This is contradictory data and very unclear. What is even more surprising is Fig 5D where insulin is presented as day/night data. Insulin is rhythmic in CD, which is expected. However, there is so rhythmicity in HFN group at all. This data does not support the hypothesis of the authors that HFN is improving metabolic health.*

A: We thank Reviewer #5 for these observations, and for pointing out that our results could potentially be understood in this way. We would like to clarify that, from data in Fig 1D, we do not claim that insulin levels are “exactly the same”. Instead, we argue that insulin levels in HFN mice are significantly lower than in HF, and comparable to CD mice. To clarify this, we performed the analysis of circadian rhythmicity as provided by CirWave. This study showed that for all CD, HF and HFN groups of mice, 24-hour rhythms can be detected in circulating insulin levels; however, the acrophases were different, and also amplitudes, as illustrated in the resulting cartesian plots for the time series (Figure S1F). Phase shifts appeared between all tested conditions, yet overall, circulating insulin in HFN mice was comparable to the CD, and significantly lower than in the HF group, as proven by the AUC in fig. 1D, and also shown when a One-way ANOVA with Tukey post-test is performed for all insulin data:

This explanation has been improved in the text, as follows:

Added to lines 136-140:

“We sought to assess physiological indicators of metabolic health and found that circulating insulin levels were much lower in the HFN group when compared to the HF group, with a major effect during the early active phase (Figure 1D, ZT12-18, $P < 0.001$ Two-way ANOVA, Tukey post-test) and a six-hour phase delay in the oscillatory pattern (Figure S1F). Indeed, circulating insulin in HFN mice appeared largely comparable to their control littermates.”

Added to Fig. S1F:

Along these same lines, regarding the graph in Fig. 5D, please note in the figure legend that these data were collected at ZT6 (middle of the light period, day), and ZT18 (middle of the dark phase, night). A closer look to these graphs reveals that indeed, data for these two time points in Fig 1D are largely comparable to 5D. Below, you can find the two graphs compared side by side with the same scale

As shown, the graph on the left corresponds to Fig 5D, while the graph on the right are the corresponding ZTs for data on Fig 1D. We find these two graphs remarkably similar and comparable. This reviewer claims that from the Fig. 5D, “*Insulin is rhythmic in CD, which is expected. However, there is so rhythmicity in HFN group at all*”. Yet, from two data points, it is not possible to assess circadian rhythmicity, while in this particular case, the scale is too high to appreciate differences between the two measurements on the HFN mice. If this reviewer is referring that day-to-night transitions in insulin levels can be identified in Fig. 5D for the CD group but not for the HFN group, it should be noted that this assumption is not accurate. In fact, when analyzing this data using a Two-way ANOVA followed by Bonferroni posttest, no significant differences between day and night are detected for the CD, HF and HFN mice ($P > 0.05$). Instead, insulin measurements from HFN23 mice were significantly higher at ZT18 than at ZT6 ($P < 0.001$). However, circadian oscillations cannot be inferred from two time points, while as previously related, sampling every six hours evidenced daily oscillations in circulating insulin for all tested groups of mice (Fig. S1F). As also suggested by Reviewer#1, in this revised version of the manuscript, we have refrained to use the word “circadian” when rhythmicity was not assessed in the adequate conditions to evidence an endogenous rhythm, or with the minimum data points that require nycthemeral rhythms. Finally, we have changed the format from histograms which were constructed with less than 10 replicates, to illustrate the complete set of data, and provided the source data in this new version of the manuscript. We hope this helps for clarification.

Additionally, this Reviewer finds contradictory “that the activated insulin signaling pathways are not similar between CD and HFN in Fig 4F and 4G”. Indeed, Fig. S4A shows that there is a phase delay in the activation of AKT between CD and HFN. Also, as related, circulating insulin also shows a phase shift and amplitude differences between CD and HFN. Yet, most important is that HFN mice are being treated with a pharmacological intervention raising hepatic NAD^+

levels in obese mice. Notably, increased hepatic NAD⁺ induces AMPK activation through distinct mechanisms, largely via SIRT1-LKB1-AMPK axis[45-47], which in turn is able to activate AKT signaling[48]. Indeed, activation of SIRT1 has been extensively reported to improve hepatic insulin sensitivity which relies on AKT regulation[49, 50]. Additional mechanisms potentially regulate AKT phosphorylation and improve insulin sensitivity, including increased OXPHOS[51] or ROS[52], which are also promoted by increased NAD⁺. Along the same lines, circulating insulin levels are highly dependent on the feeding state, and generally respond to a balance between pancreatic secretion and sensitivity in distinct tissues including liver, muscle, or adipose tissue. We argue that after the NAD⁺ treatment, strict time-correlation between hepatic AKT phosphorylation and circulating insulin is not necessarily expected, because the pharmacological effects of NAD⁺ are intimately linked to insulin sensitization by modulating PI3K/AKT signaling, while daily fluctuations in circulating insulin might be largely influenced by food consumption. While it is true that the HFN mice show low levels of circulating insulin along the day, this has also been reported to enhance insulin sensitivity in mice[53].

R#5 *The most concerning data is presented in Figure 6. Metabolic gene expression presented in Fig 6E/F/G does not support the hypothesis that NAD supplementation improves metabolic markers, especially the gene expression profiles presented here. For example, Cpt1, Acot2, etc are all highly expressed and metabolic gene expression of HFN and HFN23 all resemble gene expression profiles similar to HF. The authors state that increasing NAD does the following: “favoring a transcriptional program of genes involved in fatty acid oxidation, which contribute to weight loss and decreased hepatic and circulating triglycerides in these mice.” This claim is not correct. The gene expression of fatty acid oxidation genes is high in all HF fed groups, with or without NAD. Therefore, the argument of the authors that FA oxidation contributes to weight loss cannot be true as the HF group gains weight.*

A: We thank this reviewer for pointing this out, we apologize for not clearly describing this set of data. What we observe is that in the HFN group, the breadth of transcriptional activity from the genes in figure 6 extends through the active period, and at ZT18, they show significantly higher expression levels than the rest of the groups. Please, note that these are RT-qPCR experiments, and the data is expressed in logarithmic scale to show fold change. This allows to better visualize decreased expression levels between samples, but we acknowledge that Log₂ transformed values might not be intuitive to interpret. To assist with this, we have now included a new Figure S6D, where expression values at ZT18 are shown:

Added to Fig. S6D:

It can be clearly identified that the HF group shows significantly increased expression in 9 out of these 13 genes when compared to the CD; remarkably, expression levels within the HFN group are significantly increased respect to the HF group for all tested genes. Indeed, within these data we find two consistent effects in HFD-fed mice: treatment with NAD⁺ at ZT11 significantly increases the expression of all these lipid oxidation genes, while at ZT23 they are significantly less expressed. Of note, when mice eat high fat diets, genes related to lipid metabolism increase their expression, hence it is not surprising that FAO genes are overexpressed in HF-fed mice compared to the CD group, as extensively reported[54]. Importantly, we found that NAD⁺ treatment significantly alters the expression of these genes depending on the time of treatment. In the context of nocturnal animals such as mice, which eat during the dark period, it is conceptually apparent that overexpressing genes related to lipid catabolic processes at night might drive lipid catabolism during food intake and hereby, contribute to sustained weight loss in HFN mice during the treatment. This is not observed in the HFN23 group of mice, coincident with weight gain. Note that the housekeeping genes *Tbp* and *Rplp* are equally expressed. We have modified the description of these results, as follows:

Added to lines 381-394

“Indeed, fatty acid oxidation-related genes were highly expressed at the end of the rest period (~ZT12); yet, unique to the HFN group was that the breadth of transcriptional activity further extended through the active period, reaching significantly higher levels than in the non-treated, obese mice (HF) at ZT18 (Figures 6E-G, $P < *0,05$, $**0,01$, $***0,001$ Two-way ANOVA with Tukey post-test). When comparing all groups of mice fed with high fat diet, expression of these genes at ZT18 was altered depending on the time of NAD⁺ treatment, in a way that the treatment at ZT11 significantly enhanced their expression, whereas in mice treated at ZT23, expression was significantly reduced to levels largely comparable to the CD littermates (Figure S6C, One way ANOVA with Tukey posttest). Accordingly, housekeeping genes *Tbp* and *Rplp* presented no significant variations (Figure S6D). Together, these data suggest that increased hepatic NAD⁺ levels at the beginning of the active phase induce AMPK-phosphorylation and activity, favoring a transcriptional program of genes involved in fatty acid oxidation which extends through the active phase, possibly contributing to weight loss and decreased hepatic and circulating triglycerides specifically in HFN mice.”

Additionally, expression of FAO genes is regulated to some extent by the clock machinery, but overall, their expression is determined by the integration of a wide array of metabolic and epigenetic signaling. Hence, considering that food intake remains mostly aligned to light dark-cycles, it is not surprising that their expression in HFN23 mice appears substantially altered, but not completely inverted as shown in genes whose expression is controlled mostly or exclusively by the clock machinery, such as *Dbp* or *Tef*. A mention to this has been modified for clarification, in the discussion section:

Added to lines 552-562:

“Our analyses revealed substantial differences in expression from genes involved in fatty acid oxidation, with marked downregulation in obese mice treated with NAD⁺ at ZT23 (Figure 6E-G). In mouse liver, these genes are oscillatory with a peak of expression at the end of the rest phase[54]. Its expression is to some extent clock-controlled; however, their transcriptional regulation mostly relies on nutritional cues integrated by intracellular signaling, multiple nuclear receptors and transcription factors such as PPAR_γ, PPAR_α or SREBP1, epigenetic regulators including MLL1 or SIRT1, and even neural circuits^{16,90,120-122}. Untimed NAD⁺ rise, through resetting the circadian machinery and the subsequent misalignment from feeding rhythms,

might hinder the coordinated action between the clock and cooperative transcriptional regulators on chromatin, hereby obstructing the adequate control of specific transcriptional programs.”

Finally, following this Reviewers’ observation, we reasoned that the genes related to NAD⁺ metabolism might be good candidates to be altered by time-dependent NAD⁺ bioavailability in livers from our groups of mice, as many of these genes are, also to some extent, clock-controlled, and influenced by NAD⁺ metabolism itself. Hence, we investigated the expression of *Nampt*, *Nmnat3*, *Nadk* and *Nmnk1*, which were previously reported to have BMAL1 binding sites in their regulatory regions[55-57]. We found that in HFN23 mice, these genes appeared overall phase-inverted, and BMAL1 chromatin recruitment was reorganized. A similar case was found for the expression of *Pparg2*, *Ppara*, *Srebf1c* and other lipid-related transcription factors. Notably, PPAR γ 2 protein levels were found to oscillate with an inverted phase in livers from HFN23 mice (Figure 5L, M), which prompted us to hypothesize that its expression might follow a similar trend. We describe this as follows:

Added to lines 441-461:

“We further evaluated the effect of NAD⁺ supplementation in the expression of NAD⁺ biosynthesis and salvage genes *Nmrk1*, *Nampt*, *Nmnat3* and *Nadk* which also showed inverted phase specifically in HFN23 mice (Figure 7G). Accordingly, BMAL1 binding to their regulatory elements was increased at ZT18 in HFN23 mice, yet specific to these group of genes was that NAD⁺ treatment significantly potentiated BMAL1 recruitment to chromatin. Finally, we explored TFs which collaborate with the clock machinery to sustain a rhythmic transcriptional reprogramming in obesity^{16,84}: *Pparg2*, *Ppara* and *Srebf1c*. Transcription for these genes was phase-inverted specifically in HFN23 mice, which was also accompanied by differential BMAL1 chromatin recruitment (Figure 7H). Also, expression levels of additional TFs related to hepatic lipid metabolism *Hnf4a*, *Foxa2*, *Foxo1* and *Cebpa* were altered to a similar extent (Figure S7B). Antiphase expression of key transcription factors regulating hepatic lipid metabolism might underlie the inverted pattern of circulating triglycerides in HFN23 mice (Figure 5H), but other lipids synthesized in the liver might be affected. Accordingly, hepatic cholesterol levels also showed a phase inverted pattern in the liver of HFN23 mice (Figure S7C, S7D), reinforcing the idea that NAD⁺-mediated synchronization of transcriptional rhythms in the liver inverts hepatic lipid metabolism. Together, this data demonstrates a time-dependent transcriptional response to NAD⁺ therapy in the liver of obese mice, through the synchronization of BMAL1 recruitment to

chromatin and rhythmic transcription of clock and clock-controlled genes. Hereby, BMAL1 plays a pivotal role translating fluctuations in NAD⁺ levels to shape circadian transcription.”

Added to Figure 7G, 7H:

Added to Figure S7B, S7C, S7D:

R#5 The data presented in Figure 7 is concerning and unclear. How does NAD invert the circadian cycle? Inversion of the circadian cycle in the liver would cause serious problems and therefore would have a negative impact on liver physiology. This data, if true, does not fit with the argument that HFN23 has no major impact. It would be expected that HFN23 mice would

also have inverted or disrupted metabolic gene expression profiles. Again, this would argue that HFN23 is detrimental to metabolic health and this is counter to the argument of the authors.

A: We thank this Reviewer for these observations. To answer the question “*How does NAD invert the circadian cycle?*”, there are numerous previous reports in the literature showing that NAD⁺ has profound impact in the function of the molecular clock. The first evidence linking NAD to the clock function was probably reported by Rutter et al. in which the ratio between oxidized and reduced forms of NAD was demonstrated to regulate the DNA-binding activity of the CLOCK/NPAS2 (neuronal PAS domain protein 2)-BMAL1 heterodimer[2]. Since then, the field evolved very rapidly and in, in 2008 two reports by leaders in the field were published showing that the NAD⁺-dependent Sirtuin SIRT1 deacetylates the clock components BMAL1 and PER2, modifying its function and hereby, circadian gene expression[4, 5]. Specifically, BMAL1 acetylation promotes its chromatin eviction through recruitment of the PER repressor complex[58]. Furthermore, in 2009 two important reports showed that a circadian rhythm in NAD⁺ levels exists *in vivo*, which imposes rhythms to SIRT1 activity and BMAL1 acetylation (and hereby, its chromatin recruitment)[57, 59]. Redox rhythms in NAD have also been described in the SCN[60], which controls the function of the circadian machinery through SIRT1 activity[61]. Interestingly, a recent report shows that supplementation with NR in mice increases by at least ~25% the BMAL1 binding density in the genome and demonstrates that under normal conditions, “NAD⁺ drives circadian transcription by stabilizing BMAL1 chromatin binding and facilitating the recruitment of collaborative TFs”[3]. Hereby, our work builds over these previous reports and defines the impact of rising NAD⁺ levels at the time they should be low on the clock machinery in obese mice. All these reports extensively demonstrated that the clock machinery is altered by NAD⁺, and here we show that the extent of this alteration can be larger than expected, to the point of inverting rhythmicity. Hereby, to respond this reviewer’s question about “*How does NAD invert the circadian cycle?*”, we performed CHIP experiments on BMAL1 chromatin recruitment to regulatory elements in the genome of selected genes, as follows:

Added to lines 432-441:

“Because redox rhythms regulate DNA binding of CLOCK:BMAL1 heterodimers *in vitro*⁸⁹, and the NAD⁺ precursor NR increases BMAL1 recruitment to chromatin in livers from aged mice⁴⁶, we hypothesized that inverted expression of clock genes in HFN23 might be driven by time-specific recruitment of BMAL1 to chromatin. To test this, we performed CHIP analyses to measure BMAL1 binding to regulatory E-boxes of clock and clock-controlled genes (Figures 7E and 7F). As described⁹⁰, we observed increased recruitment of BMAL1 at ZT6 in livers from

CD, HF and HFN groups of mice for all tested E-boxes; notably, in livers from HFN23 mice, BMAL1 binding appeared significantly increased at ZT18, consistent with inverted expression (Figures 7E, 7F; $P < 0.05$, Two-way ANOVA with Tukey's post-test). A non-related region at the 3' UTR region of *Dbp* gene was used as a negative control. We further evaluated... " (the rest of the experiments are related in the response to the previous question; hence they are not described here to avoid redundancies)

Added to Figure 7E, 7F:

Added to the Methods section:

“Chromatin immunoprecipitation (ChIP)

100-200 mg of liver tissue were homogenized with a pestle in PBS. Dual crosslinking was performed in a final volume of 1ml using 2 mM of DSG (Disuccinimidyl glutarate, ProteoChem, CAS: 79642-50-5) for 10 min at RT on a rotary shaker. DSG was washed out and a second crosslink was performed using 1% formaldehyde (Sigma-Aldrich, F8775) in PBS for 15 min at RT on a rotary shaker. Crosslinking was stopped with 0.125 M glycine for 5 min at 4°C. After two washes with ice-cold PBS, nuclei were isolated by resuspending the tissue in 600 μ L of ice-cold nuclei preparation buffer (NPB: 10 mM HEPES, 10 mM KCl, 1.5 mM MgCl₂, 250 mM sucrose, 0.1% IGEPAL CA-630) and incubating at 4°C for 5 min in rotation. Nuclei were collected by centrifugation at 1,500g for 12 min at 4°C and, and resuspended in 600 μ L of cold nuclear lysis buffer (10 mM Tris pH 8, 1 mM EDTA, 0.5 mM EGTA, 0.3% SDS, 1X cOmplete™ Protease Inhibitor Cocktail, Roche) for 30 min on ice. Nuclear lysates were stored at -80°C. 300 μ L of lysates were sonicated using a Bioruptor Pico Sonicator (Diagenode) for 15 cycles (30 s ON/30 s OFF). Chromatin fragments (100-500 bp) were evaluated on agarose gels using 10 μ L of sonicated chromatin for DNA purification using the phenol method. 600 μ L of ice-cold ChIP-

dilution buffer (1% Triton X-100, 2 mM EDTA, 20 mM Tris pH 8, 150 mM NaCl, 1 mM PMSF, 1X cOmplete™ Protease Inhibitor Cocktail, Roche) was added to the fragmented chromatin, and 10% volume was recovered as the Input. Immunoprecipitation was set up overnight at 4°C, by adding 20 µL of magnetic beads (Magna ChIP Protein G Magnetic Beads C #16-662, Sigma-Aldrich) and a combination of two anti BMAL1 antibodies: 1.25 µL rabbit anti-BMAL1 (ab3350, Abcam) and 2.5 µL rabbit anti-BMAL1 (ab93806, Abcam). Immunoprecipitations with 4 µL of normal mouse IgG (Sigma-Aldrich, Cat. No. 18765) were performed simultaneously. Sequential washes of the magnetic beads were performed for 10 min at 4°C, as follows: Wash buffer 1 (20 mM Tris pH 8, 0.1% SDS, 1% Triton X-100, 150 mM NaCl, 2 mM EDTA), Wash buffer 2 (20 mM Tris pH 8, 0.1% SDS, 1% Triton X-100, 500 mM NaCl, 2 mM EDTA), Wash buffer 3 (10 mM Tris pH 8, 250 mM LiCl, 1% IGEPAL CA-630, 1% sodium deoxycolate) and TE buffer (10 mM Tris pH 8, 1 mM EDTA). Chromatin was eluted by adding 400 µL of fresh elution buffer (10 mM Tris pH 8, 0.5% SDS, 300 mM NaCl, 5 mM EDTA, 0.05 mg/mL proteinase K) to the magnetic beads and incubating overnight at 65°C. A treatment with RNase A at 0.1 mg/ml for 30 min at 37 °C was performed. The DNA was purified from the IPs and Inputs by adding one volume of phenol:chloroform:isoamyl alcohol (25:24:1). After mixing and centrifugation, the aqueous phase was recovered, and DNA was precipitated by adding 1/10 volumes of sodium acetate (0.3 M pH 5.2), 20 µg of glycogen (10901393001, Roche) and 2 volumes of ice-cold ethanol, at -80°C overnight. DNA was pelleted by centrifugation at 13500 rpm for 30 min at 4°C. The DNA was washed with 70% ethanol, and resuspended in 50 µL of molecular grade water. 1.5 µL were used for subsequent qRT-PCR reactions with specific primers designed using Primer3web, within regulatory regions previously identified as BMAL1 binding sites in mouse liver, as reported in the ChIP-Atlas database¹⁴⁶. Primer sequences are available in Supplementary Table 4.”

This Reviewer argues that *“Inversion of the circadian cycle in the liver would cause serious problems and therefore would have a negative impact on liver physiology. This data, if true, does not fit with the argument that HFN23 has no major impact”*. Indeed, we do not claim in our manuscript that *“HFN23 has no major impact”*, as this reviewer indicates. Instead, we describe that:

- 1) HFN23 mice show significantly worse performance in glucose and insulin tolerance tests when compared with HFN mice. This was also demonstrated in HFNAM_23 mice

(Figure 5E, F, S8). Remarkably, performance of HFN mice in these tests is largely comparable to lean, CD mice.

- 2) HFN23 mice show significantly increased circulating triglycerides than HFN mice, and comparable to HF mice.
- 3) HFN23 mice show molecular markers compatible with metabolic disease, including loss of rhythms in hepatic AMPK phosphorylation and constitutively increased S6K phosphorylation, contrariwise to what we observe in HFN mice.
- 4) HFN23 mice show decreased expression levels in genes related to lipid oxidative pathways, whose expression is considered to improve lipid metabolism under HFD conditions.
- 5) Strikingly, HFN23 mice keep gaining weight after the pharmacological intervention of NAD⁺, while HFN mice consistently loose wight. Weight loss has been pervasively reported as a major effect of boosting NAD⁺ levels in mouse models of obesity. We don't find this effect in HFN23, or HFNAM23 mice, but is sharply detected in HFN and also shown in HFNAM11 mice.

All these detrimental effects observed in HFN23 mice are accompanied by hepatic inversion of the clock machinery after three weeks of treatment, which is “true” because we consistently observed this effect in biological replicates using NAD⁺, with n= 5-6 mice per time point, and we consistently saw the same hepatic inversion in an independent experiment using NAM as an NAD⁺ precursor. We have detailed these experimental procedures in the methods section in a way that any interested laboratory could easily reproduce this data, while also providing the source data.

Likely, this inversion in hepatic clock is progressive, that is, it might take several days to be established, considering that *in vivo*, adjusting the clock to new environmental conditions, such as in jet lag situations, is generally gradual. For example, in the liver, restricted feeding inverts the phase of the hepatic clock after six days[62]. In our model, after three weeks of NAD⁺ treatment at ZT23, the clock is completely inverted in the liver, and it is conceptually apparent that prolonging this treatment might progressively increase the detrimental effects. With our approach, we cannot rule out this possibility as it would require an extensive study at different lengths of treatment. We consider that deciphering the length of the NAD⁺ treatment at ZT23 necessary to observe specific detrimental effects compared to non-treated obese mice would require an extensive amount of work, mice and resources and is now out of the scope of this research.

R#5 *The authors also claim that HFN is beneficial while HFN23 is not. The ORO staining does not support this claim in Fig 5I and S5G, ORO looks exactly the same between the 2 groups. Therefore, how is HFN only beneficial? This data does not support the claim that time-dependent NAD supplementation is beneficial.*

A: We thank this reviewer for pointing this out, as we think this is an important observation in our obesity model. From carefully examining our data, it becomes clear that NAD⁺ supplementation has time-dependent and time-independent effects, as suggested by this Reviewer. Indeed, ORO stain shows overall improvement in neutral hepatic lipid accumulation, as a time-independent effect of NAD⁺ supplementation. In fact, niacin, an important precursor for NAD⁺ via the Preiss-Handler pathway, is considered the first broad lipid-lowering drug and this effect is partially mediated by rising NAD⁺ levels[63]. Hence, it is conceptually apparent that a reduced hepatic lipid content in HFN23 mice might be due to a time-independent and robust effect of increased NAD⁺.

However, most of the data shows time-dependent effects of NAD⁺ supplementation in many metabolic parameters. As indicated, in this new version of the manuscript, we have refrained to claim that the NAD⁺ treatment at ZT23 is not beneficial, and instead, we now argue that NAD⁺ shows time-dependent effects, as dissected across the manuscript, one of them being the inversion of the hepatic clock. As NAD⁺ treatment at ZT23 does that, we also argue that its effects are potentially detrimental on the long term, because of the observed uncoupling between the central and the hepatic clock. This assumption is based on previous reports with distinct paradigms, for example, mice with food access restricted to the light period, or individuals subjected to prolonged jet-lag, where uncoupling clocks appears detrimental to health, and vice-versa, aligning clocks supports metabolic fitness[64-69]. As a proof of this concept, our HFN23 mice do not significantly improve their ITT or GTT after 20 days of treatment, neither lose weight, at least in part due to misaligned clocks, which compete against the beneficial effects of NAD⁺ shown in HFN mice. If clocks remained misaligned through this therapy, it is possible that the detrimental effects would exceed the beneficial, hereby exacerbating the metabolic syndrome. With respect, we find that investigating the balance between beneficial and detrimental effects of NAD⁺ therapy at ZT23 as a function of time of treatment would require extensive resources and time, hereby should be a subject for future research.

Added to lines 582-594:

“Circadian misalignment imposed by antiphase NAD⁺ in our HFN23 and HFNAM_23 mice might obstruct metabolic improvements, through uncoupling of the central light-synchronized and peripheral NAD⁺-synchronized clocks. Although hepatic neutral lipid content was reduced independently of time-of-treatment (Figure 5I-K), significant improvement of glucose homeostasis and hepatic insulin signaling were apparent only in HFN mice. Indeed, circadian misalignment has been extensively reported to drive metabolic dysfunction both in mouse and humans¹³¹⁻¹³³. In this scenario, expression of clock genes in the SCN was largely intact upon NAD⁺ injection, and consequently, locomotor activity remains aligned with the light-dark cycles also in HFN23 mice (Figure 8A-C). Uncoupled liver and SCN clocks have been previously reported in mice when access to food is restricted to the light period^{91,134,135}; however, our HFN23 mice did not show significant variations in eating behavior (Figure 8D-E), evidencing that uncoupling the central and hepatic clocks is a time-dependent effect of NAD⁺ supply.”

We have now responded to this Reviewer’s concerns in full, we hope that we have given clarity into the way our results are conveyed, and we thank Reviewer#5 for the thorough insights which have contributed to substantially improve the manuscript.

Bibliography

1. Menet JS, Pescatore S, Rosbash M: **CLOCK:BMAL1 is a pioneer-like transcription factor**. *Genes & development* 2014, **28**(1):8-13.
2. Rutter J, Reick M, Wu LC, McKnight SL: **Regulation of clock and NPAS2 DNA binding by the redox state of NAD cofactors**. *Science* 2001, **293**(5529):510-514.
3. Levine DC, Hong H, Weidemann BJ, Ramsey KM, Affinati AH, Schmidt MS, Cedernaes J, Omura C, Braun R, Lee C *et al*: **NAD⁺ Controls Circadian Reprogramming through PER2 Nuclear Translocation to Counter Aging**. *Molecular Cell* 2020, **78**(5):835-849.e837.
4. Asher G, Gatfield D, Stratmann M, Reinke H, Dibner C, Kreppel F, Mostoslavsky R, Alt FW, Schibler U: **SIRT1 Regulates Circadian Clock Gene Expression through PER2 Deacetylation**. *Cell* 2008, **134**(2):317-328.
5. Nakahata Y, Kaluzova M, Grimaldi B, Sahar S, Hirayama J, Chen D, Guarente LP, Sassone-Corsi P: **The NAD⁺-dependent deacetylase SIRT1 modulates CLOCK-mediated chromatin remodeling and circadian control**. *Cell* 2008, **134**(2):329-340.

6. Aguilar-Arnal L, Katada S, Orozco-Solis R, Sassone-Corsi P: **NAD(+)-SIRT1 control of H3K4 trimethylation through circadian deacetylation of MLL1.** *Nature structural & molecular biology* 2015, **22**(4):312-318.
7. Longo KA, Charoenthongtrakul S, Giuliana DJ, Govek EK, McDonagh T, Distefano PS, Geddes BJ: **The 24-hour respiratory quotient predicts energy intake and changes in body mass.** *American journal of physiology Regulatory, integrative and comparative physiology* 2010, **298**(3):R747-754.
8. Taher J, Baker CL, Cuizon C, Masoudpour H, Zhang R, Farr S, Naples M, Bourdon C, Pausova Z, Adeli K: **GLP-1 receptor agonism ameliorates hepatic VLDL overproduction and de novo lipogenesis in insulin resistance.** *Molecular metabolism* 2014, **3**(9):823-833.
9. O'Neal TJ, Friend DM, Guo J, Hall KD, Kravitz AV: **Increases in Physical Activity Result in Diminishing Increments in Daily Energy Expenditure in Mice.** *Current Biology* 2017, **27**(3):423-430.
10. Morato L, Astori S, Zalachoras I, Rodrigues J, Ghosal S, Huang W, Guillot de Suduiraut I, Grosse J, Zanoletti O, Cao L *et al*: **eNAMPT actions through nucleus accumbens NAD(+)/SIRT1 link increased adiposity with sociability deficits programmed by peripuberty stress.** *Science advances* 2022, **8**(9):eabj9109.
11. Virtue S, Even P, Vidal-Puig A: **Below thermoneutrality, changes in activity do not drive changes in total daily energy expenditure between groups of mice.** *Cell metabolism* 2012, **16**(5):665-671.
12. Bi P, Shan T, Liu W, Yue F, Yang X, Liang X-R, Wang J, Li J, Carlesso N, Liu X *et al*: **Inhibition of Notch signaling promotes browning of white adipose tissue and ameliorates obesity.** *Nature Medicine* 2014, **20**(8):911-918.
13. Szentirmai É, Kapás L: **The role of the brown adipose tissue in β 3-adrenergic receptor activation-induced sleep, metabolic and feeding responses.** *Scientific reports* 2017, **7**(1):958.
14. Krishnan J, Danzer C, Simka T, Ukropec J, Walter KM, Kumpf S, Mirtschink P, Ukropcova B, Gasperikova D, Pedrazzini T *et al*: **Dietary obesity-associated Hif1alpha activation in adipocytes restricts fatty acid oxidation and energy expenditure via suppression of the Sirt2-NAD+ system.** *Genes & development* 2012, **26**(3):259-270.
15. Lima NDS, Teixeira L, Gambero A, Ribeiro ML: **Guarana (Paullinia cupana) Stimulates Mitochondrial Biogenesis in Mice Fed High-Fat Diet.** *Nutrients* 2018, **10**(2).
16. Hwang JH, Kim DW, Jo EJ, Kim YK, Jo YS, Park JH, Yoo SK, Park MK, Kwak TH, Kho YL *et al*: **Pharmacological stimulation of NADH oxidation ameliorates obesity and related phenotypes in mice.** *Diabetes* 2009, **58**(4):965-974.
17. Thupari JN, Kim EK, Moran TH, Ronnett GV, Kuhajda FP: **Chronic C75 treatment of diet-induced obese mice increases fat oxidation and reduces food intake to reduce adipose mass.** *American journal of physiology Endocrinology and metabolism* 2004, **287**(1):E97-E104.
18. Sarver DC, Stewart AN, Rodriguez S, Little HC, Aja S, Wong GW: **Loss of CTRP4 alters adiposity and food intake behaviors in obese mice.** *American journal of physiology Endocrinology and metabolism* 2020, **319**(6):E1084-E1100.
19. Zou T, Li S, Wang B, Wang Z, Liu Y, You J: **Curcumin improves insulin sensitivity and increases energy expenditure in high-fat-diet-induced obese mice associated with activation of FNDC5/irisin.** *Nutrition* 2021, **90**:111263.
20. Rachid TL, Penna-de-Carvalho A, Bringhenti I, Aguila MB, Mandarim-de-Lacerda CA, Souza-Mello V: **PPAR-alpha agonist elicits metabolically active brown adipocytes and weight loss in diet-induced obese mice.** *Cell biochemistry and function* 2015, **33**(4):249-256.
21. Sethi J, Sanchez-Alavez M, Tabarean IV: **Loss of histaminergic modulation of thermoregulation and energy homeostasis in obese mice.** *Neuroscience* 2012, **217**:84-95.

22. Kshatriya D, Li X, Giunta GM, Yuan B, Zhao D, Simon JE, Wu Q, Bello NT: **Phenolic-enriched raspberry fruit extract (*Rubus idaeus*) resulted in lower weight gain, increased ambulatory activity, and elevated hepatic lipoprotein lipase and heme oxygenase-1 expression in male mice fed a high-fat diet.** *Nutrition Research* 2019, **68**:19-33.
23. Yang X, Wu F, Li L, Lynch EC, Xie L, Zhao Y, Fang K, Li J, Luo J, Xu L *et al*: **Celastrol alleviates metabolic disturbance in high-fat diet-induced obese mice through increasing energy expenditure by ameliorating metabolic inflammation.** *Phytotherapy Research* 2021, **35**(1):297-310.
24. Henstridge DC, Bruce CR, Drew BG, Tory K, Kolonics A, Estevez E, Chung J, Watson N, Gardner T, Lee-Young RS *et al*: **Activating HSP72 in rodent skeletal muscle increases mitochondrial number and oxidative capacity and decreases insulin resistance.** *Diabetes* 2014, **63**(6):1881-1894.
25. Zeng H-l, Huang S-l, Xie F-c, Zeng L-m, Hu Y-h, Leng Y: **Yhhu981, a novel compound, stimulates fatty acid oxidation via the activation of AMPK and ameliorates lipid metabolism disorder in ob/ob mice.** *Acta Pharmacologica Sinica* 2015, **36**(3):343-352.
26. Walewski JL, Ge F, Lobdell IV H, Levin N, Schwartz GJ, Vasselli JR, Pomp A, Dakin G, Berk PD: **Spexin is a novel human peptide that reduces adipocyte uptake of long chain fatty acids and causes weight loss in rodents with diet-induced obesity.** *Obesity* 2014, **22**(7):1643-1652.
27. Yan C, Zeng T, Lee K, Nobis M, Loh K, Gou L, Xia Z, Gao Z, Bensellam M, Hughes W *et al*: **Peripheral-specific Y1 receptor antagonism increases thermogenesis and protects against diet-induced obesity.** *Nature communications* 2021, **12**(1):2622.
28. Joo J, Cox CC, Kindred ED, Lashinger LM, Young ME, Bray MS: **The acute effects of time-of-day-dependent high fat feeding on whole body metabolic flexibility in mice.** *International Journal of Obesity* 2016, **40**(9):1444-1451.
29. Boutant M, Joffraud M, Kulkarni SS, Garcia-Casarrubios E, Garcia-Roves PM, Ratajczak J, Fernandez-Marcos PJ, Valverde AM, Serrano M, Canto C: **SIRT1 enhances glucose tolerance by potentiating brown adipose tissue function.** *Molecular metabolism* 2015, **4**(2):118-131.
30. Gerhart-Hines Z, Dominy John E, Blättler Sharon M, Jedrychowski Mark P, Banks Alexander S, Lim J-H, Chim H, Gygi Steven P, Puigserver P: **The cAMP/PKA Pathway Rapidly Activates SIRT1 to Promote Fatty Acid Oxidation Independently of Changes in NAD+.** *Molecular cell* 2011, **44**(6):851-863.
31. Mitchell Sarah J, Martin-Montalvo A, Mercken Evi M, Palacios Hector H, Ward Theresa M, Abulwerdi G, Minor Robin K, Vlasuk George P, Ellis James L, Sinclair David A *et al*: **The SIRT1 Activator SRT1720 Extends Lifespan and Improves Health of Mice Fed a Standard Diet.** *Cell reports* 2014, **6**(5):836-843.
32. Bruckbauer A, Zemel MB, Thorpe T, Akula MR, Stuckey AC, Osborne D, Martin EB, Kennel S, Wall JS: **Synergistic effects of leucine and resveratrol on insulin sensitivity and fat metabolism in adipocytes and mice.** *Nutrition & Metabolism* 2012, **9**(1):77.
33. Li Z, Zhang Z, Ke L, Sun Y, Li W, Feng X, Zhu W, Chen S: **Resveratrol promotes white adipocytes browning and improves metabolic disorders in Sirt1-dependent manner in mice.** *The FASEB Journal* 2020, **34**(3):4527-4539.
34. Simonson DC, DeFronzo RA: **Indirect calorimetry: methodological and interpretative problems.** *The American journal of physiology* 1990, **258**(3 Pt 1):E399-412.
35. Nowell NW: **Circadian rhythm of glucose tolerance in laboratory mice.** *Diabetologia* 1970, **6**(5):488-492.
36. Van Cauter E, Polonsky KS, Scheen AJ: **Roles of circadian rhythmicity and sleep in human glucose regulation.** *Endocrine reviews* 1997, **18**(5):716-738.

37. Kumar Jha P, Challet E, Kalsbeek A: **Circadian rhythms in glucose and lipid metabolism in nocturnal and diurnal mammals.** *Molecular and Cellular Endocrinology* 2015, **418**:74-88.
38. Yoshino J, Mills KF, Yoon MJ, Imai S: **Nicotinamide mononucleotide, a key NAD(+) intermediate, treats the pathophysiology of diet- and age-induced diabetes in mice.** *Cell metabolism* 2011, **14**(4):528-536.
39. Cantó C, Houtkooper RH, Pirinen E, Youn DY, Oosterveer MH, Cen Y, Fernandez-Marcos PJ, Yamamoto H, Andreux PA, Cettour-Rose P: **The NAD⁺ precursor nicotinamide riboside enhances oxidative metabolism and protects against high-fat diet-induced obesity.** *Cell metabolism* 2012, **15**(6):838-847.
40. Stromsdorfer Kelly L, Yamaguchi S, Yoon Myeong J, Moseley Anna C, Franczyk Michael P, Kelly Shannon C, Qi N, Imai S-i, Yoshino J: **NAMPT-Mediated NAD⁺ Biosynthesis in Adipocytes Regulates Adipose Tissue Function and Multi-organ Insulin Sensitivity in Mice.** *Cell Reports* 2016, **16**(7):1851-1860.
41. Pham TX, Bae M, Kim M-B, Lee Y, Hu S, Kang H, Park Y-K, Lee J-Y: **Nicotinamide riboside, an NAD⁺ precursor, attenuates the development of liver fibrosis in a diet-induced mouse model of liver fibrosis.** *Biochimica et Biophysica Acta (BBA) - Molecular Basis of Disease* 2019, **1865**(9):2451-2463.
42. Mendez-Lara KA, Rodriguez-Millan E, Sebastian D, Blanco-Soto R, Camacho M, Nan MN, Diarte-Anazco EMG, Mato E, Lope-Piedrafita S, Roglans N *et al*: **Nicotinamide Protects Against Diet-Induced Body Weight Gain, Increases Energy Expenditure, and Induces White Adipose Tissue Beiging.** *Molecular nutrition & food research* 2021, **65**(11):e2100111.
43. Mitchell SJ, Bernier M, Aon MA, Cortassa S, Kim EY, Fang EF, Palacios HH, Ali A, Navas-Enamorado I, Di Francesco A *et al*: **Nicotinamide Improves Aspects of Healthspan, but Not Lifespan, in Mice.** *Cell metabolism* 2018, **27**(3):667-676.e664.
44. Sambeat A, Ratajczak J, Joffraud M, Sanchez-Garcia JL, Giner MP, Valsesia A, Giroud-Gerbetant J, Valera-Alberni M, Cercillieux A, Boutant M *et al*: **Endogenous nicotinamide riboside metabolism protects against diet-induced liver damage.** *Nature Communications* 2019, **10**(1):4291.
45. Hou X, Xu S, Maitland-Toolan KA, Sato K, Jiang B, Ido Y, Lan F, Walsh K, Wierzbicki M, Verbeuren TJ *et al*: **SIRT1 regulates hepatocyte lipid metabolism through activating AMP-activated protein kinase.** *The Journal of biological chemistry* 2008, **283**(29):20015-20026.
46. Canto C, Auwerx J: **PGC-1alpha, SIRT1 and AMPK, an energy sensing network that controls energy expenditure.** *Current opinion in lipidology* 2009, **20**(2):98-105.
47. Wang L-F, Wang X-N, Huang C-C, Hu L, Xiao Y-F, Guan X-H, Qian Y-S, Deng K-Y, Xin H-B: **Inhibition of NAMPT aggravates high fat diet-induced hepatic steatosis in mice through regulating Sirt1/AMPK α /SREBP1 signaling pathway.** *Lipids in Health and Disease* 2017, **16**(1):82.
48. Han F, Li C-F, Cai Z, Zhang X, Jin G, Zhang W-N, Xu C, Wang C-Y, Morrow J, Zhang S *et al*: **The critical role of AMPK in driving Akt activation under stress, tumorigenesis and drug resistance.** *Nature Communications* 2018, **9**(1):4728.
49. Wang RH, Kim HS, Xiao C, Xu X, Gavriloova O, Deng CX: **Hepatic Sirt1 deficiency in mice impairs mTORC2/Akt signaling and results in hyperglycemia, oxidative damage, and insulin resistance.** *The Journal of clinical investigation* 2011, **121**(11):4477-4490.
50. Banks AS, Kon N, Knight C, Matsumoto M, Gutiérrez-Juárez R, Rossetti L, Gu W, Accili D: **Sirt1 Gain of Function Increases Energy Efficiency and Prevents Diabetes in Mice.** *Cell metabolism* 2008, **8**(4):333-341.
51. Akie TE, Liu L, Nam M, Lei S, Cooper MP: **OXPHOS-Mediated Induction of NAD⁺ Promotes Complete Oxidation of Fatty Acids and Interdicts Non-Alcoholic Fatty Liver Disease.** *PloS one* 2015, **10**(5):e0125617.

52. Loh K, Deng H, Fukushima A, Cai X, Boivin B, Galic S, Bruce C, Shields BJ, Skiba B, Ooms LM *et al*: **Reactive Oxygen Species Enhance Insulin Sensitivity**. *Cell metabolism* 2009, **10**(4):260-272.
53. Templeman NM, Flibotte S, Chik JHL, Sinha S, Lim GE, Foster LJ, Nislow C, Johnson JD: **Reduced Circulating Insulin Enhances Insulin Sensitivity in Old Mice and Extends Lifespan**. *Cell Rep* 2017, **20**(2):451-463.
54. Guan D, Xiong Y, Borck PC, Jang C, Doulias P-T, Papazyan R, Fang B, Jiang C, Zhang Y, Briggs ER *et al*: **Diet-Induced Circadian Enhancer Remodeling Synchronizes Opposing Hepatic Lipid Metabolic Processes**. *Cell* 2018, **174**(4):831-842.e812.
55. Koike N, Yoo SH, Huang HC, Kumar V, Lee C, Kim TK, Takahashi JS: **Transcriptional architecture and chromatin landscape of the core circadian clock in mammals**. *Science* 2012, **338**(6105):349-354.
56. Koronowski KB, Kinouchi K, Welz P-S, Smith JG, Zinna VM, Shi J, Samad M, Chen S, Magnan CN, Kinchen JM *et al*: **Defining the Independence of the Liver Circadian Clock**. *Cell* 2019, **177**(6):1448-1462.e1414.
57. Nakahata Y, Sahar S, Astarita G, Kaluzova M, Sassone-Corsi P: **Circadian control of the NAD⁺ salvage pathway by CLOCK-SIRT1**. *Science* 2009, **324**(5927):654-657.
58. Hirayama J, Sahar S, Grimaldi B, Tamaru T, Takamatsu K, Nakahata Y, Sassone-Corsi P: **CLOCK-mediated acetylation of BMAL1 controls circadian function**. *Nature* 2007, **450**(7172):1086-1090.
59. Ramsey KM, Yoshino J, Brace CS, Abrassart D, Kobayashi Y, Marcheva B, Hong H-K, Chong JL, Buhr ED, Lee C *et al*: **Circadian clock feedback cycle through NAMPT-mediated NAD⁺ biosynthesis**. *Science* 2009, **324**(5927):651-654.
60. Wang TA, Yu YV, Govindaiah G, Ye X, Artinian L, Coleman TP, Sweedler JV, Cox CL, Gillette MU: **Circadian rhythm of redox state regulates excitability in suprachiasmatic nucleus neurons**. *Science* 2012, **337**(6096):839-842.
61. Chang H-C, Guarente L: **SIRT1 Mediates Central Circadian Control in the SCN by a Mechanism that Decays with Aging**. *Cell* 2013, **153**(7):1448-1460.
62. Damiola F, Le Minh N, Preitner N, Kornmann B, Fleury-Olela F, Schibler U: **Restricted feeding uncouples circadian oscillators in peripheral tissues from the central pacemaker in the suprachiasmatic nucleus**. *Genes & development* 2000, **14**(23):2950-2961.
63. Romani M, Hofer DC, Katsyuba E, Auwerx J: **Niacin: an old lipid drug in a new NAD⁺ dress**. *J Lipid Res* 2019, **60**(4):741-746.
64. Parsons MJ, Moffitt TE, Gregory AM, Goldman-Mellor S, Nolan PM, Poulton R, Caspi A: **Social jetlag, obesity and metabolic disorder: investigation in a cohort study**. *International journal of obesity* 2015, **39**(5):842-848.
65. Scheer FA, Hilton MF, Mantzoros CS, Shea SA: **Adverse metabolic and cardiovascular consequences of circadian misalignment**. *Proc Natl Acad Sci U S A* 2009, **106**(11):4453-4458.
66. McHill AW, Melanson EL, Higgins J, Connick E, Moehlman TM, Stothard ER, Wright KP, Jr.: **Impact of circadian misalignment on energy metabolism during simulated nightshift work**. *Proc Natl Acad Sci U S A* 2014, **111**(48):17302-17307.
67. Leproult R, Holmback U, Van Cauter E: **Circadian misalignment augments markers of insulin resistance and inflammation, independently of sleep loss**. *Diabetes* 2014, **63**(6):1860-1869.
68. Wefers J, van Moorsel D, Hansen J, Connell NJ, Havekes B, Hoeks J, van Marken Lichtenbelt WD, Duez H, Phielix E, Kalsbeek A *et al*: **Circadian misalignment induces fatty acid metabolism gene profiles and compromises insulin sensitivity in human skeletal muscle**. *Proc Natl Acad Sci U S A* 2018, **115**(30):7789-7794.
69. Baron KG, Reid KJ: **Circadian misalignment and health**. *International review of psychiatry* 2014, **26**(2):139-154.

REVIEWER COMMENTS

Reviewer #1 (Remarks to the Author):

Authors have adequately answered all reviewer queries. They did a great effort to improve this interesting article that should be published in nature Communications.

Reviewer #2 (Remarks to the Author):

Remarks to the Authors:

The authors have addressed some questions with new experimental data. However, there are still some major concerns.

The authors state that NAD⁺ is a potent time-giver (or a synchronizer) for the hepatic molecular clock and timed NAD⁺ treatment resets the hepatic clock. However, the data showed that supplementing NAD⁺ at ZT11 hardly had an effect on the expression profiles of clock genes compared to HF group (HFN vs HF). Gene expression profiles of HF and HFN are very similar in Figure 7C/D. Moreover, in Figure 3C, rhythms in the core clock proteins BMAL1, CRY1, PER2 and REV-ERBa in HFN group were not significant different compared to HF group. Therefore, it is not convincing that "timed NAD⁺ treatment resets the hepatic clock".

Additionally, as reviewer # 5 pointed out, metabolic gene expression presented in Figure 6E/F/G does not support the hypothesis that supplementing NAD⁺ at ZT11 enhances fatty acid oxidation to contribute to weight loss and decreased hepatic and circulating triglycerides. This is because the expression of fatty acid oxidation genes is high in all HF-fed groups, of note, the HF and HFN group are comparable in those gene expression, although the authors state that the breadth of transcriptional activity extends to ZT18 in HFN. The slight differences between the two groups (HF vs HFN) make it hard to believe that NAD⁺ works through fatty acid oxidation pathway to improve metabolic markers.

The data presented in Figure 7C/D/E/F/H show hepatic inversion of the clock machinery and the NAD⁺ biosynthesis and salvages genes in HFN23 mice. Moreover, inverted pattern of circulating triglycerides in HFN23 mice was also shown in Figure 5H. As those remarkable impact of NAD⁺ treatment at ZT23 is different from NAD⁺ treatment at ZT11 (HFN group), the authors need to address why the treatment at ZT23 alleviates the disease rather than exacerbates it.

Reviewer #4 (Remarks to the Author):

NCOMMS-21-09336B

Escalante-Covarrubias et al.

"Time-of-day defines the efficacy of NAD⁺ to treat diet-induced metabolic disease by adjusting oscillations of the hepatic circadian clock"

In this further revised version, the authors adequately addressed this reviewer's request and provided a new set of results that show differences in BMAL1 recruitment between HNF and HFN23 groups. This new data set strengthens the authors' conclusion, providing insights into the mechanism of timed NAD⁺ administration. Therefore, this reviewer now believes that the manuscript is ready to be accepted for Nature Communications.

Having that said, the following minor textual revisions should be made before the official acceptance:

1) In the Discussion section, the authors describe that "NAD⁺ is produced in all tissues from the salvageable precursors NAM, nicotinamide riboside (NR) or niacin." This is not correct. The only salvageable precursor is nicotinamide (NAM). The authors should use nicotinic acid (NA), instead of niacin, because niacin means both NAM and NA, and should also include nicotinamide mononucleotide

(NMN) as an important NAD⁺ precursor/intermediate.

2) Another statement that "it is generally accepted that NR or NAM enter the cell, while extracellular NAD⁺ and NMN are converted to NR." This is not correct, either. Whereas there are still scientific arguments in the field, NAD⁺ can enter the cell through Connexin 43 hemichannels, and NMN can enter through a NMN transporter Slc12a8.

3) The statement in the last paragraph of the Discussion section that "in humans, clinical trials aiming to boost endogenous NAD⁺ for treatment of metabolic diseases are increasing, in many cases reporting conflicting results." This statement is not accurate because the results are not "conflicting", but quite clear. There have already been a significant number of clinical trials on both NR and NMN. Regarding metabolic diseases, particularly type 2 diabetes or prediabetic conditions, there has been no efficacy reported with NR, whereas NMN has been reported to increase skeletal muscle insulin sensitivity in prediabetic women.

Reviewer #5 (Remarks to the Author):

The authors have gone to great lengths to address the comments of the reviewers in a very verbose manner. It is the opinion of this reviewer that the rebuttal of the authors has not addressed the major concerns of all reviewers.

1) The new ChIP data is quite unclear and does not provide any new mechanism. Looking at the data closely, again, there are major discrepancies in Bmal1 binding and gene expression such as Tef1, Nampt, Nmrk1, ppar, srebp, etc. This does not explain how NAD supplementation is acting and the detailed molecular mechanism remains undefined.

2) The extensive response regarding body weight has not been addressed. Indirect calorimetry is the gold standard, the RER changes remain unexplained. The new data presented regarding energy intake on page 17 clearly show the large discrepancy and decrease in energy intake in the HFN23 group. This is a concern.

3) The authors simply removed leptin data because it did not support the hypothesis.

4) The GTT and ITT remains unclear and the extensive argument provided has not changed the quality of the data. Again, this could simply be a reflection of food/energy intake.

5) There is no explanation provided in the major discrepancy between ORO staining and TG levels. These data are contradictory and do not support the conclusions on the manuscript. If the authors speculate that other lipids maybe impacted, please provide that data.

6) The gene expression data in Figure 6 and the explanation provided by the authors is still highly contradictory and does not support the hypothesis. The data provided in Figure 6 does not show major improvements in metabolic gene expression between HF and HFN. The HFN does not look like control. This is a major concern and is not addressed.

7) Reviewers raised concerns regarding the HFN23 data. This remains highly speculative and the data is not well-supported. Even if the authors changed the terminology they used, this does not address this major concern.

POINT BY POINT RESPONSES TO REVIEWERS' COMMENTS

Reviewer #1 (Remarks to the Author):

R#1: *Authors have adequately answered all reviewer queries. They did a great effort to improve this interesting article that should be published in nature Communications.*

A: We thank this reviewer for their comments which have contributed to substantially improve this manuscript.

Reviewer #2 (Remarks to the Author):

R#2: *The authors have addressed some questions with new experimental data. However, there are still some major concerns.*

The authors state that NAD⁺ is a potent time-giver (or a synchronizer) for the hepatic molecular clock and timed NAD⁺ treatment resets the hepatic clock. However, the data showed that supplementing NAD⁺ at ZT11 hardly had an effect on the expression profiles of clock genes compared to HF group (HFN vs HF). Gene expression profiles of HF and HFN are very similar in Figure 7C/D. Moreover, in Figure 3C, rhythms in the core clock proteins BMAL1, CRY1, PER2 and REV-ERB α in HFN group were not significant different compared to HF group. Therefore, it is not convincing that "timed NAD⁺ treatment resets the hepatic clock".

A: This is well taken. We agree that NAD⁺ at ZT11 had a mild effect on the hepatic clock. However, this was expected since the aim of supplying NAD⁺ at ZT11 was aligning the therapy with endogenous rhythms. If NAD⁺ is a time-giver, when the time of treatment is aligned with the endogenous rhythms (ZT11), it will hardly show an effect. However, at different times, the hepatic clock presents a phase shift as shown in HFN23 mice (12 hour-phase shift). To further reinforce this idea, we supplied NAD⁺ at ZT17 and/or ZT5 in obese mice and measured circadian oscillations of clock genes and proteins in their livers (4 time points per group, n=3 mice per time point). The data is as follows:

Added to lines 447-458:

"To further reinforce the notion that NAD⁺ supply displaces the phase of circadian oscillations from the hepatic clock, we analyzed livers from obese mice treated at ZT5 (HFN5) or at ZT17 (HFN17). We found expected phase advance in clock protein oscillations in livers from HFN5 mice, while treatment at ZT17 resulted in phase delayed oscillations (Figure S7B, S7C) of distinct extents with respect to oscillations detected in HF mice (Figure S7C). This was paralleled by coherent phase shifts in clock and clock-controlled genes expression (Figures S7A, S7D). As expected, the phase shift was more evident in HFN17 than in HFN5, because the mammalian clock is reluctant to phase advances and more susceptible to phase delays⁹¹⁻⁹⁴. Overall, these analyses show that timed NAD⁺ supply reshapes hepatic circadian oscillations of the clock machinery, adjusting their phase to the time of treatment."

91 Aschoff, J., Hoffmann, K., Pohl, H. & Wever, R. Re-entrainment of circadian rhythms after phase-shifts of the Zeitgeber. *Chronobiologia* 2, 23-78, (1975).

92 Eastman, C. I. & Martin, S. K. How to use light and dark to produce circadian adaptation to night shift work. *Annals of medicine* 31, 87-98, (1999).

93 Mitchell, P. J., Hoese, E. K., Liu, L., Fogg, L. F. & Eastman, C. I. Conflicting bright light exposure during night shifts impedes circadian adaptation. *Journal of biological rhythms* 12, 5-15, (1997).

94 Revell, V. L. et al. Advancing human circadian rhythms with afternoon melatonin and morning intermittent bright light. *The Journal of clinical endocrinology and metabolism* 91, 54-59, (2006).

Added to Figure S7:

Added to Figure S7 caption:

“Figure S7. Time-of-day dependent effects of NAD⁺ supply on circadian gene expression.

(A) Acrophase and p-value for circadian rhythmicity from clock and clock-controlled gene expression was assessed by CircaWave. (B) Circadian clock protein expression from liver whole cell extracts of obese mice treated with NAD⁺ at ZT5 (HFN5) or ZT17 (HFN17). Tubulin or GAPDH were used as loading control. (C) Quantification of western blots from n = 3 mice. Measurements were normalized to the loading control, and data from CD at ZT0 was set to 1. Averaged data from CD and HF mice are provided as reference, indicated as black (CD) or HF (red) dashed lines. Black arrows show the phase differences in the circadian wave between HFN5 and HFN17 mice. (D) RT-qPCR determined circadian gene expression in the liver (n = 3 biological replicates per data point). Averaged data from CD and HF mice are provided as reference, indicated as black (CD) or red (HF) dashed lines.”

These data prove the hypothesis that NAD⁺ is a time-giver for the hepatic clock, as the corresponding phase advances or delays can be shown in the expression of clock components. Of note, we are aware that these phase displacements in HFN5 and HFN17 mice are not strictly of 6 hours for all cases. For example, in table S7A, phase delays (that is, moving the rhythm later) of ~6 hours or more can be shown in HFN17 for all core clock components: *Bmal1*, *Clock*, *Cry1*, *Per1* and *Per2*, and also *Noct*, while *RevErba* and *Dbp* are between 3 to 4 phase delayed. In HFN5 mice, which should push the clock earlier, phase advances of ~5 hours appear for *Rev-Erb α* , while the rest of the genes are advanced between 1-4 hours depending on the gene. Notably, these differences on the magnitude of phase displacements between HFN5 and HFN17 mice is expected, as the resistance of the clock to phase advances is a property of mammalian circadian rhythms, which is evident even using light as a time giver (Aschoff et al., 1975; Eastman and Martin, 1999; Mitchell et al., 1997; Revell et al., 2006). In fact, phase advances are more detrimental for health, and as a result, phase advances require longer and stronger exposure to the time giver (Sack, 2010; Wolff et al., 2013). This validates that NAD⁺ could qualify as a potent time-giver for the hepatic clock; however, in an effort to not overstate our conclusions, we have now toned down them along the text, as follows:

Added to lines 31-35 (abstract):

“Remarkably, timed NAD⁺ treatment adjusted the phase of circadian oscillations of the liver clock components to the extent of completely inverting their phase when mice were treated at the onset of the rest phase.”

-We have modified the title as follows: “Time-of-day defines the efficacy of NAD⁺ to treat diet-induced metabolic disease by displacing the phase of the oscillations of the hepatic circadian clock.”, which appears more precise and aligned to the data.

-We have substituted the verb “reset” for “adjust”, “reorganize”, “displace” or “shift the phase”

-We avoided stating that “NAD⁺ is a potent time-giver (or a synchronizer) for the hepatic molecular clock”, centering our description of the data to the phase advances or delays following timed NAD⁺ treatment.

R#2: *Additionally, as reviewer # 5 pointed out, metabolic gene expression presented in Figure 6E/F/G does not support the hypothesis that supplementing NAD⁺ at ZT11 enhances fatty acid oxidation to contribute to weight loss and decreased hepatic and circulating triglycerides. This is because the expression of fatty acid oxidation genes is high in all HF-fed groups, of note, the HF and HFN group are comparable in those gene expression, although the authors state that the breadth of transcriptional activity extends to ZT18 in HFN. The slight differences between the two groups (HF vs HFN) make it hard to believe that NAD⁺ works through fatty acid oxidation pathway to improve metabolic markers.*

A: Yes, this is an important information. We now provide additional data showing that increased expression of fatty acid oxidation genes in HFN mice are functional as follows:

Added / modified to lines 398-410:

“To address whether these transcriptional changes in HFN mice are functional, we measured global fatty acid oxidation rates and CPT1-dependent mitochondrial respiration in liver explants. Notably, we found a significant induction of fatty acid oxidation in HFN mice (Figure 6H), which was accompanied by increased CPT1-mediated mitochondrial respiration when palmitoyl-CoA was supplied as a substrate (Figure 6I). Accordingly, a significant rise in maximal respiration and ATP production was evident in mitochondria from HFN mice (Figure 6J), demonstrating functional implications for transcriptional variations from fatty acid oxidation genes in HFN mice. Together, these data indicate that increased hepatic NAD⁺ levels at the beginning of the active phase induce AMPK-phosphorylation and activity, favoring a transcriptional program of genes involved in fatty acid oxidation which extends through the active phase, increasing mitochondrial respiration and fatty acid consumption capacities. Together, these data indicate that increased hepatic NAD⁺ levels at the beginning of the active phase induce AMPK-phosphorylation and activity, favoring a transcriptional program of genes involved in fatty acid oxidation which extends through the active phase, increasing mitochondrial respiration and fatty acid consumption capacities, and possibly contributing to weight loss and decreased hepatic and circulating triglycerides specifically in HFN mice”

Added to Figure 6:

Added to Figure 6 caption:

(H) ^{14}C -palmitic acid oxidation was assessed by quantifying $^{14}\text{CO}_2$ release *ex vivo* from livers of HF and HFN mice (n=4 mice per group with 3-4 technical replicates. Two experimental replicates were performed with comparable results). (I) Oxygen consumption rate (OCR) was determined using Seahorse XF analyzer to assess CPT1-dependent mitochondrial respiration from n = 6 mice with 8 technical replicates each. OCR was measured in the absence or presence of etomoxir (a CPT1 inhibitor), with sequential addition of ADP (ATP precursor), oligomycin (complex V inhibitor), FCCP (a protonophore), and Rotenone/antimycin A (Rot/AA; complex III inhibitor). (J) Mitochondrial bioenergetic parameters were calculated from extracellular flux analyses: basal respiration, maximal respiratory capacity, proton leak, and ATP production.

Added to the Methods section:

“Fatty acid oxidation assay:

Fatty acid oxidation was quantitated *ex vivo* following our previous protocols⁴⁸, with subtle modifications. Briefly, 20-60 mg of fresh liver from mice sacrificed at ~ZT16 were weighed, and samples were minced and homogenized in 300 μl homogenization buffer (DMEM, 1 mM pyruvate, 1% BSA free fatty acid, and 0.5 mM palmitate) at 4°C. Then, 5 μl palmitic acid [^{14}C] 100 $\mu\text{C}/\text{ml}$ (Perkin Elmer) was added, and these lysates were incubated for 2 hr at 37°C. Eppendorf tubes were prepared containing small pieces of Whatman paper in the cap of the tube, which were wet with 20 μl NaOH 3M, while 150 μl 70% perchloric acid was placed inside the tube. Lysates were added to these tubes and incubated 1 hr at 37°C. The trapped $^{14}\text{CO}_2$ was determined transferring the filter discs to a scintillation vial with 4 ml of scintillation liquid and measuring in a Beckman LS6500 scintillation counter.

Palmitoyl-CoA oxidation assay in isolated liver mitochondria.

The Palmitoyl-CoA oxidation depends on the activity of Carnitine palmitoyltransferase-1 (CPT-1), the rate-controlling enzyme for long-chain fatty acid oxidation. Measurements of CPT-1-mediated mitochondrial respiration was determined in mitochondria freshly isolated from mouse liver using the Seahorse XFe96 Extracellular Flux Analyzer (Agilent Technologies), as previously described¹⁵⁴ with modifications as follows. Mice were sacrificed by cervical dislocation at ~ZT16, and approximately 100 mg of liver was dissected and placed on ice on mitochondrial isolation buffer (MIB1, 210 mM d-Mannitol, 70 mM sucrose 5 mM HEPES, 1 mM EGTA and 0.5% free fatty acid BSA) and centrifuged at 800 g for 10 min at 4°C. The tissue was homogenized using the Polytron tissue homogenizer at low potency for 8 seconds and centrifuged at 800 g (10 min at 4°C). The supernatant was collected in 15 ml falcon tubes and centrifuged at 8000 g (10 min at 4°C). The resulting pellet containing the mitochondria was washed three times with 1 ml of MIB1 and resuspended on 0.5 ml of mitochondrial assay solution (MAS1, 220mM d-Mannitol, 70 mM sucrose, 10 mM KH_2PO_4 , 5 mM MgCl_2 , 2 mM HEPES, 1 mM EGTA, 0.2% BSA, pH 7.2) with the addition of substrates (40 μM palmitoyl CoA, 0.5 mM malate and 0.5 mM carnitine). Total protein was determined using the Qubit® 3.0 Fluorometer and 14 μg of isolated mitochondria were diluted in MAS1 buffer with substrates with or without the presence of the CPT-1 inhibitor, etomoxir (3 μM) and loaded per well in the XFe96 plate. The plate with the containing mitochondria was centrifuged at 2000 g for 20 min at 4°C. The oxygen consumption rate (OCR) was measured with 7 technical replicates for each mouse, as the following compounds were injected to final concentrations per well: ADP (4 mM),

oligomycin (2.5 μM), carbonyl cyanide 4-(trifluoromethoxy)phenylhydrazone known as FCCP (2.0 μM) and antimycin A (1 μM) / rotenone (1 μM). Four mitochondrial respiration states were calculated: basal respiration (respiration of mitochondria with substrates but without ADP), ATP production or phosphorylating respiration (rate of ATP formation from ADP and inorganic phosphate), proton leak or non-phosphorylating respiration (rate of oxygen consumption while ATP synthase is inhibited with oligomycin), and Maximal respiration state after the addition of FCCP.”

R#2: *The data presented in Figure 7C/D/E/F/H show hepatic inversion of the clock machinery and the NAD⁺ biosynthesis and salvages genes in HFN23 mice. Moreover, inverted pattern of circulating triglycerides in HFN23 mice was also shown in Figure 5H. As those remarkable impact of NAD⁺ treatment at ZT23 is different from NAD⁺ treatment at ZT11 (HFN group), the authors need to address why the treatment at ZT23 alleviates the disease rather than exacerbates it.*

A: This is an interesting take, we understand this Reviewer’s concerns about HFN23 not aggravating the disease, yet we don’t fully share this viewpoint because rising NAD⁺ levels has been proven very effective to treat metabolic disease in both mouse and humans; thus, finding that this treatment aggravates the disease is highly unlikely. Our research demonstrates that this therapy is more effective at ZT11 than at ZT23, because at ZT23 there is a detrimental side effect consisting of inverting the hepatic clock leading to circadian misalignment. We consider that this is important because these findings constitute grounds to take a chronotherapeutic approach for NAD⁺-based therapies, to maximize effectiveness of the treatment, while minimizing possible side effects (Allada and Bass, 2021). Frequently, untimed therapies which are not fully aligned with circadian rhythms do not exacerbate the target disease and still show improvements; however, side effects hinder their effectiveness. Chronotherapies have been extensively explored for cancer treatments (Innominato et al., 2014); for example, optimally timed cancer chemotherapy with doxorubicin or pirarubicin (06:00h) and cisplatin (18:00h) improved the control of ovarian and endometrial cancer while minimizing side effects (Kobayashi et al., 2002). In small cell lung cancer patients, cisplatin at 18:00 h has advantage in relieving side effects of chemotherapy, although in this study no differences in total response were found between the routine and the chronotherapy group (Li et al., 2015) yet cancer patients benefit from optimal circadian-timed chemotherapy, even though the effectiveness of the compound is comparable independently of time. Surgical procedures can also be improved by timing, as shown for afternoon cardiac surgery which appears to provide perioperative myocardial protection and improve patient outcomes compared with morning surgery (Montaigne et al., 2018). Similarly, short-acting statins are recommended to be administered in the evening to potentiate their cholesterol lowering effects; however, if given in the morning they also reduce cholesterol although to a lesser extent (Awad et al., 2017).

Hence, the aim of chronotherapies is to significantly improve the tolerability and effectiveness of a treatment which usually works independently of time, as we show for NAD⁺ at ZT11. Optimally timed drugs can be administered in lower doses and their effects on their intended targets can be maximized. We argue that this is important in itself and constitutes the most relevant finding in our study.

We agree that uncoupling clocks leads to metabolic adversities, but NAD⁺ treatment at ZT23 opposes this metabolic derangement to some extent, at least on the short term (our intervention lasts three weeks). Probably, as this Reviewer suggests, a question remains how NAD⁺ at ZT23 opposes metabolic derangements derived from uncoupled clocks. We think that there are many beneficial effects of NAD⁺ which are non-related directly to the hepatic clock, at least in this three-weeks treatment. In fact, we already show that fatty liver is largely corrected independently of time of NAD⁺-supply (Figure 5I,J,K), supporting that idea. Because inflammation aggravates fatty liver disease (Tilg and Moschen, 2010), we approached this concern measuring hepatic cytokine levels using an Elisa system and their expression profiles by qPCR from CD, HF, HFN and HFN23 (n=3-5) mice. Our data show that both HFN and HFN23 mice improve the inflammatory environment; hereby, correcting inflammation and fatty liver are NAD⁺-directed effects which do not depend on time of treatment. However, body weight loss, improved glucose homeostasis and circulating TGs are significantly ameliorated by optimally timed treatment, preserving the phase of the hepatic clock aligned with rhythmic feeding patterns, which on the long term, might be critical for the success of this therapy. This has been included in the manuscript, as follows:

Added to lines 344-348

“Opposite, hepatic steatosis was reduced to a similar extent in HFN and HFN23 groups (Figure 5I-K, S5G-I), while inflammatory cytokines known to be increased in the liver from HFD-fed mice^{82, 83}, appeared overall reduced in obese mice treated with NAD⁺ (Figure 5L-M), evidencing that NAD⁺ ameliorates these obesity-associated physiological parameters independently of time of treatment.”

82 Stanton, M. C. et al. Inflammatory Signals shift from adipose to liver during high fat feeding and influence the development of steatohepatitis in mice. *Journal of Inflammation* 8, 8, (2011).

83 Lackey, D. E. & Olefsky, J. M. Regulation of metabolism by the innate immune system. *Nature Reviews Endocrinology* 12, 15-28, (2016).

While ref. 82 measures hepatic expression of pro-inflammatory cytokines in a mouse model of obesity closely similar to ours, ref. 83 is a nicely presented review illustrating the connections between metabolic derangements and inflammatory responses. These references sustain the selection of the cytokines and markers we measured in Fig. 5L, 5M.

Added to Figure 5:

Added to the Figure 5 caption:

“(L) Relative pro-inflammatory cytokines levels assayed by ELISA in livers from 3 mice for HF, HFN and HFN23 groups at ZT6 (2 CD mice were assayed for basal reference values) (M) RT-qPCR determined hepatic mRNA levels of pro-inflammatory cytokine genes (*Tgfb1*, *Il6*, *Il1a*, *Il1b*, *Ifng*), the macrophage recruiter gene *Csf2*, (also known as GM-CSF) and macrophage markers (*Cd11b*, *Cd11c*). n= 5-6 mice per group at ZT6”

Added to the Methods section:

“Determination of inflammatory cytokines in mouse liver:

The Mouse Inflammation ELISA Strip (Signosis, cat. No. EA-1051) was used for profiling inflammation cytokines, following the manufacturer’s instructions.”

Also, all new qPCR primers sequence have been added to Table S5.

Added to the discussion section, lines 568-577

“We have demonstrated time-of-day dependent and independent responses to NAD⁺ therapy. Clearly, as previously reported^{123,124}, rising NAD⁺ levels elicit positive responses including correction of hepatic steatosis and the inflammatory environment in obese mice, and we found that these positive effects occur independently of time of NAD⁺ supply (Figures 5I-M). In fact, these two processes are interconnected, since during obesity, inflammation in the liver happens and increased cytokines lead to overexpression of genes involved in de novo lipogenesis and ceramide biosynthesis^{125,126}. However, we also observed different responses to NAD⁺ between obese mice treated at ZT11 or at ZT23, where the latter did not fully recapitulate certain metabolic parameters, such as improvement in glucose tolerance, insulin sensitivity or circulating triglycerides.”

- 124 Katsyuba, E. et al. De novo NAD⁺ synthesis enhances mitochondrial function and improves health. *Nature* 563, 354-359, (2018).
- 125 Bikman, B. T. & Summers, S. A. Ceramides as modulators of cellular and whole-body metabolism. *The Journal of clinical investigation* 121, 4222-4230, (2011).
- 126 Obstfeld, A. E. et al. C-C chemokine receptor 2 (CCR2) regulates the hepatic recruitment of myeloid cells that promote obesity-induced hepatic steatosis. *Diabetes* 59, 916-925, (2010).

In addition, all newly generated data with these revisions has been added to the “source data” document.

References:

- Allada, R., and Bass, J. (2021). Circadian Mechanisms in Medicine. *The New England journal of medicine* 384, 550-561.
- Aschoff, J., Hoffmann, K., Pohl, H., and Wever, R. (1975). Re-entrainment of circadian rhythms after phase-shifts of the Zeitgeber. *Chronobiologia* 2, 23-78.
- Awad, K., Serban, M.C., Penson, P., Mikhailidis, D.P., Toth, P.P., Jones, S.R., Rizzo, M., Howard, G., Lip, G.Y.H., Banach, M., *et al.* (2017). Effects of morning vs evening statin administration on lipid profile: A systematic review and meta-analysis. *Journal of clinical lipidology* 11, 972-985 e979.
- Eastman, C.I., and Martin, S.K. (1999). How to use light and dark to produce circadian adaptation to night shift work. *Annals of medicine* 31, 87-98.
- Innominato, P.F., Roche, V.P., Palesh, O.G., Ulusakarya, A., Spiegel, D., and Lévi, F.A. (2014). The circadian timing system in clinical oncology. *Annals of medicine* 46, 191-207.
- Kobayashi, M., Wood, P.A., and Hrushesky, W.J.M. (2002). Circadian chemotherapy for gynecological and genitourinary cancers. *Chronobiology International* 19, 237-251.
- Li, J., Chen, R., Ji, M., Zou, S.L., and Zhu, L.N. (2015). Cisplatin-based chronotherapy for advanced non-small cell lung cancer patients: a randomized controlled study and its pharmacokinetics analysis. *Cancer chemotherapy and pharmacology* 76, 651-655.
- Mitchell, P.J., Hoese, E.K., Liu, L., Fogg, L.F., and Eastman, C.I. (1997). Conflicting bright light exposure during night shifts impedes circadian adaptation. *Journal of biological rhythms* 12, 5-15.
- Montaigne, D., Marechal, X., Modine, T., Coisne, A., Mouton, S., Fayad, G., Ninni, S., Klein, C., Ortmans, S., Seunes, C., *et al.* (2018). Daytime variation of perioperative myocardial injury in cardiac surgery and its prevention by Rev-Erba antagonism: a single-centre propensity-matched cohort study and a randomised study. *Lancet (London, England)* 391, 59-69.
- Revell, V.L., Burgess, H.J., Gazda, C.J., Smith, M.R., Fogg, L.F., and Eastman, C.I. (2006). Advancing human circadian rhythms with afternoon melatonin and morning intermittent bright light. *The Journal of clinical endocrinology and metabolism* 91, 54-59.
- Sack, R.L. (2010). Clinical practice. Jet lag. *The New England journal of medicine* 362, 440-447.
- Tilg, H., and Moschen, A.R. (2010). Evolution of inflammation in nonalcoholic fatty liver disease: The multiple parallel hits hypothesis. *Hepatology* 52, 1836-1846.
- Wolff, G., Duncan, M.J., and Esser, K.A. (2013). Chronic phase advance alters circadian physiological rhythms and peripheral molecular clocks. *Journal of applied physiology (Bethesda, Md : 1985)* 115, 373-382.

Reviewer #4 (Remarks to the Author):

NCOMMS-21-09336B

Escalante-Covarrubias et al.

"Time-of-day defines the efficacy of NAD⁺ to treat diet-induced metabolic disease by adjusting oscillations of the hepatic circadian clock"

R#4: *In this further revised version, the authors adequately addressed this reviewer's request and provided a new set of results that show differences in BMAL1 recruitment between HNF and HFN23 groups. This new data set strengthens the authors' conclusion, providing insights into the mechanism of timed NAD⁺ administration. Therefore, this reviewer now believes that the manuscript is ready to be accepted for Nature Communications.*

A: We thank this reviewer for their positive comments and criticisms which contributed to improve this manuscript.

R#4: *Having that said, the following minor textual revisions should be made before the official acceptance:*

1) In the Discussion section, the authors describe that "NAD⁺ is produced in all tissues from the salvageable precursors NAM, nicotinamide riboside (NR) or niacin." This is not correct. The only salvageable precursor is nicotinamide (NAM). The authors should use nicotinic acid (NA), instead of niacin, because niacin means both NAM and NA, and should also include nicotinamide mononucleotide (NMN) as an important NAD⁺ precursor/intermediate.

A: Agreed. We have now corrected this as follows:

Modified to lines 530-532:

"NAD⁺ is produced in all tissues from the salvageable metabolite NAM, or from precursors including nicotinamide riboside (NR), nicotinic acid (NA) or nicotinamide mononucleotide (β-NMN),"

R#4: *2) Another statement that "it is generally accepted that NR or NAM enter the cell, while extracellular NAD⁺ and NMN are converted to NR." This is not correct, either. Whereas there are still scientific arguments in the field, NAD⁺ can enter the cell through Connexin 43 hemichannels, and NMN can enter through a NMN transporter Slc12a8.*

A: Yes, this is an important point. We agree that mounting evidence sustain the presence of distinct transporters with different action mechanisms which confer NAD⁺ or NAM the ability to cross cell barriers, both plasmatic and mitochondrial membranes, and this constitutes an exhaustive area of research and as such, somehow controversial. We have revisited this paragraph in the discussion and realized that mentioned sentence does not add critical or relevant information, hereby we now have eliminated this statement in line 528.

R#4: *3) The statement in the last paragraph of the Discussion section that "in humans, clinical trials aiming to boost endogenous NAD⁺ for treatment of metabolic diseases are increasing, in*

many cases reporting conflicting results." This statement is not accurate because the results are not "conflicting", but quite clear. There have already been a significant number of clinical trials on both NR and NMN. Regarding metabolic diseases, particularly type 2 diabetes or prediabetic conditions, there has been no efficacy reported with NR, whereas NMN has been reported to increase skeletal muscle insulin sensitivity in prediabetic women.

A: Yes, this is a good point. We have now corrected this as follows:

Modified to lines 641-643:

"In humans, clinical trials aiming to boost endogenous NAD⁺ for treatment of metabolic diseases are increasing, showing promising results; for example, in postmenopausal women with prediabetes, a daily dose of NMN increases muscle insulin sensitivity ^{45,135,149}."

These two references have been included:

45 Yoshino, M. et al. Nicotinamide mononucleotide increases muscle insulin sensitivity in prediabetic women. *Science* 372, 1224-1229, (2021).

149 Zhong, O., Wang, J., Tan, Y., Lei, X. & Tang, Z. Effects of NAD⁺ precursor supplementation on glucose and lipid metabolism in humans: a meta-analysis. *Nutrition & Metabolism* 19, 20, (2022).

REVIEWERS' COMMENTS

Reviewer #2 (Remarks to the Author):

The authors adequately addressed my concerns. No further comments.